# Discovering cognitive strategies with tiny recurrent neural networks

Li Ji-An[1], Marcus K. Benna[1] & Marcelo G. Mattar[2,3] ✉

Understanding how animals and humans learn from experience to make adaptive decisions is a fundamental goal of neuroscience and psychology. Normative modelling frameworks such as Bayesian inference[1] and reinforcement learning[2] provide valuable insights into the principles governing adaptive behaviour. However, the simplicity of these frameworks often limits their ability to capture realistic biological behaviour, leading to cycles of handcrafted adjustments that are prone to researcher subjectivity. Here we present a novel modelling approach that leverages recurrent neural networks to discover the cognitive algorithms governing biological decision-making. We show that neural networks with just one to four units often outperform classical cognitive models and match larger neural networks in predicting the choices of individual animals and humans, across six well-studied reward-learning tasks. Critically, we can interpret the trained networks using dynamical systems concepts, enabling a unified comparison of cognitive models and revealing detailed mechanisms underlying choice behaviour. Our approach also estimates the dimensionality of behaviour[3] and offers insights into algorithms learned by meta-reinforcement learning artificial intelligence agents. Overall, we present a systematic approach for discovering interpretable cognitive strategies in decision-making, offering insights into neural mechanisms and a foundation for studying healthy and dysfunctional cognition.

From early symbolic models[4] to connectionist approaches[5], researchers have long sought computational models that capture the adaptive nature of animal behaviour. Normative frameworks such as Bayesian inference[1,6] and reinforcement learning[2,7] (RL) have been particularly influential, given their ability to formalize how agents accumulate knowledge from environmental interactions to make decisions, processes supported by prefrontal and striatal neural circuits[8–12]. A key advantage of these models is their simplicity, as they typically have few parameters and can be easily augmented with additional assumptions such as forgetting, choice biases and perseveration[11,13]. Yet, despite their simplicity and extensibility, they often lead to incorrect or incomplete characterizations of behaviour due to bias and researcher subjectivity[14].

Artificial neural networks offer an alternative approach for modelling adaptive behaviour. They impose fewer structural assumptions, require less handcrafting and provide a more flexible framework for modelling behaviour and neural activity[15,16]. A common approach involves adjusting network parameters to produce optimal task performance. This approach has been used to explain neural activity across many neuroscience domains, including vision[17,18], navigation[19–21], memory, learning, decision-making and planning[22–26]. Neural networks can also be trained to predict observable behaviour. Owing to their high parameter count, this approach often results in highly accurate behavioural predictions[27–30]. However, this increased flexibility can limit interpretability, hindering the identification of underlying cognitive and neural mechanisms.

Here we present a novel modelling framework that combines the flexibility of neural networks with the interpretability of classical cognitive models. Our framework involves fitting recurrent neural networks (RNNs) to the behaviour of individual subjects in reward-learning tasks. The fitted RNNs describe how subjects learn from environmental interactions and use that knowledge for decision-making. Unlike prior work, our framework uses very small RNNs, often composed of just 1–4 units, which facilitates their interpretation. Across eight datasets, we show that tiny RNNs outperform classical cognitive models of equal dimensionality in predicting human and animal choices. Crucially, we interpret the mechanisms that underlie these choices by visualizing the RNN dynamics as discrete dynamical systems. This framework reveals several novel behavioural patterns that are overlooked by classical models in these tasks, including variable learning rates, state-dependent perseveration and novel forms of value updating and choice biases. Our results show that tiny RNNs not only improve behavioural predictions but also provide deeper insights into cognitive mechanisms, addressing both the interpretability challenges of larger neural networks and the subjectivity of classical models.

We analysed decision-making behaviour in six widely studied reward-learning tasks, three performed by animals and three performed by humans (Figs. 1 and 2). These tasks capture fundamental processes by which animals and humans learn to make decisions through environmental interactions, which our framework aims to describe. We first present results from the three animal tasks: reversal learning[10], two-stage[12] and transition-reversal two-stage task[11] (Fig. 1d). In each

[1]Department of Neurobiology, School of Biological Sciences, University of California San Diego, La Jolla, CA, USA. [2]Department of Psychology, New York University, New York, NY, USA. [3]Center for Neural Science, New York University, New York, NY, USA. ✉e-mail: marcelo.mattar@nyu.edu

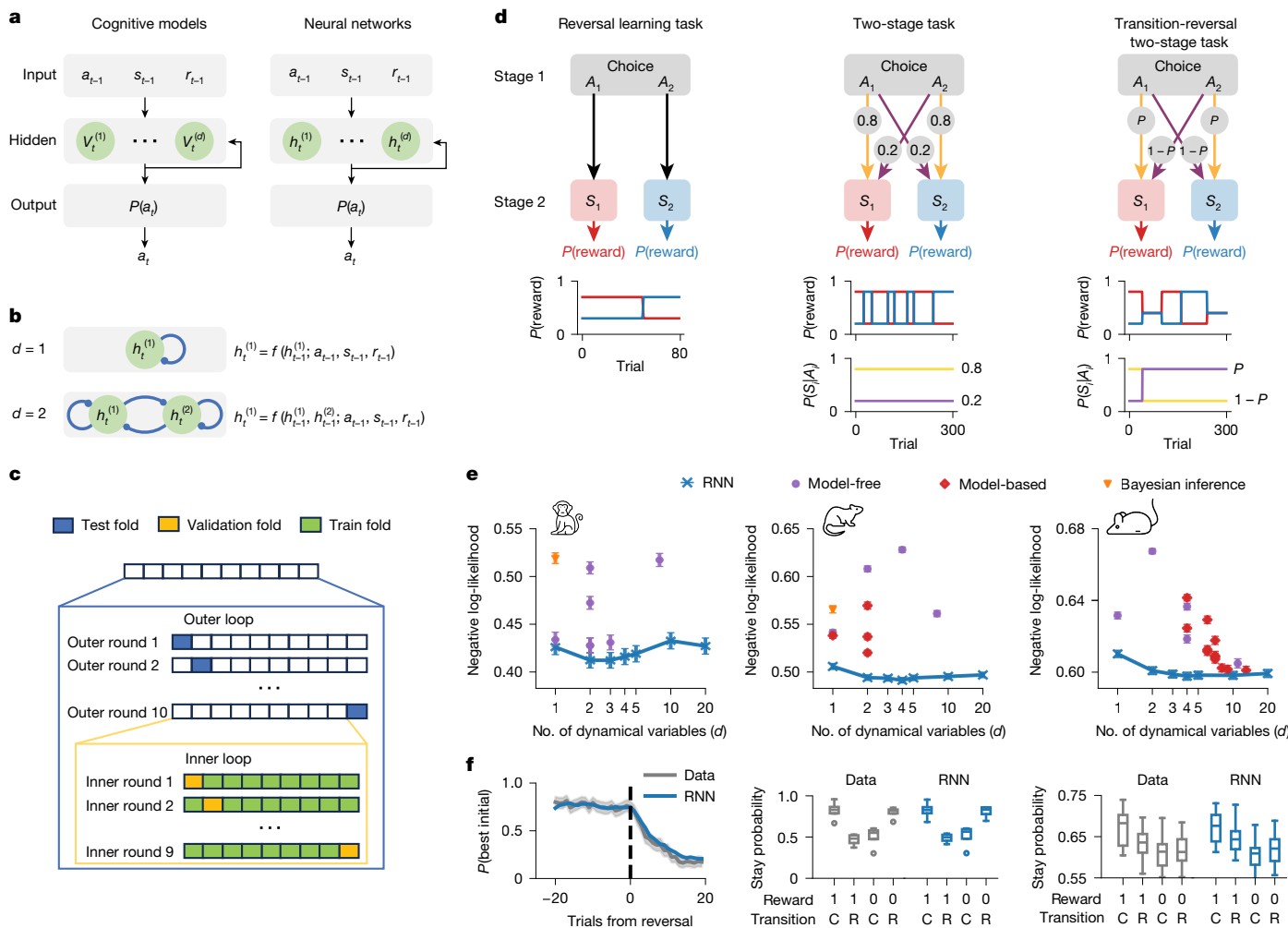

**Fig. 1 | Model overview and performance on animal tasks. a**, Cognitive models and neural networks have similar architectures. Inputs (previous action $a_{t-1}$, state $s_{t-1}$, reward $r_{t-1}$) update $d$ dynamical variables, which determine output (action probability $P(a_t)$) via softmax. Models are optimized to predict observed actions. **b**, Hidden units compute functions of inputs and previous state $h(t-1)$. Top, $d = 1$. Bottom, $d = 2$. **c**, Nested cross-validation. The whole dataset is split into ten folds of consecutive trials. Each outer loop round uses one fold for testing and nine for the inner loop. Each inner loop round uses one fold for validation (for example, hyperparameter tuning) and eight for training. **d**, Task structures: subjects choose action $A_1$ or $A_2$ at the choice state, transitioning into one of two second-stage states, $S_1$ or $S_2$, which probabilistically yield a reward. Reward probabilities change over time. **e**, Model performance (cross-validated trial-averaged negative log-likelihood; lower is better) versus number of dynamical variables $d$. Identical markers within a plot represent different variants of a model class. Error bars represent s.e.m. across 10 outer rounds, averaged over animals in each task. Left, in the reversal learning task with monkeys ($n = 2$), the two-unit RNN performed the best. Middle, in the two-stage task with rats ($n = 4$), the two-unit RNN performed not significantly worse than the best. Right, in the transition-reversal two-stage task with mice ($n = 17$), the four-unit RNN performed the best. **f**, RNN reproduction of behavioural metrics. Left, probability of choosing high-reward action pre-reversal (reversal learning, $d = 2$ RNN), presented as mean ± 95% confidence interval (across 190 blocks from 2 monkeys). Middle, probability of taking the same action (stay probability) following each trial type (two-stage task, $d = 2$ RNN). Transition: C, common; R, rare. $n = 4$. Right, stay probabilities (transition-reversal task, $d = d_*$ GRU). $n = 17$. Box plots show median (centre line) and 25th–75th percentiles (box edges), and whiskers extend to 1.5× interquartile range; points represent outliers.

task, subjects choose between actions $A_1$ and $A_2$, resulting in state $S_1$ or $S_2$. Each state carries a reward probability, which switches unpredictably during 'reversal' moments. The goal is to choose the action that is most likely to yield rewards. In the reversal learning task, each action leads deterministically to one state (for example, $A_1 \rightarrow S_1$, $A_2 \rightarrow S_2$). The two-stage task introduces probabilistic transitions (for example, $A_1 \rightarrow S_1$ with probability 0.8, $A_1 \rightarrow S_2$ with probability 0.2). The transition-reversal two-stage task adds stochastic reversals in action-state transitions (for example, $A_1$ leads to $S_1$ and $S_2$ with high and low probabilities, respectively, with these probabilities occasionally switching). We analysed reversal learning data from two monkeys (Bartolo dataset[10]) and ten mice (Akam dataset[11]); two-stage data from four rats (Miller dataset[12]) and ten mice (Akam dataset[11]); and transition-reversal two-stage data from seventeen mice (Akam dataset[11]).

We used RNNs to predict animal choices and to interpret the underlying cognitive mechanisms. As a benchmark, we also implemented more than 30 classical cognitive models previously used in these tasks, including Bayesian inference models, RL models, and many of their variants (see Methods). Notably, cognitive models and RNNs share the same input–output structure (Fig. 1a). Inputs include the agent's previous action $a_{t-1}$, second-stage state $s_{t-1}$, and reward $r_{t-1}$ (the current state is always the 'choice state', and thus omitted). These inputs update the agent's internal state, represented by a set of dynamical variables summarizing the agent's prior experience (for example, action values and beliefs) to guide future actions. The output is the agent's policy, specifying the probability of each action.

Despite sharing the same input–output structure, each model utilizes a distinct function for updating dynamical variables, leading to

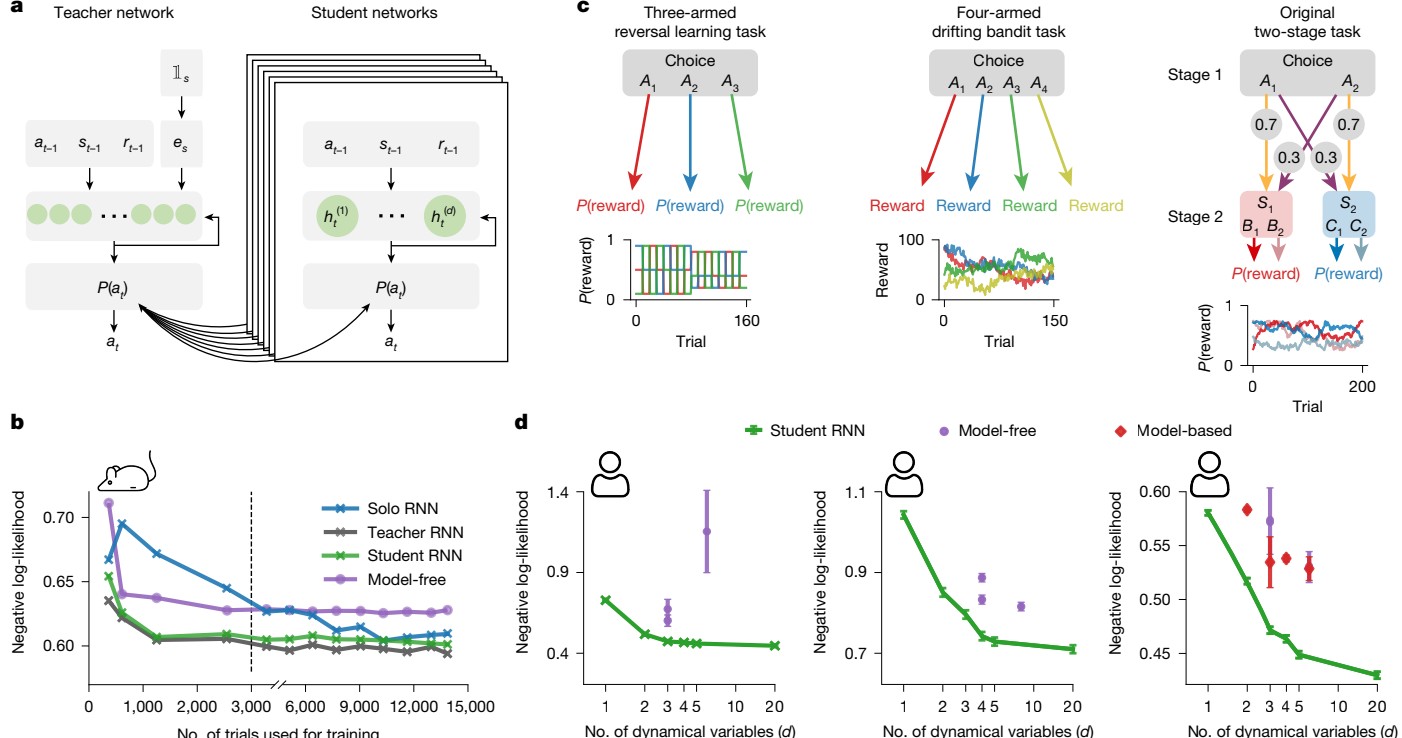

**Fig. 2 | Model performance using knowledge distillation. a**, Knowledge distillation framework: a large teacher network is trained on data from multiple subjects; the subject ID corresponding to each datapoint, provided as input, enables subject-specific embeddings $\mathbf{e}_s$ to be learned. A tiny student network is then trained on single-subject data to match the teacher network's output probabilities. **b**, Model predictive performance for a representative mouse in the transition-reversal two-stage task, across varying dataset sizes. The student RNN outperforms the best model-free RL model for all dataset sizes. Note that different $x$-axis scales are used below and above 3,000 trials. **c**, Human task structures. Left, three-armed reversal learning. Participants choose between actions $A_1$–$A_3$, each associated with a reward probability that changes over time. Middle, four-armed drifting bandit. Participants choose between actions $A_1$–$A_4$, with each associated with a 0–100 reward that fluctuates over time. Right, original two-stage task. Participants first choose action $A_1$ or $A_2$ at the choice state, transitioning into one of two second-stage states, $S_1$ or $S_2$. Participants then choose action $B_1/B_2$ or $C_1/C_2$, each probabilistically yielding a reward. Reward probabilities change over time. **d**, Performance of student RNN (cross-validated trial-averaged negative log-likelihood; lower is better) versus number of dynamical variables $d$, averaged over participants using the interspersed split protocol. Tiny RNNs outperform classical models in all tasks. Identical markers within a plot represent different variants of a model class. Error bars represent s.e.m. across individuals in each task. Left, three-armed reversal learning ($n = 1,010$). Middle, four-armed drifting bandit ($n = 918$). Right, original two-stage task ($n = 1,961$).

different representations and policies. Classical cognitive models fall into three families: model-free RL, model-based RL, and Bayesian inference. In RL models, dynamical variables represent action values: model-free RL updates these values directly from rewards; model-based RL updates them indirectly through a state-transition model. In Bayesian inference models, dynamic variables represent belief states updated via inference. We implemented standard versions of these models along with extensions accounting for factors such as choice perseveration or learned state-transition probabilities (Methods). In RNNs, each network unit corresponds to one dynamical variable that, through the network training procedure, comes to represent a potentially complex function of inputs and past activities (Fig. 1b). These functions are determined by adjustable network weights, enabling diverse mappings from past observations to policies. Our RNNs used gated recurrent units (GRUs), which selectively process past information via gating[31], although other recurrent architectures are also applicable (see Discussion).

## Predicting choices with tiny RNNs

We fit all RNNs and cognitive models to predict animal choice data by maximum likelihood (that is, minimizing cross-entropy). Owing to large differences in parameter counts between RNNs (for example, 40–80 for 1–2 units) and cognitive models (for example, 2–10 parameters; Supplementary Fig. 1), we used nested cross-validation, which

separates trials for training, validation and evaluation (Fig. 1c; see Methods for why Akaike information criteria (AIC) or Bayesian information criteria (BIC) are unsuitable). Model performance was averaged across subjects in each dataset. Our primary focus was comparing models with the same number of dynamical variables ($d$) to summarize past experiences and specify policies. Note that dynamical variables differ from model parameters: dynamical variables evolve over time, representing the agent's current beliefs and guiding actions, whereas model parameters are fixed, specifying how dynamical variables are updated.

We found that RNNs outperformed all classical models and variants in predicting animal choices across all datasets (Fig. 1e for three datasets and Supplementary Fig. 2 for two additional datasets, evidenced by the lowest scores in each plot achieved by RNNs; see Supplementary Fig. 11a–c for accuracies). Notably, RNNs outperformed even ideal Bayesian observer models, which assume exact inference with full task knowledge, indicating that animal behaviour in these tasks is suboptimal (Supplementary Discussion 2.1). Group-level performance was highest for very small RNNs—two-unit RNNs in reversal learning and two-stage tasks, and four-unit RNNs in the transition-reversal two-stage task. These small RNNs performed statistically equivalently to larger networks. For individual subjects, performance was also highest for similarly small RNNs (Supplementary Fig. 3). Crucially, each RNN with $d$-units outperformed all classical models with $d$ dynamical variables (Fig. 1e, evidenced by the absence of points below the blue line). Fitted RNNs also reproduced key behavioural metrics: choice probabilities

around reversals in the reversal learning task[10], and stay probabilities in two-stage and transition-reversal two-stage tasks[11,12] (Fig. 1f). Finally, fitted RNNs produced consistent predictions across model instances trained on different subsets of data, suggesting that discovered strategies are stable signatures of subject behaviour (Supplementary Fig. 13). These results demonstrate that tiny RNNs are versatile behavioural models, capturing more variance and reproducing established patterns better than classical models.

Adding dynamical variables to tiny RNNs did not always improve predictions, suggesting that animal behaviour in these tasks is low-dimensional. The dimensionality of behaviour ($d_*$) is defined as the minimal number of functions of the past required to optimize the predictability of future behaviour[3,32] (Supplementary Discussion 2.3). In the reversal learning and two-stage tasks, RNN predictions were optimized with just 1 or 2 units (Supplementary Fig. 3a–d), suggesting that most individual behaviours have dimensionality $d_* = 1$ or $d_* = 2$. Similarly, in the transition-reversal two-stage task, individual animal behaviour has dimensionality ranging from $d_* = 1$ to $d_* = 4$ (Supplementary Fig. 3e). In some cases, predictive performance declined with added dynamical variables, indicating insufficient data for fitting more flexible models. These low $d_*$ values are expected in such simple tasks with binary choices and a single choice state in these tasks. Although such tasks are extremely common in neuroscience and psychology for their experimental control and interpretability, we demonstrate below that tiny RNNs remain effective in more complex, higher-dimensional scenarios, highlighting their versatility.

## Flexibility and data requirements

The superior performance of tiny RNNs stems from the increased flexibility afforded by a much larger number (4–40 times) of free parameters. This flexibility allows RNNs to capture a wider range of behaviours than classical models. To evaluate the limits of this flexibility, we simulated RL and Bayesian agents in the reversal learning and two-stage tasks and fitted both RNN and cognitive models to these synthetic data (Supplementary Results 1.1 and Supplementary Figs. 4–7). Tiny RNNs matched the predictive performance of the ground-truth model that generated the behaviour, with the best RNN having the same dimensionality as the ground-truth model. These results suggest that RNNs form a superset of classical models, despite using a single architecture across tasks with minimal manual design. They also demonstrate that cognitive strategies are identifiable and robustly recoverable (Supplementary Results 1.1 and Supplementary Figs. 12 and 14), and that our training procedure prevented overfitting (had overfitting occurred, we would have observed models outperforming the ground-truth on training data while underperforming on test data).

A caveat of RNN flexibility is their higher data requirement compared with simpler models. When sufficient data are available, RNNs can continue improving beyond the plateau of cognitive models (Supplementary Fig. 8). With scarce data, however, training may not adequately constrain RNN parameters. Indeed, RNN performance declined rapidly with less data, eventually falling below that of simpler, more data-efficient cognitive models. Specifically, 500–3,000 trials per subject were required for training and validation before RNNs outperformed cognitive models (Supplementary Fig. 8). Although datasets of this size are typical in animal experiments, this presents challenges for human studies, which use less data but more participants to achieve statistical power.

To address this challenge, we developed a knowledge distillation framework that uses data from multiple subjects to improve individual-level RNNs[33]. The framework requires training a large 'teacher' model on all subjects, then using it to train smaller, more interpretable 'student' RNNs for individuals (Fig. 2a). Each student RNN is trained on data from one subject to match the teacher's probabilistic policy (which captures decision patterns across all subjects) rather than to predict binary choices. In the transition-reversal two-stage task (see a representative mouse in Fig. 2b), tiny student RNNs trained via distillation outperformed the best cognitive models (model-free RL with the same dimensionality) with just 350 trials per subject, compared to the 3,000 trials that were required without knowledge distillation ('solo RNNs'). This demonstrates that tiny RNNs, when leveraging multi-subject data, can outperform classical models even with dataset sizes typical in human experiments.

## Predicting choices in human tasks

We next tested our framework on human decision-making tasks, which typically involve fewer trials per participant and more complex designs than animal studies. We applied our method to three commonly studied tasks in cognitive neuroscience: a three-armed reversal learning task (three actions), a four-armed drifting bandit task (four actions and continuous rewards), and the original two-stage task (six actions and three choice states), each with 150–200 trials per participant (Fig. 2c). Given the limited per-participant data, we used knowledge distillation and an interspersed split protocol for training and evaluation (Methods; also see the cross-participant split protocol in Supplementary Fig. 41). As before, we compared tiny RNNs to more than ten established cognitive models.

Student RNNs with 5–20 units provided the best fit to human behaviour despite limited trials per participant (Fig. 2d), with 50 units offering similar performance (Supplementary Fig. 9). The higher dimensionality is likely to stem from higher task complexity (with more actions, states or continuous rewards) and the inherently richer cognitive processes of human participants (see Supplementary Fig. 10 for the distribution of dimensionality). Notably, even 2- to 4-unit RNNs outperformed all cognitive models of equal dimensionality (Fig. 2d; see Supplementary Fig. 11d–f for accuracies). These results demonstrate that tiny RNNs efficiently capture complex behaviour and outperform cognitive models across diverse tasks.

## Interpreting one-dimensional models

A major advantage of smaller RNNs is their potential to yield mechanistic insights that larger networks often obscure. Leveraging this advantage, we developed an interpretative framework grounded in the theory of discrete dynamical systems, which describes how a system's state (for example, dynamical variables in RNNs or cognitive models) changes over time as a function of inputs (rewards and past actions). We illustrate this framework first in models containing a single dynamical variable ($d = 1$; Fig. 3a). Here, the state at trial $t$ is fully characterized by the policy's logit (log odds) ($L(t) = \log(\Pr_t(A_1)/\Pr_t(A_2))$), representing the agent's current action preference. State evolution is given by the logit change ($\Delta L(t) = L(t+1) - L(t)$), representing how inputs alter preferences. The 'logit' and 'logit change' under 'inputs' is analogous to a mechanical system's 'position' and 'velocity' under 'forces'. Thus, visualizing $L(t)$ and $\Delta L(t)$ for each trial in phase portraits reveals important insights about the system's behaviour.

We first compared the phase portraits of a one-dimensional model-free RL and a one-dimensional Bayesian model, each fitted to reversal learning data from a monkey (Fig. 3b). In model-free RL, rewards for action $A_1$ (dark blue) lead to positive logit changes, whereas rewards for $A_2$ (dark red) lead to negative logit changes, increasing and decreasing preference for $A_1$ over $A_2$, respectively. Unrewarded actions (light blue and light red) cause smaller preference changes in model-free RL. By contrast, the Bayesian model treats an unrewarded action as equivalent to a reward for the other action. We also examine fixed points ($L_I^*$), where preferences remain unaffected by a given input ($\Delta L_I = 0$). Model-free RL exhibits three types of stable fixed points (attractors):

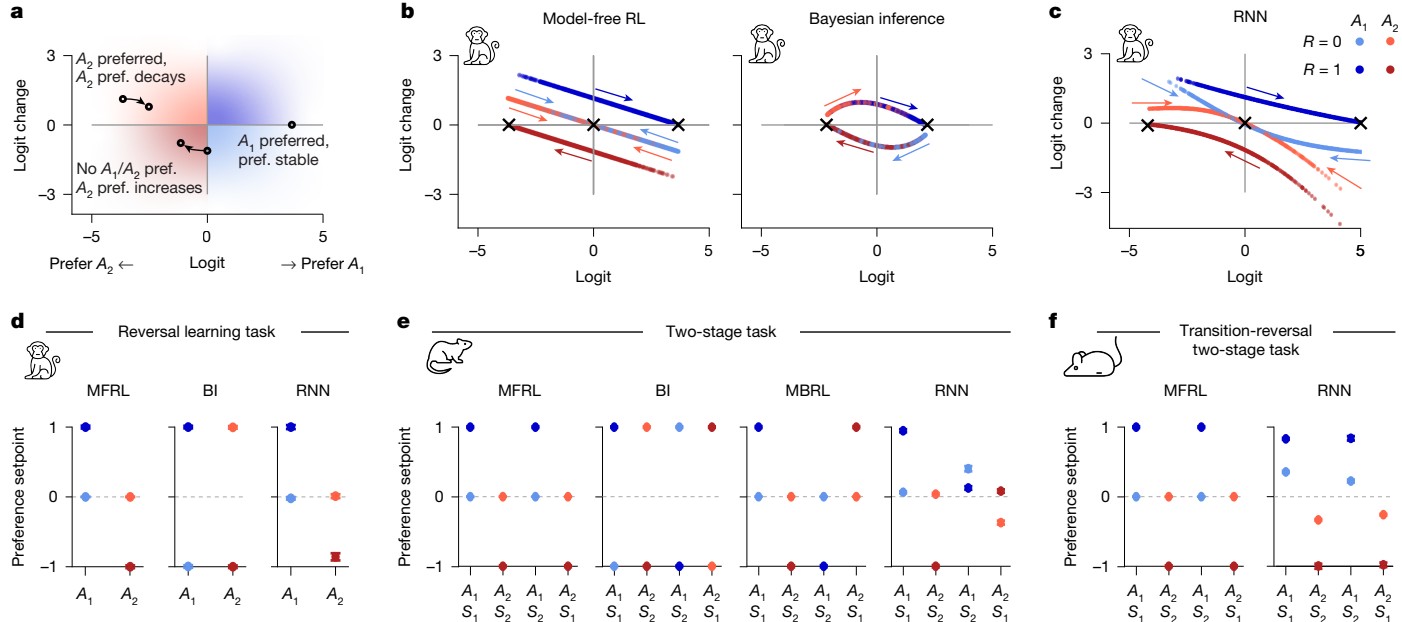

**Fig. 3 | Dynamical systems analyses for interpretation and comparison of one-dimensional models. a**, Schematic showing how the agent's preference (pref.) evolves over consecutive trials. When action $A_2$ is favoured (logit $L(t) < 0$), a positive logit change ($\Delta L(t) > 0$) results in a reduced preference for $A_2$. When neither action is favoured (indifference; logit $L(t) = 0$), a negative logit change ($\Delta L(t) < 0$) results in a preference for $A_2$. A preference level associated with a zero logit change ($\Delta L(t) = 0$) is a stable fixed point. Colours indicate the preferred action (light and dark blue, $A_1$; light and dark red, $A_2$) and preference change (darker, increase; lighter, decrease). **b,c**, Phase portraits illustrating how action preferences change as a function of current preferences (logit), action taken ($A_1$, blue; $A_2$, red) and reward received ($R = 0$, light; $R = 1$, dark).

Points represent trials and are coloured according to the trial input; coloured arrows indicate the flow direction of the model state (logit) after receiving the corresponding input. **b**, Two one-dimensional cognitive models fitted to the choices of one monkey in the reversal learning task. **c**, One-unit GRU fitted to the same monkey data. **d–f**, Preference setpoints ($u_I$), long-term action preferences after repeated exposure to input $I$ (or, analogously, the instantaneous effect of $I$ on normalized preferences). MFRL, model-free RL; MBRL, model-based RL; BI, Bayesian inference. Colours reflect input types as in **c**. Dots show mean values and error bars represent s.d. across ten outer rounds in nested cross-validation. **d**, Reversal learning task; monkey data. **e**, Two-stage task; rat data. **f**, Transition-reversal two-stage task; mouse data.

high preference for $A_1$, high preference for $A_2$, and indifference. Bayesian inference has two types of attractors—high preferences for either action—as the reward–action symmetry prevents convergence to indifference. The shape of the portraits (straight or curved) illustrates how each model processes unexpected rewards. When a disfavoured action is rewarded (dark blue dots with extreme negative logits), model-free RL shows large state changes (due to high prediction error), and a Bayesian model shows small state changes (due to strong priors at extreme logits).

Next, we analysed the phase portraits of tiny RNNs to uncover cognitive processes underlying animal behaviour. Since RNNs are capable of reproducing model-free RL and Bayesian strategies (Supplementary Results 1.1 and Supplementary Figs. 4–7), their phase portrait can reveal signatures of either cognitive model. We found that the phase portrait of a one-unit RNN fitted to the same monkey data showed multiple model-free RL characteristics (Fig. 3c; see Supplementary Fig. 15g for another monkey). For example, unrewarded trials drove the system towards indifference (attractor at $L = 0$); unexpected rewards caused large preference changes (large positive logit changes for dark blue and red dots in the left or right region); and unrewarded actions were not treated as rewards for the unchosen action (non-overlapping light and dark curves for different actions). This suggests that the behaviour of the monkey more closely resembles model-free RL than Bayesian inference. Although conventional model comparisons could reach similar conclusions, phase portraits offer more nuanced insights into how monkeys' behaviour maps onto each model.

Beyond known model signatures, phase portraits also reveal entirely novel signatures. For example, curved lines suggest a non-constant, state-dependent learning rate (Fig. 3c). The decoupling of $R = 0$ curves

(light blue and light red) at extreme logits suggests a peculiar pattern of state-dependent choice perseveration. The asymmetry between the two non-zero fixed points suggests a reward-dependent choice bias favouring $A_1$ over $A_2$ (Fig. 3c). These three signatures—'state-dependent learning rate', 'state-dependent perseveration' and 'reward-dependent bias'—are absent from all cognitive model variants and most prior analyses of these tasks. Some signatures were found across animals, whereas others were individual-specific (for example, Supplementary Fig. 17a), highlighting the value of individual-level modelling. Crucially, we validated these insights via targeted hypothesis testing (Supplementary Results 1.2 and Supplementary Fig. 15) and a novel behaviour-feature identifier approach (Methods and Supplementary Fig. 29).

Although comprehensive, phase portraits can be challenging to interpret, especially for tasks with many (or infinite) inputs. In such cases, we can simplify and summarize portrait dynamics using preference setpoints. A preference setpoint $u_I$ for input $I$ represents an agent's normalized, asymptotic action preference after repeated exposure to input $I$ (that is, $u_I = L_I^* / \max_I L_I^*$). Preference setpoints also indicate the instantaneous directional effect of input $I$ on the system's state. Essentially, $u_I$ summarizes the effect of input $I$: $|u_I| = 1$ implies maximum preference; $u_I = 0$ implies indifference. We computed $u_I$ for all fitted models across the three animal tasks (Fig. 3d–f). In the two-stage task, the RNN's preference setpoints revealed a pattern of 'reward-induced indifference', in which rewards following rare transitions led to indifference ($|u_I| \approx 0$) instead of increased preference for a specific action (Fig. 3e; dark blue marker for $A_1$, $S_2$ and the dark red marker for $A_2$, $S_1$). A similar but weaker effect ($|u_I| < 1$) was found in the transition-reversal two-stage task (Fig. 3f). Notably, this effect was found in several rats and mice (Supplementary Fig. 16), despite being absent from all cognitive models and the literature at large. Notably, its strength correlated

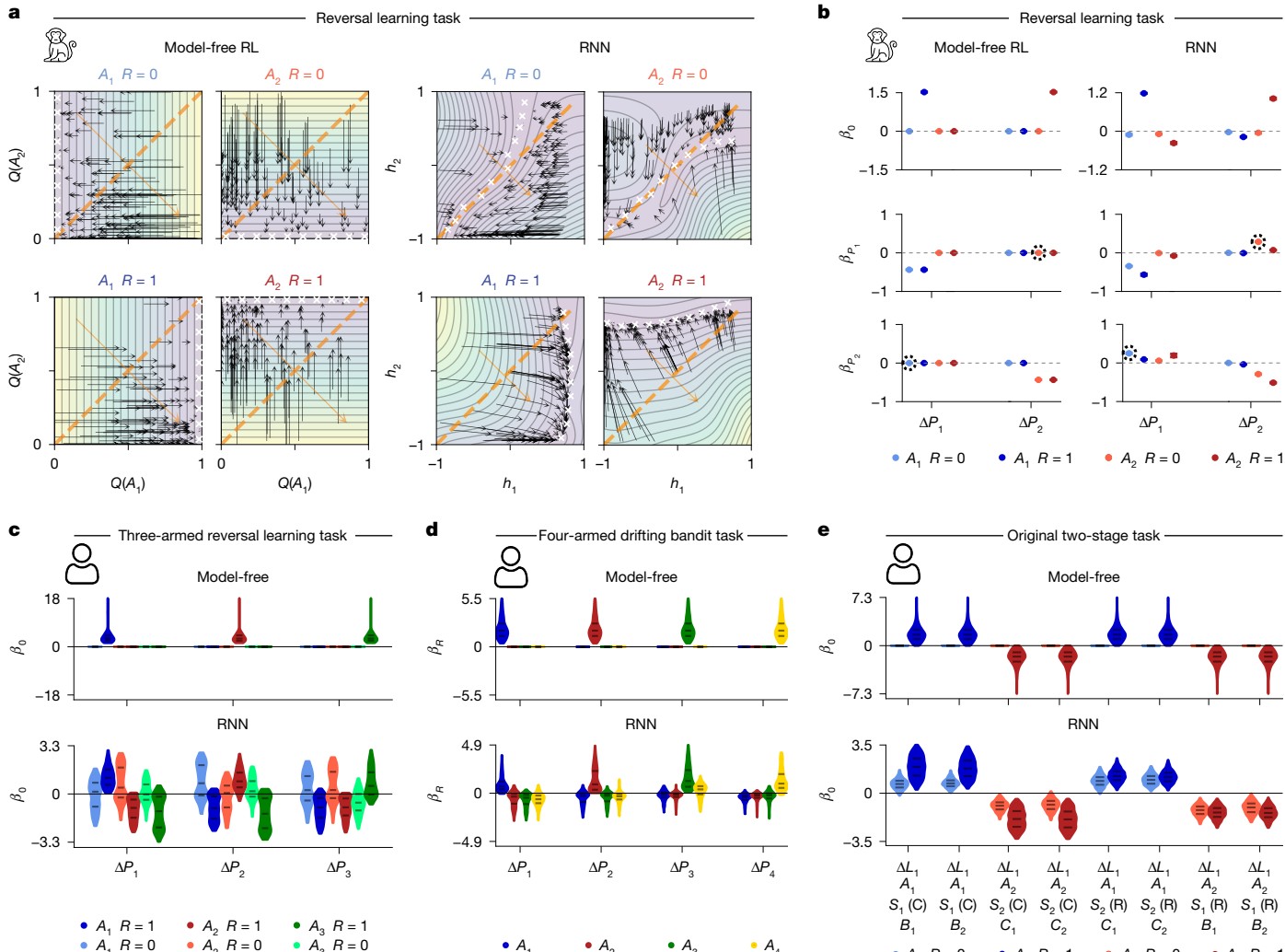

**Fig. 4 | Dynamical systems analyses for interpretation and comparison of multi-dimensional models. a**, Vector field analysis of two-dimensional models fitted to a monkey's choices in the reversal learning task. Each panel illustrates the effect of one input on state variables (axes). Black arrows, flow lines indicating state changes per trial; white crosses, attractor states; dashed lines, indifference states; orange arrows, readout vectors; background colour, dynamics speed (purple, slow; green, medium; yellow, fast). Left, model-free RL model. Axis-aligned arrows indicate that only chosen action value updates, converging to reward magnitude. Right, two-unit RNN. Top, in unrewarded trials ($R = 0$), arrows converge to the diagonal, suggesting a drift-to-the-other pattern. **b**, Dynamical regression analysis for two-dimensional model-free RL (left) and two-unit RNN (right). $P_i$ and $\Delta P_i$ are preference and preference change for action $A_i$, respectively. Coefficients describe how preferences change independently of one's current preferences (baseline $\beta_0$) as a function of

current preference for the same action ($\beta_{P_i}$) and current preference for other action ($\beta_{P_j}$). Dots show mean values and error bars represent s.d. across ten outer rounds in nested cross-validation. Circled coefficients indicate a drift-to-the-other pattern. **c**–**e**, Dynamical regression analysis for a model-free RL and RNN models fitted to human behaviour. Violin plots show distributions of participant-level coefficients. **c**, Three-dimensional models fitted to data from three-armed reversal learning task. $\Delta P_i$: preference change for action $A_i$. $\beta_0$ is the constant coefficient in the regression with $\Delta P_i$ as target. **d**, Four-dimensional models fitted to data from the four-armed drifting bandit task. $\Delta P_i$ is preference change for action $A_i$ and $\beta_R$ is the coefficient for the continuously valued reward in the regression with $\Delta P_i$ as target. **e**, Three-dimensional models fitted to data from the original two-stage task. $\Delta L_i$ is the logit change for each task state ($\Delta L_1$ for the first stage), $\beta_0$ is the constant coefficient in the regression with $\Delta L_1$ as target. C, common transition; R, rare transition.

with better task performance across animals ($\rho = 0.62$, $P = 0.008$; Supplementary Fig. 18), demonstrating that discovered patterns have meaningful behavioural relevance.

## Interpreting multi-dimensional models

We next extended our interpretive framework to models with more than one dynamical variable ($d > 1$). Although phase portraits still apply, the policy logit no longer fully characterizes the state of the system. For models with $d = 2$, we use two-dimensional vector fields, where axes represent dynamical variables and arrows show trial-by-trial state updates. We examined the vector fields of a two-dimensional

model-free RL model and a two-unit RNN with diagonal readout (Methods), both fitted to a monkey's reversal learning data (Fig. 4a,b and Supplementary Fig. 21). In model-free RL, arrows are axis-aligned, indicating that only the value of the chosen action is updated. In the RNNs, arrows were slightly tilted in rewarded trials, indicating decay (forgetting) of the unchosen action value (Fig. 4a). Both models showed line attractor dynamics, with arrows converging to a line (white crosses; Fig. 4a). Notably, in the RNN, arrows in unrewarded trials converged to the diagonal ($h_1 = h_2$), suggesting that values of an unrewarded action drifted towards the value of the alternative action, and not towards zero as expected. We validated this peculiar 'drift-to-the-other' effect by augmenting model-free RL with a drift-to-the-other forgetting, finding

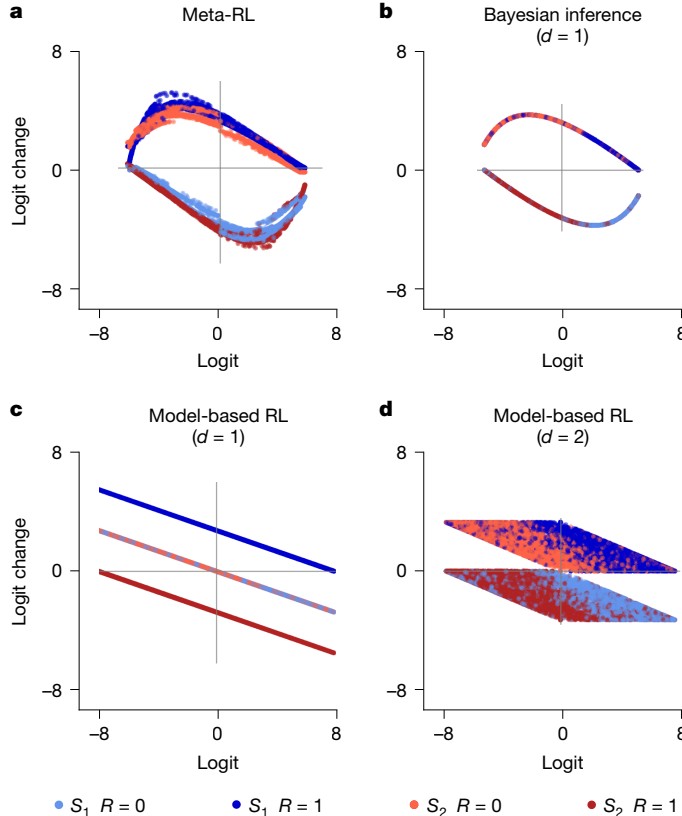

**Fig. 5 | Logit analysis of meta-RL agent and cognitive models in the two-stage task. a**, Meta-RL agent. **b**, One-dimensional Bayesian inference model fitted to meta-RL agent behaviour (includes task-optimal Bayesian model). **c**, One-dimensional model-based RL model fitted to meta-RL agent behaviour. **d**, Two-dimensional model-based RL model[22] fitted to meta-RL agent behaviour. Logit change patterns suggest similarities between meta-RL agent and Bayesian inference model, and differences relative to model-based RL models and observed animal strategies (compare with Supplementary Fig. 16).

Legend: $S_1$ $R = 0$ · $S_1$ $R = 1$ · $S_2$ $R = 0$ · $S_2$ $R = 1$

that it outperformed models with conventional 'drift-to-zero' forgetting and nearly matched the two-unit RNN (Supplementary Fig. 22).

To interpret models with $d > 2$, we introduce an alternative method: dynamical regression. This method approximates one-step state dynamics (for example, logit change) as a linear function of the current state (for example, logit). Applied to the above RNN, regression coefficients indicate that the unrewarded chosen action values are positively influenced by the unchosen action values, confirming the drift-to-the-other pattern (Fig. 4b). We then applied this method to the three human tasks above (Fig. 2c), performing separate regressions on participant-specific distilled RNNs. Distributions of regression coefficients revealed novel insights into human behaviour: (1) in the three-armed reversal learning task, trial outcomes affect preferences for all actions, but differently for chosen versus unchosen actions (Fig. 4c); (2) in the four-armed drifting bandit task, rewards sometimes reduce unchosen action values (Fig. 4d); and (3) in the original two-stage task, rewarded action values drift towards non-zero setpoints depending on transition types (Fig. 4e). These effects are absent in classical models, but augmenting them with RNN-derived mechanisms improves performance (Supplementary Results 1.4, 1.5 and 1.6 and Supplementary Figs. 30–38).

## Interpreting task-optimized networks

Our interpretative framework can also analyse larger neural networks that are trained for optimal task performance in decision-making tasks,

offering a new way to compare biological and artificial behaviour. To illustrate, we trained RNN agent via meta-RL to maximize rewards in a two-stage task (Fig. 5 and Supplementary Fig. 27). Meta-RL agents can learn this task but employs an unidentified decision strategy. Whereas previous work interpreted the agents' 'stay probability' and 'prediction errors' patterns as evidence for model-based RL[22], such patterns could arise from alternative strategies[34].

To analyse meta-RL agents, we compared their phase portraits with those of various cognitive models. We found that the dynamics of meta-RL agent (Fig. 5a) resembled a one-dimensional Bayesian agent (Fig. 5b), not a model-based RL strategy (Fig. 5d). Its dynamics also differ from those observed in animals performing the same task (Supplementary Fig. 16). Notably, before convergence, the agent's representations differed substantially from any known cognitive models (Supplementary Fig. 28b). Even after convergence, we identified subtle deviations from the Bayesian model (Fig. 5a,b): each logit value in the meta-RL agent corresponded to a range of logit change values (Fig. 5a, parallel lines), unlike the single logit change in the Bayesian inference. These parallel curves reflect a 'history effect', whereby exact Bayesian inference is distorted by the representation of input history (Supplementary Fig. 28c). This illustrates how our approach reveals computational processes in both biological and artificial systems.

## Discussion

We introduced a novel method for modelling decision-making using tiny RNNs. Across six reward-learning tasks and eight animal and human datasets, tiny RNNs outperformed classical models in predicting individual subject behaviour. Despite their small size (often just 1 to 4 units), these RNNs demonstrated superior predictive power while requiring fewer manual assumptions than traditional cognitive models. Their flexibility stems from the increased number of free parameters, allowing them to model a broader behavioural repertoire than classical models, including normative behaviour such as RL and Bayesian inference. For human studies with limited per-participant data, our knowledge distillation approach leveraged multi-participant data to achieve excellent performance with only a few hundred trials per participant. This significantly expands our method's applicability, particularly for human studies in cognitive neuroscience and computational psychiatry.

Our interpretative framework, grounded in dynamical systems theory, transforms these tiny RNNs from black boxes to mechanistically transparent models of cognition. By visualizing how internal states evolve in response to rewards and actions, we uncovered several novel cognitive strategies without requiring extensive model comparisons. These include state-dependent learning rates, reward-induced indifference effects, and unique patterns of value updating and choice biases. For higher-dimensional models, our dynamical regression method effectively summarizes system dynamics. This unified approach enables direct comparison between RNN-learned strategies and those assumed by classical cognitive models. We validated these strategies using standard model recovery and validation techniques[13].

Our approach offered robust and statistically significant improvements over traditional methods. Although the improvements in some cases were numerically modest, the value of our approach lies in its consistent success in discovering interpretable patterns across various species and tasks. Additionally, the same strategies were often observed across individuals (Supplementary Fig. 26), suggesting that they reflect general learning strategies rather than task-specific effects. Several strategies identified by the RNNs had been overlooked by cognitive models tailored to these tasks. They do, however, reflect established principles of learning—adaptive learning rates[35–37], trial-to-trial variability[38] and side biases[12] (Supplementary

Discussion 2.1)—while providing mechanistic insights into their task-specific implementation. Thus, our framework may accelerate scientific discovery by modelling task-specific behaviour, while uncovering generalizable principles transcending task-specific contexts.

The number of RNN units needed for optimal prediction estimates behavioural dimensionality[3,32]. In our study, tiny RNNs matched or outperformed larger RNNs, suggesting low-dimensional behaviour. While counting dynamical variables provides a principled metric, models vary in task-encoding efficiency—RNNs optimize dimensional efficiency through training, whereas cognitive models use hand-designed, theory-based variables. We emphasize the important distinction between the dimensionality of dynamics and model expressivity (for example, nonlinearity and parameter count), which determines input–output complexity. These two concepts offer complementary notions of model complexity in dynamical systems[39–42]. Identifying dynamical variables from experimental data is an active research field across neuroscience, complex systems and physics, with efforts to extract key variables from neural data, physical and multiscale complex systems[41,43–45].

Our approach involved key technical choices, each with specific strengths and limitations. First, we use GRUs for their Markovian property and selective information processing[31]. However, the GRU updating equation may limit the complexity of dynamics captured by tiny RNNs (see theoretical analysis in ref. 46 and Supplementary Fig. 39 for the case of one-dimensional discrete dynamics). Although they are sufficient for the tasks studied here, more complex behaviours may demand different architectures (for example, disentangled RNNs[47]). Additionally, our choices in training and regularization balanced flexibility and generalization. Despite broad effectiveness, our approach may face challenges in scaling to more complex tasks. Future work should explore alternative architectures, training schemes, or interpretability techniques to address these limitations.

Our findings have broad implications for cognitive and neural mechanisms, with applications in computational psychiatry and beyond. The accuracy of tiny RNNs in modelling individual behaviour makes them well-suited for studying individual differences in decision-making (for example, Supplementary Figs. 3, 10 and 15–17), a central theme in computational psychiatry. In future, this framework could be extended to more complex, naturalistic settings and domains such as perceptual decision-making and memory. Our approach may also yield predictions of neural activity, enabling more integrated models of cognition and bridging computational and neurobiological levels. In conclusion, tiny RNNs offer a powerful tool for revealing computational principles of adaptive behaviour, opening new research avenues in cognitive science, neuroscience and artificial intelligence.

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

## Methods

All data were analysed using Python 3.9 and PyTorch 1.13.

### Tasks and datasets

No statistical methods were used to predetermine sample sizes in this study. All datasets were drawn from previously published studies, and we included all available subjects (with enough trials for modelling) in each task. Allocation to experimental groups was not randomized by us; instead, randomization was previously performed by the original authors. Our study does not include any direct behavioural experimentation. Therefore, blinding was not required.

**Reversal learning task.** The reversal learning task is a paradigm designed to assess subjects' ability to adapt their behaviour in response to changing reward contingencies. In each trial, subjects are presented with two actions, $A_1$ and $A_2$, yielding a unit reward with probability $p_1^{\text{reward}}$ and $p_2^{\text{reward}}$, respectively. These reward probabilities remain constant for several trials before switching unpredictably and abruptly, without explicit cues. When this occurs, the action associated with the higher reward probability becomes linked to the lower reward probability, and vice versa. The task necessitates continuous exploration of which action currently has a higher reward probability in order to maximize total rewards. For consistency with the other animal tasks, we assume that actions ($A_1$ and $A_2$) are made at the choice state, and $A_i$ deterministically leads to state $S_i$, where the reward is delivered.

In the Bartolo dataset[10,48,49], 2 male monkeys (Rhesus macaque, *Macaca mulatta*; age 4.5 years) completed a total of 15,500 trials of the reversal learning task with 2 state-reward types: (1) $p_1^{\text{reward}} = 0.7$ and $p_2^{\text{reward}} = 0.3$; (2) $p_1^{\text{reward}} = 0.3$ and $p_2^{\text{reward}} = 0.7$. Blocks were 80 trials long, and the switch happened at a 'reversal trial' between trials 30 and 50. We predicted the behaviour from trials 10 to 70, similar to the original preprocessing procedure[10] because the monkeys were inferring the current block type ('what' block, choosing from two objects; 'where' block, choosing from two locations) in the first few trials.

In the Akam dataset[11,50], 10 male mice (C57BL6; aged between 2–3 months) completed a total of 67,009 trials of the reversal learning task with 3 state-reward types: (1) $p_1^{\text{reward}} = 0.75$ and $p_2^{\text{reward}} = 0.25$; (2) $p_1^{\text{reward}} = 0.25$ and $p_2^{\text{reward}} = 0.75$; (3) $p_1^{\text{reward}} = 0.5$ and $p_2^{\text{reward}} = 0.5$ (neutral trials). Block transitions from non-neutral blocks were triggered 10 trials after an exponential moving average (tau = 8 trials) crossed a 75% correct threshold. Block transitions from neutral blocks occurred with a probability of 10% on each trial after the 15th of the block to give an average neutral block length of 25 trials.

**Two-stage task.** The two-stage task is a paradigm commonly used to distinguish between the influences of model-free and model-based RL on animal behaviour[51], and later reduced in ref. 34. In each trial, subjects are presented with two actions, $A_1$ and $A_2$, while at the choice state. Action $A_1$ leads with a high probability to state $S_1$ and a low probability to state $S_2$, while action $A_2$ leads with a high probability to state $S_2$ and a low probability to state $S_1$. From second-stage states $S_1$ and $S_2$, the animal can execute an action for a chance of receiving a unit reward. Second-stage states are distinguishable by visual cues and have different probabilities of yielding a unit reward: $p_1^{\text{reward}}$ for $S_1$ and $p_2^{\text{reward}}$ for $S_2$. These reward probabilities remain constant for several trials before switching unpredictably and abruptly. When this occurs, the second-stage state associated with the higher reward probability becomes linked to the lower reward probability, and vice versa.

In the Miller dataset[12,52], 4 adult male Long-Evans rats (Taconic Biosciences; Hilltop Lab Animals) completed a total of 33,957 trials of the two-stage task with 2 state-reward types: (1) $p_1^{\text{reward}} = 0.8$ and $p_2^{\text{reward}} = 0.2$; (2) $p_1^{\text{reward}} = 0.2$ and $p_2^{\text{reward}} = 0.8$. Block switches occurred with a 2% probability on each trial after a minimum block length of 10 trials.

In the Akam dataset[11,50], 10 male mice (C57BL6; aged between 2–3 months) completed a total of 133,974 trials of the two-stage task with 3 state-reward types: (1) $p_1^{\text{reward}} = 0.8$ and $p_2^{\text{reward}} = 0.2$; (2) $p_1^{\text{reward}} = 0.2$ and $p_2^{\text{reward}} = 0.8$; (3) $p_1^{\text{reward}} = 0.4$ and $p_2^{\text{reward}} = 0.4$ (neutral trials). Block transitions occur 20 trials after an exponential moving average (tau = 8 trials) of the subject's choices crossed a 75% correct threshold. In neutral blocks, block transitions occurred with 10% probability on each trial after the 40th trial. Transitions from non-neutral blocks occurred with equal probability either to another non-neutral block or to the neutral block. Transitions from neutral blocks occurred with equal probability to one of the non-neutral blocks.

**Transition-reversal two-stage task.** The transition-reversal two-stage task is a modified version of the original two-stage task, with the introduction of occasional reversals in action-state-transition probabilities[11]. This modification was proposed to facilitate the dissociation of state prediction and reward prediction in neural activity and to prevent habit-like strategies that may produce model-based control-like behaviour without forward planning. In each trial, subjects are presented with two actions, $A_1$ and $A_2$, at the choice state. One action commonly leads to state $S_1$ and rarely to state $S_2$, while the other action commonly leads to state $S_2$ and rarely to state $S_1$. These action-state-transition probabilities remain constant for several trials before switching unpredictably and abruptly, without explicit cues. In the second-stage states $S_1$ and $S_2$, subjects execute an action for a chance of receiving a unit reward. The second-stage states are visually distinguishable and have different reward probabilities that also switch unpredictably and abruptly, without explicit cues, similar to the other two tasks.

In the Akam dataset[11,50], 17 male mice (C57BL6; aged between 2–3 months) completed a total of 230,237 trials of the transition-reversal two-stage task with 2 action-state types: (1) $\Pr(S_1|A_1) = \Pr(S_2|A_2) = 0.8$ and $\Pr(S_2|A_1) = \Pr(S_1|A_2) = 0.2$; (2) $\Pr(S_1|A_1) = \Pr(S_2|A_2) = 0.2$ and $\Pr(S_2|A_1) = \Pr(S_1|A_2) = 0.8$. There were also 3 state-reward types: (1) $p_1^{\text{reward}} = 0.8$ and $p_2^{\text{reward}} = 0.2$; (2) $p_1^{\text{reward}} = 0.2$ and $p_2^{\text{reward}} = 0.8$; (3) $p_1^{\text{reward}} = 0.4$ and $p_2^{\text{reward}} = 0.4$ (neutral trials). Block transitions occur 20 trials after an exponential moving average (tau = 8 trials) of the subject's choices crossed a 75% correct threshold. In neutral blocks, block transitions occurred with 10% probability on each trial after the 40th trial. Transitions from non-neutral blocks occurred with equal probability (25%) either to another non-neutral block via reversal in the reward or transition probabilities, or to one of the two neutral blocks. Transitions from neutral blocks occurred via a change in the reward probabilities only to one of the non-neutral blocks with the same transition probabilities.

**Three-armed reversal learning task.** In the Suthaharan dataset[53], 1,010 participants (605 participants from the pandemic group and 405 participants from the replication group) completed a three-armed probabilistic reversal learning task. This task was framed as either a non-social (card deck) or social (partner) domain, each lasting 160 trials divided evenly into 4 blocks. Participants were presented with 3 actions ($A_1$, $A_2$ and $A_3$; 3 decks of cards in the non-social domain frame or 3 avatar partners in the social domain frame), each containing different amounts of winning (+100) and losing (−50) points. The objective was to find the best option and earn as many points as possible, knowing that the best option could change.

The task contingencies started with 90%, 50% and 10% reward probabilities, with the best deck/partner switching after 9 out of 10 consecutive rewards. Unknown to the participants, the underlying contingencies transitioned to 80%, 40%, and 20% reward probabilities at the end of the second block, making it more challenging to distinguish between probabilistic noise and genuine changes in the best option.

**Four-armed drifting bandit task.** The Bahrami dataset[54] includes 975 participants who completed the 4-arm bandit task[55]. Participants were

asked to choose between 4 options on 150 trials. On each trial, they chose an option and were given a reward. The rewards for each option drifted over time in a manner known as a restless bandit, forcing the participants to constantly explore the different options to obtain the maximum reward. The rewards followed one of three predefined drift schedules[54].

During preprocessing, we removed 57 participants (5.9%) who missed more than 10% of trials. For model fitting, missing trials from other subjects are excluded from the loss calculation.

**Original two-stage task.** In the Gillan dataset[56,57], the original version of the two-stage task[51] was used to assess goal-directed (model-based) and habitual (model-free) learning in individuals with diverse psychiatric symptoms. In total, 1,961 participants (548 from the first experiment and 1413 from the second experiment) completed the task. In each trial, participants were presented with a choice between two options ($A_1$ or $A_2$). Each option commonly (70%) led to a particular second-stage state ($A_1{\to}S_1$ or $A_2{\to}S_2$). However, on 30% of 'rare' trials, choices led to the alternative second-stage state ($A_1{\to}S_2$ or $A_2{\to}S_1$). In the second-stage states, subjects chose between two options ($B_1/B_2$ in $S_1$ or $C_1/C_2$ in $S_2$), each associated with a distinct probability of being rewarded. The reward probabilities associated with each second-stage option drifted slowly and independently over time, remaining within the range of 0.25 to 0.75. To maximize rewards, subjects had to track which second-stage options were currently best as they changed over time.

For model fitting, missing stages or trials from some participants are excluded from the loss calculation.

## Recurrent neural networks
**Network architectures.** We investigated several architectures, as described below. Our primary goal is to capture the maximum possible behavioural variance with $d$ dynamical variables. While we generally prefer more flexible models due to their reduced bias, such models typically require more data for training, and insufficient data can result in underfitting and poorer performance in comparison to less flexible (simpler) models. Therefore, we aimed to balance data efficiency and model capacity through cross-validation.

After finding the best-performing model class, we performed an investigation of the network properties that contributed the most to the successfully explained variance. Analogous to ablation studies, our approach consisted of gradually removing components or adding constraints to the architectures, such as eliminating nonlinearity or introducing symmetric weight constraints. The unaffected predictive performance suggests that the examined components are not essential for the successfully explained variance. If affected, this indicates that these components can contribute to explaining additional behavioural patterns. Following this approach, we can establish connections between architectural components and their corresponding underlying behavioural patterns. The primary objective of this approach is to capture maximum variance with minimal components in the models, resulting in highly interpretable models.

**Recurrent layer.** The neural network models in this paper used the vanilla GRUs in their hidden layers[31]. The hidden state $h_t$ at the beginning of trial $t$ consists of $d$ elements (dynamical variables). The initial hidden state $h_1$ is set to 0 and $h_t$ ($t > 1$) is updated as follows:

$$
\begin{aligned}
r_t &= \sigma(W_{ir}x_{t-1} + b_{ir} + W_{hr}h_{t-1} + b_{hr}) \\
z_t &= \sigma(W_{iz}x_{t-1} + b_{iz} + W_{hz}h_{t-1} + b_{hz}) \\
n_t &= \tanh(W_{in}x_{t-1} + b_{in} + r_t \odot (W_{hn}h_{t-1} + b_{hn})) \\
h_t &= (1 - z_t) \odot n_t + z_t \odot h_{t-1}
\end{aligned}
\tag{1}
$$

where $\sigma$ is the sigmoid function, $\odot$ is the Hadamard (element-wise) product, $x_{t-1}$ and $h_{t-1}$ are the input and hidden state from the last trial

$t - 1$, and $r_t$, $z_t$ and $n_t$ are the reset, update and new gates (intermediate variables) at trial $t$, respectively. The weight matrices $W_{..}$ and biases $b_{..}$ are trainable parameters. The $d$-dimensional hidden state of the network, $h_t$, represents a summary of past inputs and is the only information used to generate outputs.

Importantly, the use of GRUs means that the set of $d$-unit activations fully specifies the network's internal state, rendering the system Markovian (that is, $h_t$ is fully determined by $h_{t-1}$ and $x_{t-1}$). This is in contrast to alternative RNN architectures such as the long short-term memory[58], where the use of a cell state renders the system non-Markovian (that is, the output state $h_t$ cannot be fully determined by $h_{t-1}$ and $x_{t-1}$).

To accommodate discrete inputs, we also introduce a modified architecture called switching GRU, where recurrent weights and biases are input-dependent, similar to discrete-latent-variable-dependent switching linear dynamical systems[59]. In this architecture, the hidden state $h_t$ ($t > 1$) is updated as follows:

$$
\begin{aligned}
r_t &= \sigma\left(b_{ir}^{(x_{t-1})} + W_{hr}^{(x_{t-1})}h_{t-1} + b_{hr}^{(x_{t-1})}\right) \\
z_t &= \sigma\left(b_{iz}^{(x_{t-1})} + W_{hz}^{(x_{t-1})}h_{t-1} + b_{hz}^{(x_{t-1})}\right) \\
n_t &= \tanh\left(b_{in}^{(x_{t-1})} + r_t \odot \left(W_{hn}^{(x_{t-1})}h_{t-1} + b_{hn}^{(x_{t-1})}\right)\right) \\
h_t &= (1 - z_t) \odot n_t + z_t \odot h_{t-1}
\end{aligned}
\tag{2}
$$

where $W_{h.}^{(x_{t-1})}$ and $b_{..}^{(x_{t-1})}$ are the weight matrices and biases selected by the input $x_{t-1}$ (that is, each input $x_{t-1}$ induces an independent set of weights $W_{h.}$ and biases $b_{..}$).

For discrete inputs, switching GRUs are a generalization of vanilla GRUs (that is, a vanilla GRU can be viewed as a switching GRU whose recurrent weights do not vary with the input). Generalizations of switching GRUs from discrete to continuous inputs are closely related to multiplicative integration GRUs[60].

For animal datasets, we found that the switching GRU models performed similarly to the vanilla GRU models for $d \geq 2$, but consistently outperformed the vanilla GRU models for $d = 1$. Therefore, for the results of animal datasets in the main text, we reported the performance of the switching GRU models for $d = 1$ and the performance of the vanilla GRU models for $d \geq 2$. Mathematically, these vanilla GRU models can be directly transformed into corresponding switching GRU models:

$$
\begin{aligned}
b_{i.}^{(x_{t-1})} &\leftarrow W_{i.}x_{t-1} + b_{i.} \\
b_{h.}^{(x_{t-1})} &\leftarrow b_{h.} \\
W_{h.}^{(x_{t-1})} &\leftarrow W_{h.}
\end{aligned}
\tag{3}
$$

We also proposed the switching linear neural networks (SLIN), where the hidden state $h_t$ ($t > 1$) is updated as follows:

$$
h_t = W^{(x_{t-1})}h_{t-1} + b^{(x_{t-1})}
\tag{4}
$$

where $W^{(x_{t-1})}$ and $b^{(x_{t-1})}$ are the weight matrices and biases selected by the input $x_{t-1}$. In some variants, we constrained $W^{(x_{t-1})}$ to be symmetric.
**Input layer.** The network's input $x_t$ consists of the previous action $a_{t-1}$, the previous second-stage state $s_{t-1}$, and the previous reward $r_{t-1}$ (but $a_t = s_t$ in the reversal learning task). In the vanilla GRU networks, the input $x_t$ is three-dimensional and projects with linear weights to the recurrent layer. In the switching GRU networks, the input $x_t$ is used as a selector variable where the network's recurrent weights and biases depend on the network's inputs. Thus, switching GRUs trained on the reversal learning task have four sets of recurrent weights and biases corresponding to all combinations of $a_{t-1}$ and $r_{t-1}$, and switching GRUs trained on the two-stage and transition-reversal two-stage tasks have eight sets of recurrent weights and biases corresponding to all combinations of $a_{t-1}$, $s_{t-1}$ and $r_{t-1}$.

**Output layer.** The network's output consists of two units whose activities are linear functions of the hidden state $h_t$. A softmax function (a generalization of the logistic function) is used to convert these activities into a probability distribution (a policy). In the first trial, the network's output is read out from the initial hidden state $h_1$, which has not yet been updated on the basis of any input. For $d$-unit networks, the network's output scores were computed either from a fully connected readout layer (that is, $s_t^{(i)} = \sum_{j=1}^{d} \beta_{i,j} \cdot h_t^{(j)}, i = 1, ..., d$) or from a diagonal readout layer (that is, $s_t^{(i)} = \beta_i \cdot h_t^{(i)}, i = 1, ..., d$). The output scores are sent to the softmax layer to produce action probabilities.

**Network training.** Networks were trained using the Adam optimizer (learning rate of 0.005) on batched training data with cross-entropy loss, recurrent weight L1-regularization loss (coefficient drawn between $10^{-5}$ and $10^{-1}$, depending on experiments), and early stop (if the validation loss does not improve for 200 iteration steps). All networks were implemented with PyTorch.

## Classical cognitive models
**Models for the reversal learning task.** In this task, we implemented one model from the Bayesian inference family and eight models from the model-free family (adopted from[34] and[12], or constructed from RNN phase portraits).

**Bayesian inference strategy ($d$ = 1).** This model (also known as latent-state) assumes the existence of the latent-state $h$, with $h = i$ representing a higher reward probability following action $A_i$ (state $S_i$). The probability $\text{Pr}_t(h = 1)$, as the dynamical variable, is first updated via Bayesian inference:

$$\widehat{\text{Pr}}_t(h = 1) = \frac{\text{Pr}(r_{t-1}|h = 1, s_{t-1})\text{Pr}_{t-1}(h = 1)}{\text{Pr}(r_{t-1}|h = 1, s_{t-1})\text{Pr}_{t-1}(h = 1) + \text{Pr}(r_{t-1}|h = 2, s_{t-1})\text{Pr}_{t-1}(h = 2)}, \quad (5)$$

where the left-hand side is the posterior probability (we omit the conditions for simplicity). The agent also incorporates the knowledge that, in each trial, the latent-state $h$ can switch (for example, from $h = 1$ to $h = 2$) with a small probability $p_r$. Thus the probability $\text{Pr}_t(h)$ reads,

$$\text{Pr}_t(h = 1) = (1 - p_r)\widehat{\text{Pr}}_t(h = 1) + p_r(1 - \widehat{\text{Pr}}_t(h = 1)). \quad (6)$$

The action probability is then derived from softmax ($\beta\text{Pr}_t(h = 1), \beta\text{Pr}_t(h = 2)$) with inverse temperature $\beta$ ($\beta \geq 0$).

**Model-free strategy ($d$ = 1).** This model hypothesizes that the two action values $Q_t(A_i)$ are fully anti-correlated ($Q_t(A_1) = -Q_t(A_2)$) as follows:

$$\begin{aligned} Q_t(a_{t-1}) &= Q_{t-1}(a_{t-1}) + \alpha(r_{t-1} - Q_{t-1}(a_{t-1})) \\ Q_t(\bar{a}_{t-1}) &= Q_{t-1}(\bar{a}_{t-1}) - \alpha(r_{t-1} + Q_{t-1}(\bar{a}_{t-1})), \end{aligned} \quad (7)$$

where $\bar{a}_{t-1}$ is the unchosen action, and $\alpha$ is the learning rate ($0 \leq \alpha \leq 1$). We specify the $Q_t(A_1)$ as the dynamical variable.

**Model-free strategy ($d$ = 2).** This model hypothesizes that the two action values $Q_t(A_i)$, as two dynamical variables, are updated independently:

$$Q_t(a_{t-1}) = Q_{t-1}(a_{t-1}) + \alpha(r_{t-1} - Q_{t-1}(a_{t-1})). \quad (8)$$

The unchosen action value $Q_t(\bar{a}_{t-1})$ is unaffected.

**Model-free strategy with value forgetting ($d$ = 2).** The chosen action value is updated as in the previous model. The unchosen action value $Q_t(\bar{a}_{t-1})$, instead, is gradually forgotten:

$$Q_t(\bar{a}_{t-1}) = DQ_{t-1}(\bar{a}_{t-1}), \quad (9)$$

where $D$ is the value forgetting rate ($0 \leq D \leq 1$).

**Model-free strategy with value forgetting to mean ($d$ = 2).** This model is the 'forgetful model-free strategy' proposed in[61]. The chosen action value is updated as in the previous model. The unchosen action value $Q_t(\bar{a}_{t-1})$, instead, is gradually forgotten to a initial value ($\widetilde{V} = 1/2$):

$$Q_t(\bar{a}_{t-1}) = DQ_{t-1}(\bar{a}_{t-1}) + (1 - D)\widetilde{V}, \quad (10)$$

where $D$ is the value forgetting rate ($0 \leq D \leq 1$).

**Model-free strategy with the drift-to-the-other rule ($d$ = 2).** This strategy is constructed from the phase diagram of the two-unit RNN. When there is a reward, the chosen action value is updated as follows,

$$Q_t(a_{t-1}) = D_1 Q_{t-1}(a_{t-1}) + 1, \quad (11)$$

where $D_1$ is the value drifting rate ($0 \leq D_1 \leq 1$). The unchosen action value is slightly decreased:

$$Q_t(\bar{a}_{t-1}) = Q_{t-1}(\bar{a}_{t-1}) - b, \quad (12)$$

where $b$ is the decaying bias ($0 \leq b \leq 1$, usually small). When there is no reward, the unchosen action value is unchanged, and the chosen action value drifts to the other:

$$Q_t(a_{t-1}) = Q_{t-1}(a_{t-1}) + \alpha_0(Q_{t-1}(\bar{a}_{t-1}) - Q_{t-1}(a_{t-1})), \quad (13)$$

where $\alpha_0$ is the drifting rate ($0 \leq \alpha_0 \leq 1$).

For all model-free RL models with $d$ = 2, the action probability is determined by softmax ($\beta Q_t(A_1), \beta Q_t(A_2)$).

**Model-free strategy with inertia ($d$ = 2).** The action values are updated as the model-free strategy ($d$ = 1). The action perseveration (inertia) is updated by:

$$\begin{aligned} X_t(a_{t-1}) &= X_{t-1}(a_{t-1}) + \alpha_{\text{pers}}(k_{\text{pers}} - X_{t-1}(a_{t-1})) \\ X_t(\bar{a}_{t-1}) &= X_{t-1}(\bar{a}_{t-1}) - \alpha_{\text{pers}}(k_{\text{pers}} + X_{t-1}(\bar{a}_{t-1})) \end{aligned} \quad (14)$$

where $\alpha_{\text{pers}}$ is the perseveration learning rate ($0 \leq \alpha_{\text{pers}} \leq 1$), and $k_{\text{pers}}$ is the single-trial perseveration term, affecting the balance between action values and action perseverations.

**Model-free strategy with inertia ($d$ = 3).** The action values are updated as the model-free strategy ($d$ = 2). The action perseveration (inertia) is updated by the same rule in the model-free strategy with inertia ($d$ = 2).

The action probabilities in all model-free models with inertia are generated via softmax ($\{\beta(Q_t(A_i) + X_t(A_i))\}_i$). Both the action values and action perseverations are dynamical variables.

**Model-free reward-as-cue strategy ($d$ = 8).** This model assumes that the animal considers the combination of the second-stage state $s_{t-1}$ and the reward $r_{t-1}$ from the trial $t - 1$ as the augmented state $\mathcal{S}_t$ for trial $t$. The eight dynamical variables are the values for the two actions at the four augmented states. The action values are updated as follows:

$$Q_t(\mathcal{S}_{t-1}, a_{t-1}) = Q_{t-1}(\mathcal{S}_{t-1}, a_{t-1}) + \alpha(r_{t-1} - Q_{t-1}(\mathcal{S}_{t-1}, a_{t-1})). \quad (15)$$

The action probability at trial $t$ is determined by softmax ($\beta Q_t(\mathcal{S}_t, A_1), \beta Q_t(\mathcal{S}_t, A_2)$).

**Models for the two-stage task.** We implemented one model from the Bayesian inference family, four models from the model-free family, and four from the model-based family (adopted from refs. 12,34).

**Bayesian inference strategy ($d$ = 1).** Same as Bayesian inference strategy ($d$ = 1) in the reversal learning task, except that $h = i$ represents a higher reward probability following state $S_i$ (not action $A_i$).

**Model-free strategy ($d$ = 1).** Same as the model-free strategy ($d$ = 1) in the reversal learning task by ignoring the second-stage states $s_{t-1}$.

**Model-free Q(1) strategy ($d$ = 2).** Same as the model-free strategy ($d$ = 2) in the reversal learning task by ignoring the second-stage states $s_{t-1}$.

**Model-free Q(0) strategy ($d = 4$).** This model first updates the first-stage action values $Q_t(a_{t-1})$ with the second-stage state values $V_{t-1}(s_{t-1})$:

$$Q_t(a_{t-1}) = Q_{t-1}(a_{t-1}) + \alpha(V_{t-1}(s_{t-1}) - Q_{t-1}(a_{t-1})), \quad (16)$$

while the unchosen action value $Q_t(\bar{a}_{t-1})$ is unaffected. Then the second-stage state value $V_t(s_{t-1})$ is updated by the observed reward:

$$V_t(s_{t-1}) = V_{t-1}(s_{t-1}) + \alpha(r_{t-1} - V_{t-1}(s_{t-1})). \quad (17)$$

The four dynamical variables are the two action values and two state values.

**Model-free reward-as-cue strategy ($d = 8$).** Same as model-free reward-as-cue strategy ($d = 8$) in the reversal learning task.

**Model-based strategy ($d = 1$).** In this model, the two state values $V_t(S_i)$ are fully anti-correlated ($V_t(S_1) = -V_t(S_2)$):

$$\begin{aligned} V_t(s_{t-1}) &= V_{t-1}(s_{t-1}) + \alpha(r_{t-1} - V_{t-1}(s_{t-1})) \\ V_t(\bar{s}_{t-1}) &= V_{t-1}(\bar{s}_{t-1}) - \alpha(r_{t-1} + V_{t-1}(\bar{s}_{t-1})), \end{aligned} \quad (18)$$

where $\bar{s}_{t-1}$ is the unvisited state. The dynamical variable is the state value $V_t(S_1)$.

**Model-based strategy ($d = 2$).** The visited state value is updated:

$$V_t(s_{t-1}) = V_{t-1}(s_{t-1}) + \alpha(r_{t-1} - V_{t-1}(s_{t-1})). \quad (19)$$

The unvisited state value is unchanged. The two dynamical variables are the two state values.

**Model-based strategy with value forgetting ($d = 2$).** The visited state value is updated as in the previous model. The unvisited state value is gradually forgotten:

$$V_t(\bar{s}_{t-1}) = DV_{t-1}(\bar{s}_{t-1}), \quad (20)$$

where $D$ is the value forgetting rate ($0 \le D \le 1$).

For all model-based RL models, the action values at the first stage are directly computed using the state-transition model:

$$Q_t^{mb}(A_i) = \sum_j Pr(S_j|A_i)V_t(S_j), \quad (21)$$

where $Pr(S_j|A_i)$ is known. The action probability is determined by softmax $(\beta Q_t^{mb}(A_1), \beta Q_t^{mb}(A_2))$.

**Model-based mixture strategy ($d = 2$).** This model is a mixture of the model-free strategy ($d = 1$) and the model-based strategy ($d = 1$). The net action values are determined by:

$$Q_t^{net}(A_i) = (1-w)Q_t^{mf}(A_i) + wQ_t^{mb}(A_i), \quad (22)$$

where $w$ controls the strength of the model-based component. The action probabilities are generated via softmax $(\beta Q_t^{net}(A_1), \beta Q_t^{net}(A_2))$. $Q_t^{mf}(A_1)$ and $V_t(S_1)$ are the dynamical variables.

**Models for the transition-reversal two-stage task.** For this task, we further include cognitive models proposed in ref. 11. We first describe different model components (ingredients) and corresponding numbers of dynamical variables, and then specify the components employed in each model.

**Second-stage state value component.** The visited state value is updated:

$$V_t(s_{t-1}) = V_{t-1}(s_{t-1}) + \alpha_Q(r_{t-1} - V_{t-1}(s_{t-1})). \quad (23)$$

The unvisited state value $V_t(\bar{s}_{t-1})$ is either unchanged or gradually forgotten with $f_Q$ as the value forgetting rate. This component requires two dynamical variables.

**Model-free action value component.** The first-stage action values $Q_t^{mf}(a_{t-1})$ are updated by the second-stage state values $V_{t-1}(s_{t-1})$ and the observed reward:

$$Q_t^{mf}(a_{t-1}) = Q_{t-1}^{mf}(a_{t-1}) + \alpha(\lambda r_{t-1} + (1-\lambda)V_{t-1}(s_{t-1}) - Q_{t-1}^{mf}(a_{t-1})), \quad (24)$$

where $\lambda$ is the eligibility trace. The unchosen action value $Q_t^{mf}(\bar{a}_{t-1})$ is unaffected or gradually forgotten with $f_Q$ as the value forgetting rate. This component requires two dynamical variables.

**Model-based component.** The action-state-transition probabilities are updated as:

$$\begin{aligned} P_t(s_{t-1}|a_{t-1}) &= P_{t-1}(s_{t-1}|a_{t-1}) + \alpha_T(1 - P_{t-1}(s_{t-1}|a_{t-1})) \\ P_t(\bar{s}_{t-1}|a_{t-1}) &= P_{t-1}(\bar{s}_{t-1}|a_{t-1}) + \alpha_T(0 - P_{t-1}(\bar{s}_{t-1}|a_{t-1})), \end{aligned} \quad (25)$$

where $\alpha_T$ is the transition probability learning rate. For the unchosen action, the action-state-transition probabilities are either unchanged or forgotten:

$$\begin{aligned} P_t(s_{t-1}|\bar{a}_{t-1}) &= P_{t-1}(s_{t-1}|\bar{a}_{t-1}) + f_T(0.5 - P_{t-1}(s_{t-1}|\bar{a}_{t-1})) \\ P_t(\bar{s}_{t-1}|\bar{a}_{t-1}) &= P_{t-1}(\bar{s}_{t-1}|\bar{a}_{t-1}) + f_T(0.5 - P_{t-1}(\bar{s}_{t-1}|\bar{a}_{t-1})), \end{aligned} \quad (26)$$

where $f_T$ is the transition probability forgetting rate.

The model-based action values at the first stage are directly computed using the learned state-transition model:

$$Q_t^{mb}(A_i) = \sum_j P_t(S_j|A_i)V_t(S_j). \quad (27)$$

This component requires two dynamical variables ($P_t(S_1|A_1)$ and $P_t(S_1|A_2)$), since other variables can be directly inferred.

**Motor-level model-free action component.** Due to the apparatus design in this task[11], it is proposed that the mice consider the motor-level actions $a_{t-1}^{mo}$, defined as the combination of the last-trial action $a_{t-1}$ and the second-stage state $s_{t-2}$ before it. The motor-level action values $Q_t^{mo}(a_{t-1}^{mo})$ are updated as:

$$Q_t^{mo}(a_{t-1}^{mo}) = Q_{t-1}^{mo}(a_{t-1}^{mo}) + \alpha(\lambda r_{t-1} + (1-\lambda)V_{t-1}(s_{t-2}) - Q_{t-1}^{mo}(a_{t-1}^{mo})), \quad (28)$$

where $\lambda$ is the eligibility trace. The unchosen motor-level action value $Q_t^{mo}$ is unaffected or gradually forgotten with $f_Q$ as the value forgetting rate. This component requires four dynamical variables (four motor-level actions).

**Choice perseveration component.** The single-trial perseveration $\bar{X}_{t-1}^{cp}$ is set to $-0.5$ for $a_{t-1} = A_1$ and $0.5$ for $a_{t-1} = A_2$. The multi-trial perseveration $Q_{t-1}^{cp}$ (exponential moving average of choices) is updated as:

$$X_t^{cp} = X_{t-1}^{cp} + \alpha_c(\bar{X}_{t-1}^{cp} - X_{t-1}^{cp}), \quad (29)$$

where $\alpha_c$ is the choice perseveration learning rate. In some models, the $\alpha_c$ is less than 1, so one dynamical variable is required; while in some other models, the $\alpha_c$ is fixed to 1, suggesting that it is reduced to the single-trial perseveration and no dynamical variable is required.

**Motor-level choice perseveration component.** The multi-trial motor-level perseveration $X_{t-1}^{mocp}(s_{t-2})$ is updated as:

$$X_t^{mocp}(s_{t-2}) = X_{t-1}^{mocp}(s_{t-2}) + \alpha_m(\bar{X}_{t-1}^{cp} - X_{t-1}^{mocp}(s_{t-2})), \quad (30)$$

where $\alpha_m$ is the motor-level choice perseveration learning rate. This component requires two dynamical variables.

**Action selection component.** The net action values are computed as follows:

$$Q_t^{net}(A_i) = G^{mf}Q_t^{mf}(A_i) + G^{mo}Q_t^{mo}(A_i, s_{t-1}) + G^{mb}Q_t^{mb}(A_i) + X_t(A_i), \quad (31)$$

where $G^{mf}$, $G^{mo}$ and $G^{mb}$ are model-free, motor-level model-free and model-based inverse temperatures, respectively, and $X_t(A_i)$ is:

$$X_t(A_1) = 0$$
$$X_t(A_2) = B_c + B_r \widetilde{X}_{t-1}^s + P_c X_t^{cp} + P_m X_t^{mocp}(s_{t-1}), \qquad (32)$$

where $B_c$ (bias), $B_r$ (rotation bias), $P_c$, $P_m$ are weights controlling each component, and $\widetilde{X}_{t-1}^s$ is $-0.5$ for $s_{t-1} = S_1$ and $0.5$ for $s_{t-1} = S_2$.

The action probabilities are generated via softmax ($Q_t^{net}(A_1)$, $Q_t^{net}(A_2)$).

**Model-free strategies.** We include five model-free RL models:
(1) the model-free strategy ($d = 1$) same as the two-stage task;
(2) the model-free Q(1) strategy ($d = 2$) same as the two-stage task;
(3) state value [2] + model-free action value [2] + bias [0] + rotation bias [0] + single-trial choice perseveration [0];
(4) state value [2] + model-free action value with forgetting [2] + bias [0] + rotation bias [0] + single-trial choice perseveration [0];
(5) state value [2] + model-free action value with forgetting [2] + motor-level model-free action value with forgetting [4] + bias [0] + rotation bias [0] + multi-trial choice perseveration [1] + multi-trial motor-level choice perseveration [2].

Here, we use the format of 'model component [required number of dynamical variables]' (more details in ref. 11).

**Model-based strategies.** We include 12 model-based RL models:
(1) state value [2] + model-based [2] + bias [0] + rotation bias [0] + single-trial choice perseveration [0];
(2) state value [2] + model-free action value [2] + model-based [2] + bias [0] + rotation bias [0] + single-trial choice perseveration [0];
(3) state value [2] + model-based with forgetting [2] + bias [0] + rotation bias [0] + single-trial choice perseveration [0];
(4) state value [2] + model-free action value with forgetting [2] + model-based with forgetting [2] + bias [0] + rotation bias [0] + single-trial choice perseveration [0];
(5) state value [2] + model-free action value with forgetting [2] + model-based [2] + bias [0] + rotation bias [0] + single-trial choice perseveration [0];
(6) state value [2] + model-free action value [2] + model-based [2] + bias [0] + rotation bias [0] + multi-trial choice perseveration [1];
(7) state value [2] + model-free action value with forgetting [2] + model-based with forgetting [2] + bias [0] + rotation bias [0] + multi-trial choice perseveration [1];
(8) state value [2] + model-free action value with forgetting [2] + model-based [2] + bias [0] + rotation bias [0] + multi-trial choice perseveration [1];
(9) state value [2] + model-free action value with forgetting [2] + model-based with forgetting [2] + bias [0] + rotation bias [0] + multi-trial motor-level choice perseveration [2];
(10) state value [2] + model-based with forgetting [2] + bias [0] + rotation bias [0] + multi-trial choice perseveration [1] + multi-trial motor-level choice perseveration [2];
(11) state value [2] + model-free action value with forgetting [2] + model-based with forgetting [2] + bias [0] + rotation bias [0] + multi-trial choice perseveration [1] + multi-trial motor-level choice perseveration [2];
(12) state value [2] + model-free action value with forgetting [2] + model-based with forgetting [2] + motor-level model-free action value with forgetting [4] + bias [0] + rotation bias [0] + multi-trial choice perseveration [1] + multi-trial motor-level choice perseveration [2].

Here, we use the format of model component [required number of dynamical variables] (more details in ref. 11).

**Models for the three-armed reversal learning task.** We implemented four models ($n = 3$ actions) from the model-free family, one of which is constructed from the strategies discovered by the RNN.

**Model-free strategy ($d = n$).** This model hypothesizes that each action value $Q_t(A_i)$, as a dynamical variable, is updated independently. The chosen action value is updated by:

$$Q_t(a_{t-1}) = Q_{t-1}(a_{t-1}) + \alpha(r_{t-1} - Q_{t-1}(a_{t-1})). \qquad (33)$$

The unchosen action values $Q_t(A_j)$ ($A_j \neq a_{t-1}$) are unaffected.

**Model-free strategy with value forgetting ($d = n$).** The chosen action value is updated as in the previous model. The unchosen action value $Q_t(A_j)$ ($A_j \neq a_{t-1}$), instead, is gradually forgotten:

$$Q_t(A_j) = D Q_{t-1}(A_j), \qquad (34)$$

where $D$ is the value forgetting rate ($0 \leq D \leq 1$).

**Model-free strategy with value forgetting and action perseveration ($d = 2n$).** The action values are updated as the model-free strategy with value forgetting. The chosen action perseveration is updated by:

$$X_t(a_{t-1}) = D_{pers} X_{t-1}(a_{t-1}) + k_{pers}, \qquad (35)$$

and the unchosen action perseverations are updated by:

$$X_t(A_j) = D_{pers} X_{t-1}(A_j), \qquad (36)$$

where $D_{pers}$ is the perseveration forgetting rate ($0 \leq D_{pers} \leq 1$), and $k_{pers}$ is the single-trial perseveration term, affecting the balance between action values and action perseverations.

**Model-free strategy with unchosen value updating and reward utility ($d = n$).** This model is constructed from the strategy discovered by the RNN (see Supplementary Results 1.4). It assumes that the reward utility $U(r)$ (equivalent to the preference setpoint) is different in four cases (corresponding to four free parameters): no reward for chosen action ($U_c(0)$), one reward for chosen action ($U_c(1)$), no reward for unchosen action ($U_u(0)$), and one reward for chosen action ($U_u(1)$).

The chosen action value is updated by:

$$Q_t(a_{t-1}) = Q_{t-1}(a_{t-1}) + \alpha_c(U_c(r_{t-1}) - Q_{t-1}(a_{t-1})). \qquad (37)$$

The unchosen action value $Q_t(A_j)$ ($A_j \neq a_{t-1}$) is updated by:

$$Q_t(A_j) = Q_{t-1}(A_j) + \alpha_u(U_u(r_{t-1}) - Q_{t-1}(A_j)). \qquad (38)$$

The action probabilities for these models are generated via softmax ($\{\beta(Q_t(A_i) + X_t(A_i))\}_i$) ($X_t = 0$ for models without action perseverations). Both the action values and action perseverations are dynamical variables.

**Models for the four-armed drifting bandit task.** We implemented five models ($n = 4$ actions) from the model-free family, two of which are constructed from the strategies discovered by the RNN.

**Model-free strategy ($d = n$).** This model is the same as the model-free strategy in the three-armed reversal learning task.

**Model-free strategy with value forgetting ($d = n$).** This model is the same as the model-free strategy with value forgetting in the three-armed reversal learning task.

**Model-free strategy with value forgetting and action perseveration ($d = 2n$).** This model is the same as the model-free strategy with value forgetting and action perseveration in the three-armed reversal learning task.

**Model-free strategy with unchosen value updating and reward reference point ($d = n$).** This model is constructed from the strategy discovered by the RNN (see Supplementary Results 1.5). It assumes that the reward utility $U(r)$ is different for chosen action ($U_c(r) = \beta_c(r - R_c)$) and for unchosen action ($U_u(r) = \beta_u(r - R_u)$), where $\beta_c$ and $\beta_u$ are reward sensitivities, and $R_c$ and $R_u$ are reward reference points.

The chosen action value is updated by:

$$Q_t(a_{t-1}) = (1 - \alpha_c)Q_{t-1}(a_{t-1}) + U_c(r_{t-1}), \tag{39}$$

where $1 - \alpha_c$ is the decay rate for chosen actions. The unchosen action value $Q_t(A_j)$ ($A_j \neq a_{t-1}$) is updated by:

$$Q_t(A_j) = (1 - \alpha_u)Q_{t-1}(A_j) + U_u(r_{t-1}), \tag{40}$$

where $1 - \alpha_u$ is the decay rate for unchosen actions. We additionally fit a reduced model of this strategy where $\beta_c = \alpha_c$ and $\beta_u = \alpha_u$ (similarly inspired by the RNN's solution).

The action probabilities for these models are generated via softmax $(\{\beta(Q_t(A_i) + X_t(A_i))\}_i)$ ($X_t = 0$ for models without action perseverations). Both the action values and action perseverations are dynamical variables.

**Models for the original two-stage task. Model-free strategy ($d = 3$).** This model hypothesizes that the action values for each task state (first-stage state $S_0$, second-stage states $S_1$ and $S_2$) are fully anti-correlated $(Q_t^{S_0}(A_1) = -Q_t^{S_0}(A_2), Q_t^{S_1}(B_1) = -Q_t^{S_1}(B_2), Q_t^{S_2}(B_3) = -Q_t^{S_2}(B_3))$.

The action values at the chosen second-stage state (for example, assuming $B_1$ or $B_2$ at $S_1$ is chosen) are updated by:

$$Q_t^{S_1}(a_{t-1}^{S_1}) = Q_{t-1}^{S_1}(a_{t-1}^{S_1}) + \alpha_2\left(r_{t-1} - Q_{t-1}^{S_1}(a_{t-1}^{S_1})\right)$$
$$Q_t^{S_1}(\overline{a}_{t-1}^{S_1}) = Q_{t-1}^{S_1}(\overline{a}_{t-1}^{S_1}) - \alpha_2\left(r_{t-1} + Q_{t-1}^{S_1}(\overline{a}_{t-1}^{S_1})\right), \tag{41}$$

where $\overline{a}_{t-1}^{S_1}$ is the unchosen second-stage action at the chosen second-stage state, and $\alpha_2$ is the learning rate for the second-stage states ($0 \leq \alpha_2 \leq 1$). The second-stage action probabilities are generated via softmax $(\beta_2 Q_t^{S_1}(B_1), \beta_2 Q_t^{S_1}(B_2))$.

The action values at the first-stage state ($A_1$ or $A_2$ at $S_0$) are updated by:

$$Q_t^{S_0,\mathrm{mf}}(a_{t-1}^{S_0}) = Q_{t-1}^{S_0,\mathrm{mf}}(a_{t-1}^{S_0}) + \alpha_1(\lambda r_{t-1} + (1-\lambda)Q_t^{S_1}(a_{t-1}^{S_1})$$
$$- Q_{t-1}^{S_0,\mathrm{mf}}(a_{t-1}^{S_0}))$$
$$Q_t^{S_0,\mathrm{mf}}(\overline{a}_{t-1}^{S_0}) = Q_{t-1}^{S_0,\mathrm{mf}}(\overline{a}_{t-1}^{S_0}) - \alpha_1(\lambda r_{t-1} + (1-\lambda)Q_t^{S_1}(a_{t-1}^{S_1})$$
$$+ Q_{t-1}^{S_0,\mathrm{mf}}(\overline{a}_{t-1}^{S_0})), \tag{42}$$

where $\overline{a}_{t-1}^{S_0}$ is the unchosen first-stage action, $\alpha_1$ is the learning rate for the first-stage state ($0 \leq \alpha_1 \leq 1$), and $\lambda$ specifies the TD($\lambda$) learning rule. The first-stage action probabilities are generated via softmax $(\beta_1 Q_t^{S_0,\mathrm{mf}}(A_1), \beta_1 Q_t^{S_0,\mathrm{mf}}(A_2))$.

Here $Q_t^{S_0,\mathrm{mf}}(A_1), Q_t^{S_1}(B_1)$, and $Q_t^{S_2}(C_1)$ are the dynamical variables.

**Model-based strategy ($d = 2$).** The update of action values at the chosen second-stage state is the same as the model-free strategy. The action values at the first-stage state ($A_1$ or $A_2$ at $S_0$) are determined by:

$$Q_t^{S_0,\mathrm{mb}}(A_i) = \Pr[S_1|A_i]\max_{B_j} Q_t^{S_1}(B_j) + \Pr[S_2|A_i]\max_{C_j} Q_t^{S_2}(C_j). \tag{43}$$

The first-stage action probabilities are generated via softmax $(\beta_1 Q_t^{S_0,\mathrm{mb}}(A_1), \beta_1 Q_t^{S_0,\mathrm{mb}}(A_2))$.

Only $Q_t^{S_1}(B_1)$ and $Q_t^{S_2}(C_1)$ are the dynamical variables.

**Model-based mixture strategy ($d = 3$).** This model considers the mixture of model-free and model-based strategies for the first-stage states. The net action values are determined by:

$$Q_t^{S_0,\mathrm{net}}(A_i) = (1-w)Q_t^{S_0,\mathrm{mf}}(A_i) + wQ_t^{S_0,\mathrm{mb}}(A_i), \tag{44}$$

where $w$ controls the strength of the model-based component. The first-stage action probabilities are generated via softmax $(\beta_1 Q_t^{S_0,\mathrm{net}}(A_1), \beta_1 Q_t^{S_0,\mathrm{net}}(A_2))$. $Q_t^{S_0,\mathrm{mf}}(A_1), Q_t^{S_1}(B_1)$ and $Q_t^{S_2}(C_1)$ are the dynamical variables.

**Model-free strategy ($d = 6$).** Compared to the model-free strategy ($d = 3$), only the chosen action values at $S_0$, $S_1$, and $S_2$ are updated. The unchosen values are unchanged. $Q_t^{S_0,\mathrm{mf}}(A_1), Q_t^{S_0,\mathrm{mf}}(A_2), Q_t^{S_1}(B_1), Q_t^{S_1}(B_2)$, $Q_t^{S_2}(C_1)$ and $Q_t^{S_2}(C_2)$ are the dynamical variables.

**Model-based strategy ($d = 4$).** Compared to the model-based strategy ($d = 2$), only the chosen action values at $S_1$, and $S_2$ are updated. The unchosen values are unchanged. $Q_t^{S_1}(B_1), Q_t^{S_1}(B_2), Q_t^{S_2}(C_1)$ and $Q_t^{S_2}(C_2)$ are the dynamical variables.

**Model-based mixture strategy ($d = 6$).** Compared to the model-based mixture strategy ($d = 3$), only the chosen action values at $S_0$, $S_1$ and $S_2$ are updated. The unchosen values are unchanged. $Q_t^{S_0,\mathrm{mf}}(A_1)$, $Q_t^{S_0,\mathrm{mf}}(A_2)$, $Q_t^{S_1}(B_1)$, $Q_t^{S_1}(B_2)$, $Q_t^{S_2}(C_1)$ and $Q_t^{S_2}(C_2)$ are the dynamical variables.

**Model-free strategy with reward utility ($d = 3$).** This model is constructed from the RNN's strategy. Similar to the model-free strategy ($d = 3$), it hypothesizes that the action values for each task state (first-stage state $S_0$, second-stage states $S_1$ and $S_2$) are fully anti-correlated $(Q_t^{S_0}(A_1) = -Q_t^{S_0}(A_2), Q_t^{S_1}(B_1) = -Q_t^{S_1}(B_2), Q_t^{S_2}(B_3) = -Q_t^{S_2}(B_3))$.

It assumes that when receiving one reward, the reward utility (that is, equivalently, the preference setpoint) for the chosen action at the first-stage state $S_0$ is $U^{S_0}(1) = 1$, for the chosen action at the chosen second-stage state $S_1$ (or $S_2$) is $U^{S_1}(1) = 1$, and for the (motor-level) chosen action at the unchosen second-stage state $S_2$ (or $S_1$) is $U^{S_2}(1) = U_{\mathrm{other}}$ (for example, $B_1$ at the chosen $S_1$ and $C_1$ at unchosen $S_2$ are the same motor-level action). When receiving no reward, the reward utility for the chosen action at the first-stage state $S_0$ is $U^{S_0}(0) = U_{\mathrm{1st,zero}}$, for the chosen action at the chosen second-stage state (assuming $S_1$) is $U^{S_1}(0) = U_{\mathrm{2nd,zero}}$, and for the (motor-level) chosen action at the unchosen second-stage state (assuming $S_2$) is $U^{S_2}(0) = -U_{\mathrm{other}}$. The chosen action values at the chosen second-stage state (for example, assuming $B_1$ or $B_2$ at $S_1$) are updated by:

$$Q_t^{S_1}(a_{t-1}^{S_1}) = Q_{t-1}^{S_1}(a_{t-1}^{S_1}) + \alpha_2(U^{S_1}(r_{t-1}) - Q_{t-1}^{S_1}(a_{t-1}^{S_1})), \tag{45}$$

where $\alpha_2$ is the learning rate for the second-stage states ($0 \leq \alpha_2 \leq 1$). The (motor-level) chosen action values (that is, $\tilde{a}_{t-1}^{S_2} = C_1$ if $a_{t-1}^{S_1} = B_1$ and, $\tilde{a}_{t-1}^{S_2} = C_2$ if $a_{t-1}^{S_1} = B_2$) at the unchosen second-stage state (for example, assuming $C_1$ or $C_2$ at $S_2$) are updated by:

$$Q_t^{S_2}(\tilde{a}_{t-1}^{S_2}) = Q_{t-1}^{S_2}(\tilde{a}_{t-1}^{S_2}) + \alpha_2(U^{S_2}(r_{t-1}) - Q_{t-1}^{S_2}(\tilde{a}_{t-1}^{S_2})). \tag{46}$$

The second-stage action probabilities are generated via softmax $(\beta_2 Q_t^{S_1}(B_1), \beta_2 Q_t^{S_1}(B_2))$.

The action values at the first-stage state ($A_1$ or $A_2$ at $S_0$) are updated by:

$$Q_t^{S_0}(a_{t-1}^{S_0}) = Q_{t-1}^{S_0}(a_{t-1}^{S_0}) + \alpha_1(U^{S_0}(r_{t-1}) - Q_{t-1}^{S_0}(a_{t-1}^{S_0})) \tag{47}$$

where $\alpha_1$ is the learning rate for the first-stage state ($0 \leq \alpha_1 \leq 1$). The first-stage action probabilities are generated via softmax $(\beta_1 Q_t^{S_0}(A_1), \beta_1 Q_t^{S_0}(A_2))$.

Here $Q_t^{S_0}(A_1), Q_t^{S_1}(B_1)$, and $Q_t^{S_2}(C_1)$ are the dynamical variables.

**Model fitting**

**Maximum likelihood estimation.** The parameters in all models were optimized on the training dataset to maximize the log-likelihood (that is, minimize the negative log-likelihood, or cross-entropy) for the next-action prediction. The loss function is defined as follows:

$$\mathcal{L} = -\log\Pr[\text{action sequences from one subject given one model}]$$
$$= -\sum_{n=1}^{N_{\mathrm{session}}}\sum_{t=1}^{T_n}\log\Pr[\text{observing } a_t \text{ given past observations and the model}], \tag{48}$$

where $N_{\text{session}}$ is the number of sessions and $T_n$ is the number of trials in session $n$.

**Nested cross-validation.** To avoid overfitting and ensure a fair comparison between models with varying numbers of parameters, we implemented nested cross-validation. For each animal, we first divided sessions into non-overlapping shorter blocks (approximately 150 trials per block) and allocated these blocks into ten folds. In the outer loop, nine folds were designated for training and validation, while the remaining fold was reserved for testing. In the inner loop, eight of the nine folds were assigned for training (optimizing a model's parameters for a given set of hyperparameters), and the remaining fold of the nine was allocated for validation (selecting the best-performing model across all hyperparameter sets). Notice that this procedure allows different hyperparameters for each test set.

RNNs' hyperparameters encompassed the L1-regularization coefficient on recurrent weights (drawn from $10^{-5}$, $10^{-4}$, $10^{-3}$, $10^{-2}$ or $10^{-1}$, depending on the experiments), the number of training epochs (that is, early stopping), and the random seed (three seeds). For cognitive models, the only hyperparameter was the random seed (used for parameter initialization). The inner loop produced nine models, with the best-performing model, based on average performance in the training and validation datasets, being selected and evaluated on the unseen testing fold. The final testing performance was computed as the average across all ten testing folds, weighted by the number of trials per block. This approach ensures that test data is exclusively used for evaluation and is never encountered during training or selection.

During RNN training, we employed early stopping if the validation performance failed to improve after 200 training epochs. This method effectively prevents RNN overfitting on the training data. According to this criterion, a more flexible model may demonstrate worse performance than a less flexible one, as the training for the former could be halted early due to insufficient training data. However, it is expected that the more flexible model would continue to improve with additional training data (for example, see Supplementary Fig. 8).

We note that, in the rich-data situation, this training–validation–test split in (nested) cross-validation is better than the typical usage of AIC[62], corrected AIC (AICc)[63] or BIC[64] in cognitive modelling, due to the following reasons[65]: the (nested) cross-validation provides a direct and unbiased estimate of the expected extra-sample test error, which reflects the generalization performance on new data points with inputs not necessarily appearing in the training dataset; by contrast, AIC, AICc and BIC can only provide asymptotically unbiased estimates of in-sample test error under some conditions (for example, models are linear in their parameters), measuring the generalization performance on new data points with inputs always appearing in the training dataset (the labels could be different from those in the training dataset due to noise). Furthermore, in contrast to regular statistical models, neural networks are singular statistical models with degenerate Fisher information matrices. Consequently, estimating the model complexity (the number of effective parameters, as used in AIC, AICc or BIC) in neural networks requires estimating the real log canonical threshold[66], which falls outside the scope of this study.

## Estimating the dimensionality of behaviour

For each animal, we observed that the predictive performance of RNN models initially improves and then saturates, or sometimes declines as the number $d$ of dynamical variables increases. To operationally estimate the dimensionality $d_*$ of behaviour, we implemented a statistical procedure that satisfies two criteria: (1) the RNN model with $d_*$ dynamical variables significantly outperforms all RNN models with $d < d_*$ dynamical variables (using a significance level of 0.05 in the $t$-tests of predictive performance conducted over outer folds); (2) any RNN model with $d'$ ($d' > d_*$) dynamical variables does not exhibit significant improvement over all RNN models with $d < d'$ dynamical variables.

Our primary objective is to estimate the intrinsic dimensionality (reflecting the latent variables in the data-generating process), not the embedding dimensionality[67]. However, it is important to consider the practical limitations associated with the estimation procedure. For instance, RNN models may fail to uncover certain latent variables due to factors such as limited training data or variables operating over very long time scales, leading to an underestimation of $d_*$. Additionally, even if all $d_*$ latent variables are accurately captured, the RNN models may still require $d \geq d_*$ dynamical variables to effectively and losslessly embed $d_*$-dimensional dynamics, particularly if they exhibit high nonlinearity, potentially resulting in an overestimation of $d_*$. A comprehensive understanding of these factors is crucial for future studies.

## Knowledge distillation

We employ the knowledge distillation framework[33] to fit models to individual subjects, while simultaneously leveraging group data: first fitting a teacher network to data from multiple subjects, and then fitting a student network to the outputs of the teacher network corresponding to an individual subject.

**Teacher network.** In the teacher network (TN), each subject is represented by a one-hot vector. This vector projects through a fully connected linear layer into a subject-embedding vector $e_{\text{sub}}$, which is provided as an additional input to the RNN. The teacher network uses 20 units in its hidden layer and uses the same output layer and loss (cross-entropy between the next-trial action and the predicted next-trial action probability) as in previous RNN models.

**Student network.** The student network (SN) has the same architecture as previous tiny RNNs. The only difference is that, during training and validation, the loss is defined as cross-entropy between the next-trial action probability provided by the teacher and the next-trial action probability predicted by the student:

$$\mathcal{L} = -\sum_{n=1}^{N_{\text{session}}} \sum_{t=1}^{T_n} \sum_{a=1}^{N_a} \text{Pr}^{\text{TN}}[a_t = a | \text{past observations}] \tag{49}$$
$$\times \log \text{Pr}^{\text{SN}}[a_t = a | \text{past observations}],$$

where $N_{\text{session}}$ is the number of sessions, $T_n$ is the number of trials in session $n$, and $N_a$ is the number of actions.

**Training, validation and test data in knowledge distillation for the mouse in the Akam dataset.** To study the influence of the number of training trials from one representative mouse on the performance of knowledge distillation, we employed a procedure different from nested cross-validation. This procedure splits the data from animal $M$ into two sets. The first set consisted of 25% of the trials and was used as a hold-out $M$-test dataset. The second set consisted of the remaining 75% trials, from which smaller datasets of different sizes were sampled. From each sampled dataset, 90% of the trials were used for training ($M$-training dataset) and 10% for validation ($M$-validation dataset). Next, we split the data from all other animals, with 90% of the data used for training ($O$-training dataset) and 10% for validation ($O$-validation dataset).

After dividing the datasets as described above, we trained the models. The solo RNNs were trained to predict choices on the $M$-training dataset and selected on the $M$-validation dataset. The teacher RNNs were trained to predict choices on the $M$- and $O$-training datasets and selected on the $M$- and $O$-validation datasets. The number of embedding units in the teacher RNNs was selected based on the $M$-validation dataset. The student RNNs were trained on the $M$-training dataset and selected on the $M$-validation dataset, but with the training target of action probabilities provided by the teacher RNNs. Here the student RNNs and the corresponding teacher RNNs were trained on the same

*M*-training dataset. Finally, all models were evaluated on the unseen *M*-test data.

When training the student RNNs, due to symmetry in the task, we augment the *M*-training datasets by flipping the action and second-stage states, resulting in an augmented dataset that is four times the size of the original one, similar to[29]. One key difference between our augmentation procedure and that of[29] is that the authors augmented the data for training the group RNNs, where the potential action bias presented in the original dataset (and other related biases) becomes invisible to the RNNs. By contrast, our teacher RNNs are trained only on the original dataset, where any potential action biases can be learned. Even if we augment the training data later for the student networks, the biases learned by the teacher network can still be transferred into the student networks. In addition to direct augmentation, simulating the teacher network can be another method to generate pseudo-data. The benefit of these pseudo-data was discussed in model compression[68].

## Protocols for training, validating and testing models in human datasets

**Interspersed split protocol.** In the three human datasets, each subject only performs one block of 100–200 trials. In the standard practice of cognitive modelling, the cognitive models are trained and tested on the same block, leading to potential overfitting and exaggerated performance. While it is possible to directly segment one block into three sequences for training, validation, and testing, this might introduce undesired distributional shifts in the sequences due to the learning effect. To ensure a fair comparison between RNNs and cognitive models, here we propose a new interspersed split protocol to define the training, validation and testing trials, similar to the usage of goldfish loss to prevent the memorization of training data in language models[69]. Specifically, we randomly sample without replacement ~75% trial indexes for training, ~12.5% trial indexes for validation and ~12.5% trial indexes for testing (three-armed reversal learning task: 120/20/20 (training/validation/testing); four-armed drifting bandit task: 110/20/20; original two-stage task: 150/25/25). We then feed in the whole block of trials as the model's inputs, obtain the output probabilities for each trial, and calculate the training, validation, and testing losses for each set of trial indexes, separately. This protocol guarantees the identical distribution between three sets of trials.

One possible concern is whether the test data is leaked into the training data in this protocol. For instance, the models are trained on the input sequence $((a_1, r_1), (a_2, r_2), (a_3, r_3))$ to predict $a_4$ and later tested on the input sequence $((a_1, r_1), (a_2, r_2))$ to predict $a_3$. In this scenario, while the models see $a_3$ in the input during training, they never see $a_3$ in the *output*. Thus, models are not trained to learn the input–output mapping from $((a_1, r_1), (a_2, r_2))$ to $a_3$, which is evaluated during testing. We confirmed that this procedure prevents data leakage on artificially generated choices (Supplementary Fig. 40).

**Cross-subject split protocol.** In addition to the interspersed split protocol, it is possible to train the RNNs on a proportion of subjects and evaluate them on held-out subjects (that is, zero-shot generalization), a cross-subject split protocol. To illustrate this protocol, we first divided all subjects into six folds of cross-validation. The teacher network was trained and validated using five folds and tested on the remaining one fold. For each subject in the test fold, because each subject only completed one task block, student networks are trained on the action-augmented blocks (to predict the teacher's choice probabilities for the subject), validated on the original block (to predict the teacher's choice probabilities for the subject), and tested on the original block (to predict actual choices of the subject). By design, both teacher networks and student networks will not overfit the subjects' choices in the test data. The cognitive models were trained and validated using five folds and tested on the remaining one fold. We presented the results in Supplementary Fig. 41.

## Phase portraits

**Models with $d=1$. Logit.** In each trial $t$, a model predicts the action probabilities $\Pr(a_t = A_1)$ and $\Pr(a_t = A_2)$. We define the logit $L(t)$ (log odds) at trial $t$ as $L(t) = \log(\Pr(a_t = A_1)/\Pr(a_t = A_2))$. When applied to probabilities computed via softmax, the logit yields $L(t) = \log(e^{\beta o_t^{(1)}}/e^{\beta o_t^{(2)}}) = \beta(o_t^{(1)} - o_t^{(2)})$, where $o_t^{(i)}$ is the model's output for action $a_t = A_i$ before softmax. Thus, the logit can be viewed as reflecting the preference for action $A_1$ over $A_2$: in RNNs, the logit corresponds to the score difference $o_t^{(1)} - o_t^{(2)}$; in model-free and model-based RL models, the logit is proportional to the difference in first-stage action values $Q_t(A_1) - Q_t(A_2)$; in Bayesian inference models, the logit is proportional to the difference in latent-state probabilities $\Pr_t(h = 1) - \Pr_t(h = 2) = 2\Pr_t(h = 1) - 1$.

**Logit change.** We define the logit change, $\Delta L(t)$, in trial $t$ as the difference between $L(t + 1)$ and $L(t)$. In one-dimensional models, $\Delta L(t)$ is a function of the input and $L(t)$, forming a vector field.

**Stability of fixed points.** Here we derive the stability of a fixed point in one-dimensional discrete dynamical systems. The system's dynamics update according to:

$$L_{\text{next}} = f_I(L), \tag{50}$$

where $L$ is the current-trial logit, $L_{\text{next}}$ is the next-trial logit, and $f_I$ is a function determined by input $I$ (omitted for simplicity). At a fixed point, denoted by $L = L^*$, we have

$$L^* = f(L^*). \tag{51}$$

Next, we consider a small perturbation $\delta L$ around the fixed point:

$$\begin{aligned} L_{\text{next}} &= f(L^* + \delta L) \\ &\approx f(L^*) + f'(L^*)\delta L \\ &= L^* + f'(L^*)\delta L. \end{aligned} \tag{52}$$

The fixed point is stable only when $-1 < f'(L^*) < 1$. Because the logit change $\Delta L$ is defined as $\Delta L = g(L) = f(L) - L$, we have the stability condition $-2 < g'(L^*) < 0$.

**Effective learning rate and slope.** In the one-dimensional RL models with prediction error updates and constant learning rate $\alpha$, we have

$$g(L) = \alpha(L^* - L), \tag{53}$$

where $g(L)$ is the logit change at $L$. In general, to obtain a generalized form of $g(L) = \alpha(L)(L^* - L)$ with a non-constant learning rate, we define the effective learning rate $\alpha(L)$ at $L$ relative to a stable fixed point $L^*$ as:

$$\alpha(L) = -\frac{g(L) - g(L^*)}{L - L^*} = -\frac{g(L)}{L - L^*}. \tag{54}$$

At $L^*$, $\alpha(L^*)$ is the negative slope $-g'(L^*)$ of the tangent at $L^*$. However, for general $L \neq L^*$, $\alpha(L)$ is the negative slope of the secant connecting $(L, g(L))$ and $(L^*, 0)$, which is different from $-g'(L)$.

We have

$$\begin{aligned} \alpha'(L)\delta L &\approx \alpha(L + \delta L) - \alpha(L) \\ &= \frac{g(L)}{L - L^*} - \frac{g(L + \delta L)}{L + \delta L - L^*} \\ &\approx \frac{-\alpha(L) - g'(L)}{L - L^*}\delta L. \end{aligned} \tag{55}$$

Letting $\delta L$ go to zero, we have:

$$\alpha(L) = -g'(L) - \alpha'(L)(L - L^*), \tag{56}$$

which provides the relationship between the effective learning rate $\alpha(L)$ and the slope of the tangent $g'(L)$.

**Models with $d > 1$.** In models with more dynamical variables, $\Delta L(t)$ is no longer solely a function of the input and $L(t)$ due to added degrees of freedom. In these models, the state space is spanned by a set of dynamical variables, collected by the vector $F(t)$. For example, the action value vector is the $F(t) = (Q_t(A_1), Q_t(A_2))^T$ in the two-dimensional RL models. The vector field $\Delta F(t)$ can be defined as $\Delta F(t) = F(t+1) - F(t) = (Q_{t+1}(A_1) - Q_t(A_1), Q_{t+1}(A_2) - Q_t(A_2))^T$, a function of $F(t)$ and the input in trial $t$.

## Dynamical regression

For one-dimensional models with states characterized by the policy logit $L(t)$, we can approximate the one-step dynamics for a given input with a linear function—that is, $\Delta L \sim \beta_0 + \beta_L L$. The coefficients $\beta_0$ and $\beta_L$ can be computed via linear regression, or 'dynamical regression' given its use in modelling dynamical systems. Here, $\beta_0$ is similar to the preference setpoint and $\beta_L$ is similar to learning rates in RL models.

For models with more than one dynamical variable, we can use a similar dynamical regression approach to extract a first-order approximation of the model dynamics via linearizations of vector fields. To facilitate interpretation, we consider only $d$-dimensional RNNs with a $d$-unit diagonal readout layer (denoted by $L_i(t)$ or $P_i(t)$; a non-degenerate case).

For tasks with a single choice state (Supplementary Results 1.4 and 1.5), the diagonal readout layer means that $d$ is equal to the number of actions. Thus $P_i(t)$ corresponds to the action preference for $A_i$ at trial $t$ (before softmax). A special case of $P_i(t)$ is equal to $\beta V_t(A_i)$ in cognitive models. We use $\Delta P_i(t) = P_i(t+1) - P_i(t)$ to denote preference changes between two consecutive trials. For the reversal learning task and three-armed reversal learning task, we consider $\Delta P_i(t)$ as an (approximate) linear function of $P_1(t), ..., P_d(t)$ for different (discrete) task inputs (that is, $\Delta P_i \sim \beta_0^{(P_i)} + \sum_{j=1}^d \beta_{P_j}^{(P_i)} P_j$). For the four-armed drifting bandit task, we further include the continuous reward $r$ as an independent variable (that is, $\Delta P_i \sim \beta_0^{(P_i)} + \beta_R^{(P_i)} r + \sum_{j=1}^d \beta_{P_j}^{(P_i)} P_j$).

For the original two-stage task, where there are three choice states (Supplementary Result 1.6), we focus on the three-dimensional model with a diagonal readout layer. Here, $L_1, L_2$ and $L_3$ represent the logits for $A_1/A_2$ at the first-stage state, logits for $B_1/B_2$ at the second-stage state $S_1$ and logits for $C_1/C_2$ at the second-stage state $S_2$, respectively. We similarly consider the regression $\Delta L_i \sim \beta_0^{(L_i)} + \sum_{j=1}^3 \beta_{L_j}^{(L_i)} L_j$.

Collecting all the $\beta_{L_j}^{(L_i)}$ (similarly for $\beta_{P_j}^{(P_i)}$) regression coefficients for a given input condition, we have the input-dependent state-transition matrix $\mathbf{A}$, akin to the Jacobian matrix of nonlinear dynamical systems:

$$\mathbf{A} = \begin{bmatrix} \beta_{L_1}^{(L_1)} & \beta_{L_2}^{(L_1)} & \cdots & \beta_{L_d}^{(L_1)} \\ \beta_{L_1}^{(L_2)} & \beta_{L_2}^{(L_2)} & \cdots & \beta_{L_d}^{(L_2)} \\ \vdots & \vdots & \ddots & \vdots \\ \beta_{L_1}^{(L_d)} & \beta_{L_2}^{(L_d)} & \cdots & \beta_{L_d}^{(L_d)} \end{bmatrix}$$

Note that the model-free RL models in these tasks are fully characterized by the collection of all regression coefficients in our dynamical regression.

## Symbolic regression

Apart from the two-dimensional vector field analysis, symbolic regression is another method for discovering concise equations that summarize the dynamics learned by RNNs. To accomplish this, we used PySR[70] to search for simple symbolic expressions of the updated dynamical variables as functions of the current dynamical variables for each possible input $I$ (for the RNN with $d = 2$ and a diagonal readout matrix). Ultimately, this process revealed a model-free strategy featuring the drift-to-the-other rule.

## Model validation via behaviour-feature identifier

We proposed a general and scalable approach based on a 'behaviour-feature identifier'. In contrast to conventional model recovery, this approach provides a model-agnostic form of validation to identify and verify the hallmark of the discovered strategy in the empirical data.

For a given task, we collect the behavioural sequences generated by models that exhibit a specific feature (positive class) and by those that do not (negative class). An RNN identifier is then trained on these sequences to discern their classes. Subsequently, this identifier is applied to the actual behavioural sequences produced by subjects.

We built identifiers to distinguish between the RNN models (positive class) and model-free RL models (negative class) in the reversal learning task, and between the RNN models (positive class) and model-based RL models (negative class) in the two-stage task. We presented the results in Supplementary Fig. 29.

## Meta-RL models

We trained meta-RL agents on the two-stage task (common transition: $\Pr(S_1|A_1) = \Pr(S_2|A_2) = 0.8$, rare transition: $\Pr(S_2|A_1) = \Pr(S_1|A_2) = 0.2$; see Supplementary Fig. 27) implemented in NeuroGym (v.0.0.1)[71]. Each second-stage state leads to a different probability of a unit reward, with the most valuable state switching stochastically ($\Pr(r = 1|S_1) = 1 - \Pr(r = 1|S_2) = 0.8$ or $0.2$ with a probability of 0.025 on each trial). There are three periods (discrete time steps) on one trial: Delay 1, Go and Delay 2. During Delay 1, the agent receives the observation (choice state $S0$ and a fixation signal), and the reward (1 or 0) from second-stage states on the last trial. During Go, the agent receives the observation of the choice state and a go signal. During Delay 2, the agent receives the observation of state $S_1/S_2$ and a fixation signal. If the agent does not select action $A_1$ or $A_2$ during Go or select action F (Fixate) during Delay periods, a small negative reward (−0.1) is given. The contributions of second-stage states, rewards, and actions on networks are thus separated in time.

The agent architecture is a fully connected, gated RNN (long short-term memory[58]) with 48 units[22]. The input to the network consists of the current observation (state $S_0/S_1/S_2$ and a scalar fixation/go signal), a scalar reward signal of the previous time step, and a one-hot action vector of the previous time step. The network outputs a scalar baseline (value function for the current state) serving as the critic and a real-valued action vector (passed through a softmax layer to sample one action from $A_1/A_2/F$) serving as the actor. The agents are trained using the Advantage Actor-Critic RL algorithm[72] with the policy gradient loss, value estimate loss, and entropy regularization. We trained and analysed agents for five seeds. Our agents obtained 0.64 rewards on average on each trial (0.5 rewards for chance level), close to optimal performance (0.68 rewards obtained by an oracle agent knowing the correct action).

## Reporting summary

Further information on research design is available in the Nature Portfolio Reporting Summary linked to this article.

## Data availability

All datasets used in this study are publicly available. The monkey dataset on the reversal learning task is available at https://data.mendeley.com/datasets/p7ft2bvphx (ref. 49). The rat dataset on the two-stage task is available at https://doi.org/10.6084/m9.figshare.20449140 (ref. 52). The three mouse datasets on the reversal learning task, two-stage task, and transition-reversal two-stage task are available at https://osf.io/8jwhm (ref. 50). The human dataset on the three-armed reversal learning task is available at https://github.com/psuthaharan/covid19paranoia. The human dataset on the four-armed drifting bandit task is available at https://osf.io/f3t2a (ref. 54). The human dataset on the original two-stage task is available at https://osf.io/usdgt (ref. 57).

## Code availability

The code used to reproduce the results in this paper is available at https://github.com/jil095/tinyRNN.

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

**Acknowledgements** This work was supported by the Kavli Institute for Brain and Mind (KIBM) Innovative Research Grant 2022-2209 and in part by National Science Foundation (NSF) awards CNS-1730158, ACI-1540112, ACI-1541349, OAC-1826967, OAC-2112167, CNS-2100237 and CNS-2120019, the University of California Office of the President, and the University of California San Diego's California Institute for Telecommunications and Information Technology/Qualcomm Institute. The authors thank the support from the Swarma Club and Causal Emergence Reading Group supported by the Save 2050 Programme jointly sponsored by the Swarma Club and X-Order. The authors thank CENIC for the 100 Gbps networks and K. T. Jensen, H. Xiong, Z. Jin and X. Li for their inspirational discussions.

**Author contributions** L.J.-A.: conceptualization, methodology, software, validation, formal analysis, investigation, data curation, writing–original draft, writing–review and editing, and visualization. M.K.B.: methodology, formal analysis, resources, writing–review and editing, supervision, and funding acquisition. M.G.M.: conceptualization, methodology, formal analysis, investigation, resources, writing–original draft, writing–review and editing, visualization, supervision, project administration, and funding acquisition.

**Competing interests** The authors declare no competing interests.

**Additional information**
**Correspondence and requests for materials** should be addressed to Marcelo G. Mattar.

# Reporting Summary

## Statistics

For all statistical analyses, confirm that the following items are present in the figure legend, table legend, main text, or Methods section.

| n/a | Confirmed | |
|---|---|---|
| ☐ | ☒ | The exact sample size (n) for each experimental group/condition, given as a discrete number and unit of measurement |
| ☐ | ☒ | A statement on whether measurements were taken from distinct samples or whether the same sample was measured repeatedly |
| ☐ | ☒ | The statistical test(s) used AND whether they are one- or two-sided<br>*Only common tests should be described solely by name; describe more complex techniques in the Methods section.* |
| ☒ | ☐ | A description of all covariates tested |
| ☒ | ☐ | A description of any assumptions or corrections, such as tests of normality and adjustment for multiple comparisons |
| ☐ | ☒ | A full description of the statistical parameters including central tendency (e.g. means) or other basic estimates (e.g. regression coefficient) AND variation (e.g. standard deviation) or associated estimates of uncertainty (e.g. confidence intervals) |
| ☐ | ☒ | For null hypothesis testing, the test statistic (e.g. $F$, $t$, $r$) with confidence intervals, effect sizes, degrees of freedom and $P$ value noted<br>*Give P values as exact values whenever suitable.* |
| ☒ | ☐ | For Bayesian analysis, information on the choice of priors and Markov chain Monte Carlo settings |
| ☒ | ☐ | For hierarchical and complex designs, identification of the appropriate level for tests and full reporting of outcomes |
| ☐ | ☒ | Estimates of effect sizes (e.g. Cohen's $d$, Pearson's $r$), indicating how they were calculated |

*Our web collection on statistics for biologists contains articles on many of the points above.*

## Software and code

Policy information about availability of computer code

| Data collection | This study did not collect data. No software was used for data collection. |
|---|---|
| Data analysis | Data were analyzed using Python 3.9 and Pytorch 1.13. Two-step tasks for meta-reinforcement learning were implemented in NeuroGym (version 0.0.1). Custom Code related to this study is available at https://github.com/jil095/tinyRNN. |

For manuscripts utilizing custom algorithms or software that are central to the research but not yet described in published literature, software must be made available to editors and reviewers. We strongly encourage code deposition in a community repository (e.g. GitHub). See the Nature Portfolio guidelines for submitting code & software for further information.

## Data

Policy information about availability of data

All manuscripts must include a data availability statement. This statement should provide the following information, where applicable:
- Accession codes, unique identifiers, or web links for publicly available datasets
- A description of any restrictions on data availability
- For clinical datasets or third party data, please ensure that the statement adheres to our policy

All datasets used in this study are publicly available.The monkey dataset on the reversal learning task (Bartolo et al.) is available at https://data.mendeley.com/datasets/p7ft2bvphx.The rat dataset on the two-stage task (Miller et al.) is available at https://doi.org/10.6084/m9.figshare.20449140.The three mouse datasets on the reversal learning task, two-stage task, and transition-reversal two-stage task (Akam et al.) are available at https://osf.io/8jwhm.The human dataset on the three-

## Research involving human participants, their data, or biological material

Policy information about studies with human participants or human data. See also policy information about sex, gender (identity/presentation), and sexual orientation and race, ethnicity and racism.

| | |
|---|---|
| Reporting on sex and gender | This study did not collect data from human subjects. Public human datasets were used. |
| Reporting on race, ethnicity, or other socially relevant groupings | This study did not collect data from human subjects. Public human datasets were used. |
| Population characteristics | This study did not collect data from human subjects. Public human datasets were used. |
| Recruitment | This study did not collect data from human subjects. Public human datasets were used. |
| Ethics oversight | This study did not collect data from human subjects. Public human datasets were used. |

Note that full information on the approval of the study protocol must also be provided in the manuscript.

# Field-specific reporting

Please select the one below that is the best fit for your research. If you are not sure, read the appropriate sections before making your selection.

☒ Life sciences  ☐ Behavioural & social sciences  ☐ Ecological, evolutionary & environmental sciences

For a reference copy of the document with all sections, see nature.com/documents/nr-reporting-summary-flat.pdf

# Life sciences study design

All studies must disclose on these points even when the disclosure is negative.

| | |
|---|---|
| Sample size | We analyzed preexisting behavioral datasets from multiple species and tasks:<br>- For the reversal learning task, we analyzed data from two monkeys (Bartolo dataset) and ten mice (Akam dataset).<br>- For the two-stage task, we analyzed data from four rats (Miller dataset) and ten mice (Akam dataset).<br>- For the transition-reversal two-stage task, we analyzed data from seventeen mice (Akam dataset).<br>- For human participants, we analyzed data from 1,010 individuals performing the three-armed reversal learning task (Suthaharan dataset), 975 individuals performing the four-armed drifting bandit task (Bahrami dataset), and 1,961 individuals performing the original two-stage task (Gillan dataset).<br>No statistical methods were used to predetermine sample sizes. All datasets were drawn from previously published studies, and we included all available subjects (with enough trials for modeling) in each task. For animal datasets, our analyses focused primarily on individual-level behavioral effects, where sample size is less revelant compared to group-level effects. For human datasets, we analyzed the largest publicly available datasets for each task, with sample sizes exceeding those used in most prior studies. These large cohorts provide sufficient statistical power to support robust group-level inferences and model-based analyses. |
| Data exclusions | Two rats (Miller dataset) performing the two-stage task were excluded and not analyzed due to the insufficient number of trials (each animal having fewer than 2000 trials). 57 participants (5.9%) who missed more than 10% of trials were excluded from Bahrami dataset.<br>No subject was excluded from other datasets. |
| Replication | Our analysis is based on individual-level data from publicly available behavioral datasets, originally collected in independently performed and previously published studies. While we did not perform new experimental replications ourselves, our modeling conclusions were robustly replicated across multiple independent datasets (eight datasets in total), five experimental paradigms, spanning three animal species (monkeys, mice, rats) and thousands of human participants. This extensive replication across independently conducted experiments demonstrates the generalizability and reproducibility of our findings. |
| Randomization | The datasets analyzed were derived from publicly available behavioral experiments previously reported in the literature. Allocation to experimental groups was not randomized by us; instead, randomization was previously performed by the original authors. As our modeling approach focuses on computational characterization of individual cognitive strategies rather than causal inferences across experimental groups, controlling additional covariates or further randomization is not relevant. |
| Blinding | The present study involves computational modeling and analysis of existing, publicly available behavioral datasets. It does not include any direct behavioral experimentation. Therefore, blinding was not required. |

# Reporting for specific materials, systems and methods

We require information from authors about some types of materials, experimental systems and methods used in many studies. Here, indicate whether each material, system or method listed is relevant to your study. If you are not sure if a list item applies to your research, read the appropriate section before selecting a response.

## Materials & experimental systems

| n/a | Involved in the study |
|-----|----------------------|
| ☒ ☐ | Antibodies |
| ☒ ☐ | Eukaryotic cell lines |
| ☒ ☐ | Palaeontology and archaeology |
| ☒ ☐ | Animals and other organisms |
| ☒ ☐ | Clinical data |
| ☒ ☐ | Dual use research of concern |
| ☒ ☐ | Plants |

## Methods

| n/a | Involved in the study |
|-----|----------------------|
| ☒ ☐ | ChIP-seq |
| ☒ ☐ | Flow cytometry |
| ☒ ☐ | MRI-based neuroimaging |

