## [Peer Review File · Nature]

Discovering Cognitive Strategies with Tiny Recurrent Neural Networks

Corresponding Author: Dr Marcelo Mattar

Version 1:

Reviewer comments:

Referee #1

(Remarks to the Author)

This paper presents a new method for uncovering the computational principles that underlie cognition in simple reinforcement learning tasks. The authors show that animal behavior in three such tasks can be fit well by recurrent neural networks, and perhaps more surprisingly, fitting well does not require many recurrent units in the networks (1-4) or all that much training data. The small size of the best fitting RNNs provides an interesting advantage in terms of interpretability – and the authors show that fitted RNNs can be interrogated using tools from dynamical systems to gain insights into the cognitive strategy used by the animals that they were fit to. The primary claim is that “tiny” RNNs provide novel insights into cognitive strategies.

Overall, I thought the approach was interesting, the work was rigorous, and the results made me think a lot about state of the art approaches to understanding cognition – and perhaps to question some of my own assumptions. That said, I am also not sure that the paper fully lives up to its primary claim, that is, to have provided novel insights into cognitive strategies/behaviors. There are several reasons for this. One basic one is with the novelty aspect of the claim – the authors do show that their best fitting models differ subtly from existing cognitive models – but there *are* existing cognitive models that seem qualitatively consistent with the reported results – and so it makes me wonder to what extent this approach is discovering strategies rather than recasting them in a language of dynamics (which, to my mind, is still also potentially useful). Another major issue is with the insights themselves. As far as I understand it, the insights come from: 1) showing that RNNs fit better and then 2) cracking open the RNNs to see how they did the task. But I personally wouldn't trust this approach if it were applied to standard cognitive models without considerable validation of the insights (i.e. can we identify specific aspects of behavior that should be sensitive to this insight and show how it works? One way to do this would be model recovery for artificial data.) and I have an additional concern with the RNNs here which is that I suspect that there may be multiple “strategies” that the RNN could use to fit behavior in these simple task to near perfection – meaning that the strategies themselves may not be identifiable in these simple tasks (which indeed are often used in the field, but lack the complexity to really arbitrate between different learning strategies). Overall, I feel the work is interesting, and if the primary claim were fully supported, would certainly be a very important paper in the field, however I believe the current manuscript falls short of providing full evidential support. In principle, I could imagine a revision of this paper that provides airtight evidence of the claims and really drives a paradigm shift in cognitive modeling, however I think such a paper would require a good bit more science to better assess and understand the observed results, as well as a pretty substantial reframing/refocus of the paper itself. A complete listing of my major concerns is below.

I am not confident that the “strategies” identified by the model are real. I have seen many papers claim that a cognitive model shows X, since the model equations include X, and since the model fits behavior better than other models. However, often, X has been later falsified, and the original result revealed to be a product of a poorly chosen task that was not diagnostic of X. There is no single solution to this problem, but I think at minimum, it is critical to identify behavioral hallmarks of the specific cognitive strategies of interest through simulation and validate them in the original data. That is, to identify a descriptive analysis that can “reveal” the strategy and show that it does so in RNN synthetic data and animal data, but not alternative model data. Without doing this, it is very hard to know *why* the RNNs are fitting the data better – and whether it is truly related to the specific cognitive strategies revealed, or some other difference between models. Secondly it is also critical to verify that tasks themselves can distinguish between subtly different strategies such as those picked up by RNNs that deviate slightly from existing cognitive models. I don't see any reason why the RNN-based strategies for inferring cognitive

mechanisms should be held to a lower standard than cognitive models typically are, and these standards are well described in Wilson & Collins (2019, eLife).

The paper claims to reveal novel cognitive strategies, but I was not convinced that this was the case. One of the cognitive strategies revealed by the RNN looked identical to the “correlated rewards” version of Q-learning, other than small deviations in the impact of evidence for/against a chosen option at high levels of certainty. This deviation is what you’d expect if you not only learned a Q value, but built inertia by integrating recent choices, which is commonly used in cognitive models. In the two-stage tasks, it is most common for fitting to include mixture models that combine different model elements (for example model-based and model-free learning) as these models tend to fit behavior better – it was not clear to me whether the best fitting RNN models for the two stage tasks were basically picking up on mixtures of the more basic strategies, or doing something unique. Overall, it seems that the current manuscript presents a bunch of possible revealed strategies, without carefully examining or validating any of them, leaving it a bit unclear whether they really deviate qualitatively from cognitive models fit to the same sorts of data.

The comparison of cognitive models to RNNs in terms of “d” seems pretty misleading. Clearly “d” means something very different in a cognitive model than an RNN. How many variables do I need to pass information about the past (ie. current state) into the next timestep? Probably just one, if I can reversibly compress the information – that is to say, if I need to store three binary variables, I could store them all in a single continuous variable (0-0.125 = 000, 0.125-2.5 = 001, and so on). For a small amount of precision loss, I could do the same thing with continuous variables, and the RNN is well posed to do this – it has gating parameters that could nonlinearly “unpack” the previous timestep in ways that are not afforded to standard cognitive models. For this reason, I think comparing ‘d’ for cognitive and computational models is like comparing apples and oranges. I am not sure that I have a better alternative measure of complexity, but I do think that the authors should either come up with one, or refrain from comparing d across the model sets and interpreting it as a meaningful comparison.

On a related note, it is shown that the RNNs outperform alternative strategies in terms of log-likelihoods, but the presentation here needs much more context. In particular, it is not clear how scientifically (or statistically, actually) significant the differences in log-likelihood are. The reader should not have to trust the authors that a difference of 0.05 or 0.1 points in log-likelihood fit are either descriptively or mechanistically meaningful, though of course it is possible that they are. Given how heavily optimized cognitive models are (via the scientific process) our prior expectations should be that they describe the scenarios they were developed for extremely well. As such, showing that the RNNs give superior descriptions requires establishing that there was previously meaningful room for improvement, and that the RNNs are capturing important components of behavior being missed by the cognitive models. Some of the behavioral analyses start to go in this direction, but it’s far from firmly established.

Finally, while the authors note that a major issue with RNNs has been with interpretability, another issue with RNNs has been that of robustness and generalization. Getting RNNs to fit data is often trivial, but getting them to generalize to an untrained condition is often much more difficult, and perhaps a better test of whether the model is providing information that is theoretically useful (ie. revealing underlying computational principles) versus curve fitting to the training data. I had a bit of difficulty understanding exactly how the training/testing was done but my understanding is that it did not include any test of model generalization. I think supporting the core claim, that tiny RNNs can reveal cognitive strategies, would require showing at least some level of generalization beyond the training data. Perhaps an alternative would be to frame the claim instead as RNNs providing a low dimensional description of the data, rather than revealing the underlying cognitive strategies, but obviously this would be a slightly less impressive conclusion. More generally, there was not sufficient information about the holdout/cross validation procedure to fully evaluate it and the authors should be clear on exactly how models were trained/tested/evaluated.

Referee #2

(Remarks to the Author)

This article presents a novel set of tools to study the mechanisms that support behavior in non-human animals. The motivation to develop this tool is the recognition that behavioral modeling is an incremental process of discovery, with no clear signal of when a satisfactory model is obtained. The hope of developing these tools is to 1) automate discovery, and 2) identify a ceiling of explainability for a given data set. The novel set of tools consists in two main steps: 1) using tiny (1-4 GRUs) RNNs to build a predictive model of animals’ behavior; 2) using dynamical systems approaches to interpret the trained RNNs. The first step also provides an estimation of the dimensionality of behavior. The authors show convincingly in three specific data-sets that their approach works well – i.e. is more predictive than classic families of models on these tasks, and may enable discovery of new mechanisms.

This is an exciting novel approach, tackling an important problem. The paper is didactic and thorough. It has the benefit of managing individuals, rather than groups, as has been done previously, and enabling a comparison of different types of models on fair grounding. Nonetheless, it is unlikely that this approach will generalize well to a broad class of behavior, and the article vastly oversells its promise on multiple points. I note that the authors anticipated and attempted to address some of my concerns, but I do not think this was successful. I elaborate below.

1. First, the article should vastly tone down claims on multiple aspects. The use of “automatic” in the title and abstract is *not* warranted. The RNNs are indeed trained automatically, but once they are trained, researchers only have a tool to predict behavior, they absolutely have not discovered new mechanisms. This step fully relies on the second set of tools based on dynamical systems, and is far from automatic or accessible (see point 5).

2. Second, the authors emphasize *tiny* RNNs, and say *1-2* units in the abstract and intro, even though one of the tasks necessitates 4. This is not a quibble, as the difficulty of the second step (interpretation) is very non-linear in the number of units. Furthermore, it is very likely that other tasks will necessitate more units (see next point).

3. Scaling to other tasks. While the authors select three tasks of seemingly increasing complexity to attempt to show generalizability of the method, the three tasks share two important features that likely make behavior dimensionality low: the fact that choices are binary, and the fact that there is only one choice state. If either of those is untrue, as is the case even in non-human animal tasks (e.g. Rikhye et al 2018; Sweis et al 2018, Science) and frequently in human cognitive tasks (e.g. probabilistic selection task, Frank et al 2004), the dimensionality will have to be significantly higher. To convincingly argue that this tool is generalizable, it would have been better to use tasks that increase not in design complexity, but in complexity along those dimensions. There is a difference between the claim that low-D GRU nets don't overly constrain solutions on this task, and the claim that they won't in general for cognitive tasks.

4. The authors attempt to make the case that the method will be applicable to humans (where many fewer data points are available for training) via a knowledge distillation framework. The approach is very interesting, and might potentially work, however, the treatment is much too superficial to be convincing. For example, it is only tested in one non-human animal with the most data, and is not tested with actual human data, which is likely to be much higher dimensional and have very different data sizes/statistics. It might be better left for supplements or follow up work where it can actually be thoroughly tested. As it is, with the emphasis on "cognitive modeling" rather than "behavioral", the readers really get the impression that this is applicable to complex cognitive tasks and humans (despite the authors honest stating of "animal" in the abstract); and I think this is highly misleading at this point in the tool validation.

5. Second set of tools – dynamical systems. First, as stated above, this component is highly non automatic, and really pushes back the problem of discovery to another complex interpretation problem; it is not obvious that in many cases this problem will be easier than the one it is trying to replace. Furthermore, the tools offered only apply to specific cases. For example, the logit analysis requires only two choices. The authors acknowledge this late in the paper, and say that it doesn't matter as one can always do one choice vs. all others – but are insights likely to be easily discovered that way? It is known that multiple choices impact decision making in non-trivial ways, that can be discovered by considering errors, for example; something that would not be easily handled by this approach. Same, as soon as the dimensionality is $d > 1$, the tools become even less tractable, in a way that is highly non-linear with task complexity (e.g. need to plot per possible task event, which could increase multiplicatively with number of states, choices, outcomes, etc...)

6. Dynamical systems:

- a. validity of constant input regimes. In a real experiment participants may not go anywhere near the fixed points, so (a) networks will be doing out of sample generalization which is notoriously finicky for neural networks and (b) the important dynamical considerations may be obscured by effectively irrelevant fixed point topologies.
- b. Is the constant input of a single stimulus a realistic regime for any of the models? The focus on asymptotic behavior, while understandable from an analytic approach, constrains little when the system is rarely at rest dynamics can be vastly different to those implied under the static 1 input regime (e.g. oscillations in 1d system)

Minor comments:

- Dynamical variables = units in RNN, but not defined in cognitive models in the main text
- Fig. 2-3 and others: likelihoods are plotted as averages across folds and individuals. It would be important to get a sense of uncertainty, either by showing individuals, or at minimum by including distributions/errorbars.
- Fig. 4. Measures RNNs' ability to predict behavior. It is known that a model can predict well without generating behavior well (Palminteri, Wyart, Koechlin 2017). The authors visualize some success for GRUs in figure 1, but do not quantify this elsewhere.
- Have authors considered looking into what optimal behavior dimensionality is in each task, and seeing how this compares to RNNs? I.e. are the 1, 2, 4 found for the three task due to the fact that this is the dimensionality needed to solve the tasks?
- Another reason it might not generalize to some tasks is that the assumption that data can be separated into folds for cross-validation. In many tasks, there are long-term dependencies on the data that make such approaches non accessible. Even when such dependencies exist (e.g. if participants do multiple independent blocks of the same task), meta-learning across blocks is frequent.
- It would be important to acknowledge that the rule discovery process would be served by researchers analyzing data more systematically in a model independent way, rather than only focusing on reproducing unique model-predicted behavioral patterns. The discovered rules given in the paper, for example, should be identifiable by careful data visualization.
- Drift to other rule – the forgetting rule implemented by the authors drifts to 0. This is not what all models use – for example Collins & Frank 2012 instead drift towards initial value (which is not 0, but a prior expected value in between 0 and 1), such that bad outcomes are forgotten towards better values, and good ones toward lower values, indeed both drift towards indifference. Have authors considered this more reasonable version of the forgetting capture behavior as well as the "drift to the other" one? If so, the discovery would only be due to a poor implementation choice in the first place.
- Fig 7 Meta learning RL. The claim that task optimized RNNs have not been analyzed using a dynamical systems approach is wildly incorrect (both in general and in the neuro-cognitive modelling field).
- Stable and unstable fixed points should be represented differently.
- Are there no analytic results for the types of dynamics 1d and 2d GRUs can represent (# number of fixed points, oscillations etc.) beyond just sampling randomly (fig S15)? See for example Jordan, Sokol, Park ICLR 2019 (not peer-reviewed)
- There was surprisingly little focus on individual differences (only in supplement and little discussion). It seems like one of

the main strengths of tiny GRUs is the ability to fit to individuals' data, and bring a more nuanced analysis of individual fits (e.g., in what regions of the phase plane do the animals differ?)

Referee #3

(Remarks to the Author)

Ji-An Li and colleagues use tiny RNNs to capture and understand learning and decision making in rodents and non-human primates. The small size of these RNNs greatly facilitates the interpretability of the networks' properties. Moreover, the authors make another important contribution by demonstrating how tools used to measure dynamical systems can be applied to neuroscience/psychology/artificial intelligence questions and data to represent and further understand model predictions and agents' behaviors. The data and conclusions presented in the current manuscript are novel, important, amazing on their own in terms of the insights they bring to cognitive sciences dealing with either biological or artificial agents. However, this is just the tip of the iceberg. The combination of minimally complex neural networks with interpretive tools from dynamical systems has massive potential to answer longstanding questions and open up numerous new avenues of inquiry.

It has already shown that canonical approaches to modeling learning and decision making fail to explain several aspects of rodent and non-human primate behavior. That result is perhaps not surprising because we know biological agents deviate from (constraint-free) optimal learning and decision rules. Crucially though, this paper gives us a new set of tools to better identify and then understand how and why biological agents' behavior diverges from normative approaches.

If there is a downside to this paper, it is that there is so many interesting findings and implications from this work that the authors must treat some of them rather superficially. To be clear, I think they've achieved a good balance of depth and breadth given space constraints. It is obvious that this work is just the beginning of a super exciting phase of discovery that will be spurred on by these innovations and results.

I don't have any major concerns with this work, but here are some points that I think are worth further consideration.

1. This may be beyond the scope of a paper already crammed full of cool new things, but one thing I'd be interested to read the authors' thoughts on is how to aggregate and/or compare the RNN fits across groups of agents. The paper focuses on fits to individual agents. Most often in studies of biological agents, the goal is to compare agents in terms of some categorical variable (intervention vs control, diseased vs healthy, etc) or a continuous measure of individual differences. I think there are a number of ways one could go about using either the fitted networks and/or their dynamical properties to compare and contrast agents. Could the authors give some guidance on the most appropriate methods in this regard, or ideally add an example implementation to the paper? I realize that including an implementation of biological agents is probably infeasible given the low number of animals in each study used here. However, even an implementation with simulated agents designed to differ in some way may be very helpful for many readers.

2. A key point of the paper – as far as I understood it - is that the tiny RNNs that best fit and reproduce the animals' behavior on these tasks do not correspond to ANY of the canonical models proposed to underly such behavior in human and non-human animals. However, the first paragraph of the Discussion states, "our approach demonstrated that animal behavior aligns more closely with model-free RL algorithms than alternative cognitive models." What data or comparison is this statement based on? Figure 5b and c does show that the GRU is more similar to model-free than latent state, but still qualitatively different than both, in one monkey on the reversal learning task. However, Figure 5e shows that MF predicts different preference orders than the GRU on the Miller two-stage task after uncommon transitions, while both latent-state and model-based at least get the preference ordering correct. All cognitive models are clearly a poor quantitative fit to the pattern generated by the GRU, but it is hard to see how we could judge the pattern to be closest to MF (in that task at least). Again, the point of the paper is that none of these cognitive algorithms are a good qualitative or quantitative fit to animals' behavior, even on these very simple behavioral tasks. The tiny RNN approach lets us discover and describe functions that actually match animals' behavior. So the sentence noted above seems to me to be both misleading and unnecessary.

Version 2:

Reviewer comments:

Referee #1

(Remarks to the Author)

The authors have done an impressive job of addressing the criticisms raised by myself and the other reviewers. I believe that the revised manuscript is considerably improved, and only have a few remaining concerns.

The first has to do with magnitude of improvements in fit. The tiny RNNs fit better, but not by much. It is certainly not clear whether the differences between the best RNN and the best cognitive model are meaningfully different. From figure 1e it appears the differences between RNN models and cognitive models are incredibly small, in real terms (~1% difference in likelihood of chosen action per trial) – raising question as to whether the difference is meaningful – I guess, to some extent, this is addressed by the fact that the authors show that they can use the RNN fits and analysis methods to inspire changes to

the cognitive models that improve fits... however, I think at the higher level, there is always a question of whether one is discovering principles (ie. things we expect to generalize beyond this particular task or model system), and while I don't think that the authors necessarily need to prove that their method does this, I think some balanced discussion about the issue would give the reader a better perspective on what the results mean.

My second concern is that I still feel like the paper overstates the novelty of the mechanisms uncovered using their small RNNs. For example, the authors say:

"These three signatures — "state dependent learning rate", "preference-dependent perseveration", and "reward-dependent bias" — are absent from all cognitive model variants considered and from any published analysis of these tasks in the literature."

While this may be true in these particular task variants, it is not like these ideas are completely new.

I assume that by "state dependent learning rate" the authors refer to the curve in the dark blue and red lines in figure 3c, which that the logit change associated with a reward are largest when that reward was inconsistent with the action preference of the model. It may be that this hasn't been noted in the particular reversal learning task that is studied here, but there is whole field of study on adaptive learning rates, stemming from animal models of learning (Pearce Hall) and connecting to Bayesian inference in changing environments (Nassar 2010), that shows in general terms what is observed in the RNN dynamics on rewarded trials, namely that prediction errors carry more influence on behavioral updating when they are unexpected, and therefore more likely to reflect a mismatch between stored expectations and the current environment.

I think when the authors say "Preference dependent perseveration" they are referring to the curving pink and light blue lines in fig 3c. If so, I think this effect relates to the finding that "learning rate" as implemented in humans at least, is not deterministic, but instead includes trial-to-trial variability (Findling 2019). Since this variability is not observable to the experimenter (and fit models), it means that actions provide some "clue" as to the unobservable state of the system (current level of action preference). Thus, observing an unrewarded action that is inconsistent with the "inferred" choice preferences from a model can indicate that the animal actually has a less extreme choice preference than the model-estimated one. This has been captured in cognitive models that incorporate stochastic trial by trial learning rates, which provide a better fit to data, and my guess is that it would produce something similar to the light blue/pink curves.

I'm a bit unsure what reward dependent bias means in terms of the dynamics plot – so I find it harder to comment on that one.

But the point is that I am not particularly convinced that the signatures of learning revealed by the RNN are new, they seem quite related to principles that have been identified and characterized in other contexts. Just to be clear, I do not see this as a weakness of the method or paper. If the RNN framework can reveal signatures of behavior in a single task that are closely related to principles that have been observed in other tasks, this provides evidence that it can be used as a scientific tool, not only to describe data collected within a particular task paradigm, but to generalize principles that extend beyond it. Indeed, I would be highly skeptical if the RNNs revealed a completely new set of principles that bore no resemblance to empirical and theoretical work done over the last 70 years of studying learning. That said, I DO think that the authors could provide a much more balanced discussion of the contributions of their methods, and their relationship to the broader understanding of learning that has been shaped by cognitive models, and their view of how these tools might be combined to aid scientific discovery going forward.

I still have some reservations about using the same x-axis to compare cognitive and RNN models in figure 1E. I appreciate the point that the authors are making and their response to my specific concern in the last round. However, I would just ask that the authors to 1) state clearly the working of definition of dimensionality that they are using in the paper, 2) differentiate it from the definition that relates to expressivity and parameters that is often used in machine learning research, and 3) acknowledge that the working definition of dimensionality (counting dynamical variables) is imperfect in its assessment of encoding dimensions in that that two different models might encode task relevant information more or less efficiently across available dynamical variables. I think it also would be fair to mention that the RNNs are incentivized to make efficient use of their dynamic variables, whereas the variables in cognitive models are constructed by hand, typically without any prioritization of such efficiency... though I guess this may be a matter of opinion.

Minor:

"In particular, RNNs outperformed all ideal Bayesian observer models, which perform exact inference based on knowledge of the task structure, suggesting that animal behavior in these tasks is not optimal."

The authors present this as a finding of their method, but this is an observation that could be made simply from comparing the cognitive models – and indeed, has been made previously in the literature many times.

It is noteworthy that Animal behavior on 2 step task looks considerably different from human behavior on 2-step tasks... perhaps this

"Even after convergence, we identified subtle deviations from the Bayesian inference model (Fig. 5b-c). "

I think the authors mean to refer to figures 5a-b here?

Referee #2

(Remarks to the Author)

We thank the authors for a very thorough and responsive revision that we think has very significantly improved the manuscript. We appreciated the new tasks analyses, the new dynamical regression method, the new cross-validation method, and the new model comparison metrics. We agree the paper is now ready for publication. We add here three comments; two of them are minor and more curiosity on our part and shouldn't hold back publication. One we think is important to handle before publication, but should be a formality for the authors to handle.

1. Code availability. Currently, this is stated as "available from the lead author upon request". In 2024, for a paper fully based on developing a new method, this is absolutely not sufficient. Full, reproducible code should be posted on an open repository and fully available. We think this needs to be changed before publication.
2. Minor quibble/question, does not need to be addressed before publication. We are uncertain on theorem 1 and 2 in section 2.3 of the supplement (and figure S40). We're not confident here, but we feel this is somewhat "hand wavy". It feels like a real trained embedding of a one dimensional variable into two dimensions could be formed to maximize similarity of adjacent points along the natural manifold of activity in that two dimensional space better than Hilbert Curves which appear quite naive. Not clear these proofs are relevant to the paper given the nature of trained representations.
3. Line 190: "We found that student RNNs with 5-20 units provided the best fit to human behavior, despite the limited trials per subject." It looks like authors only tested up to 20 units, which leaves open the possibility that more units would have fit even better. However we agree though that this doesn't diminish the main overall claim of this section that tiny RNNs improve upon cognitive models of equal dim for more fully encapsulating behavioral strategies, so again, this is a minor quibble that shouldn't block acceptance.

(Remarks on code availability)

Code is not available, so that code review was not possible. See main review.

Referee #3

(Remarks to the Author)

The authors have done extensive work to address all reviewers' comments. My concerns have been sufficiently addressed and I have no further comments. I still think this a great paper.

Version 3:

Reviewer comments:

Referee #1

(Remarks to the Author)

The authors have fully addressed my concerns and I believe the revised manuscript makes an impressive contribution to the field.

RESPONSE TO REVIEWERS

We appreciate the reviewers' insightful and constructive feedback, which has guided a substantial revision of our manuscript. The key improvements include:

1. **Expanded task complexity.** We added three new datasets with more complex reward-learning tasks: a three-armed reversal learning task, a four-armed drifting bandit task with continuous rewards, and the original two-stage task with multiple choice states. This addresses generalizability concerns.
2. **Human behavior modeling.** We demonstrate the applicability of our approach to human datasets with limited per-subject data using knowledge distillation.
3. **Enhanced dynamical systems analysis.** We developed a novel dynamical regression method to interpret vector fields in higher-dimensional models, enabling analysis of tasks with increased complexity.
4. **Robust training protocols.** We introduced interspersed split and cross-subject split protocols to train and evaluate models in human datasets with few trials per subject.
5. **Rigorous model validation.** We performed extensive analyses including model recovery, parameter recovery, robustness testing, and behavior validation. We also introduced a novel behavior-feature identifier method, ensuring alignment between model predictions and key behavioral signatures.
6. **Expanded cognitive model comparisons.** We included additional cognitive models for each task, such as model-free variants and hybrid model-free/model-based algorithms.
7. **Group-level analysis.** We illustrate how to aggregate and compare results across individuals, strengthening the generalizability of our findings.
8. **Manuscript restructuring.** We reorganized and compressed the main text to incorporate new findings and conform to Nature guidelines.

This document includes detailed responses to each reviewer comment. Editor and reviewer comments are shown in black, our responses in blue, and quotes from the revised manuscript in green.

Reviewer 1

Summary

This paper presents a new method for uncovering the computational principles that underlie cognition in simple reinforcement learning tasks. The authors show that animal behavior in three such tasks can be fit well by recurrent neural networks, and perhaps more surprisingly, fitting well does not require many recurrent units in the networks (1-4) or all that much training data. The small size of the best fitting RNNs provides an interesting advantage in terms of interpretability – and the authors show that fitted RNNs can be interrogated using tools from dynamical systems to gain insights into the cognitive strategy used by the animals that they were fit to. The primary claim is that “tiny” RNNs provide novel insights into cognitive strategies.

Overall, I thought the approach was interesting, the work was rigorous, and the results made me think a lot about state of the art approaches to understanding cognition – and perhaps to question some of my own assumptions. That said, I am also not sure that the paper fully lives up to its primary claim, that is, to have provided novel insights into cognitive strategies/behaviors. There are several reasons for this. One basic one is with the novelty aspect of the claim – the authors do show that their best fitting models differ subtly from existing cognitive models – but there *are* existing cognitive models that seem qualitatively consistent with the reported results – and so it makes me wonder to what extent this approach is discovering strategies rather than recasting them in a language of dynamics (which, to my mind, is still also potentially useful). Another major issue is with the insights themselves. As far as I understand it, the insights come from: 1) showing that RNNs fit better and then 2) cracking open the RNNs to see how they did the task. But I personally wouldn't trust this approach if it were applied to standard cognitive models without considerable validation of the insights (i.e. can we identify specific aspects of behavior that should be sensitive to this insight and show how it works? One way to do this would be model recovery for artificial data.) and I have an additional concern with the RNNs here which is that I suspect that there may be multiple “strategies” that the RNN could use to fit behavior in these simple task to near perfection – meaning that the strategies themselves may not be identifiable in these simple tasks (which indeed are often used in the field, but lack the complexity to really arbitrate between different learning strategies). Overall, I feel the work is interesting, and if the primary claim were fully supported, would certainly be a very important paper in the field, however I believe the current manuscript falls short of providing full evidential support. In principle, I could imagine a revision of this paper that provides airtight evidence of the claims and really drives a paradigm shift in cognitive modeling, however I think such a paper would require a good bit more science to better assess and understand the observed results, as well as a pretty substantial reframing/refocus of the paper itself. A complete listing of my major concerns is below.

We are grateful for the reviewer's positive feedback and recognize the importance of the concerns raised. We took these concerns seriously and performed several additional analyses aimed at validating the discovered insights. Below, we offer a detailed, point-by-point response to each comment.

1.1

I am not confident that the “strategies” identified by the model are real. I have seen many papers claim that a cognitive model shows X, since the model equations include X, and since the model fits behavior better than other models. However, often, X has been later falsified, and the original result revealed to be a product of a poorly chosen task that was not diagnostic of X. There is no single solution to this problem, but I think at minimum, it is critical to identify behavioral hallmarks of the specific cognitive strategies of interest through simulation and validate them in the original data. That is, to identify a descriptive analysis that can “reveal” the strategy and show that it does so in RNN synthetic data and animal data, but not alternative model data. Without doing this, it is very hard to know *why* the RNNs are fitting the data better – and whether it is truly related to the specific cognitive strategies revealed, or some other difference between models.

We understand the reviewer's skepticism and we agree that it is crucial to determine whether the discovered cognitive strategies are real. As correctly noted, the tasks in our study were designed to isolate a specific set of cognitive processes. Thus, those tasks may not be sufficiently diagnostic of the cognitive strategies discovered by RNNs, leading to challenges in the model discovery process (as an example, distinguishing between model-based RL and Bayesian inference strategies can be rather challenging in the two-stage task, which was originally designed to distinguish between model-free RL and model-based RL strategies¹). We thus agree with the reviewer about the importance of directly validating any inferred strategies. With this goal in mind, our revision includes several new analyses (also described in the various parts of this document), each designed to rule out a different set of alternative explanations. As we will show, these analyses provide converging and unequivocal evidence that the identified strategies are real.

Our first approach was to perform a standard model recovery analysis. This analysis aims to identify the true data-generating model through model comparison techniques, using each competing model to simulate artificial (choice) data and subsequently fitting the simulated datasets with each competing model. A model is deemed recoverable (and, thus, identifiable) if it can fit the data it generates more accurately than any alternative model. In other words, the generated data must contain signatures of the data-generating model that are not present in any alternative model.

Using model recovery approach outlined above, we simulated data from a 1D or 2D RNN with parameters fitted to the choices of one animal in the reversal learning and two-stage tasks (termed “RNN-Sim” below for simplicity). We found that the simulated data was better fit by an RNN of equal dimensionality than by any alternative cognitive model (Supplementary Results 1.1, Fig. S4 Right — copied here as Fig. R1—, S5 Right, S6 Right, S7 Right). In other words, the RNN-Sim produced behavioral patterns that no other cognitive model predicted. Crucially, the same RNN-Sim also predicted behavioral patterns found in the data and absent from other models, as evidenced by its superior predictive performance in the experimental data (Fig. 1e, S10 — copied here as Fig. R2).

Figure R1: **Tiny RNNs can predict choices as well as the ground-truth model in the two-stage task, measured by negative log-likelihood.** (a-b) The performance of each model in predicting the choices generated by a cognitive model in the two-stage task, with choice stochasticity arising due to sampling from the policy. Performances are displayed as a function of the number of dynamical variables d , plotted on a log scale along the x-axis. Each panel corresponds to choices generated by a different cognitive model. Dashed circles indicate the predictive performance of the ground-truth cognitive model — i.e., a model fitted to the choice data generated by the same model. Note that the RNNs have a similar predictive performance as the ground-truth model, suggesting that it can accurately identify the ground-truth strategy. (a) Simulated behavior generated by three models fitted to an example rat: a Bayesian inference (latent-state) agent with $d = 1$ (left), a model-based RL agent with $d = 1$ (middle), and a GRU agent with $d = 1$ (right). (b) Simulated behavior generated by three models fitted to an example rat: a model-free (Q(1)) agent with $d = 2$ (left), a model-based RL agent with $d = 2$ without value forgetting (middle), and a GRU agent with $d = 2$ (right). The reported test performance for each model is the average over 10 outer folds in the nested cross-validation. Error bars (almost invisible) show the average SEM (standard errors first calculated across outer and inner loops, then averaged across individuals). Each model family may include multiple model variants, leading to two or more identical markers for a given d .

Conducting a similar model recovery analysis for the cognitive models (Fig. S4 — copied here as Fig. R1—, S5, S6, S7), we found that simulated data was predicted equally well by the data-generating model and by a fitted RNN of equal dimensionality. This suggests that any behavioral pattern predicted by a cognitive model can also be predicted by an RNN. In other words, there are no patterns in the data that one of the tested cognitive models predicts, but not an RNN of equal

Figure R2: **Tiny RNNs outperform classical cognitive models in predicting animals' and humans' choices at the group level, measured by predictive test accuracy.** Performances are displayed as a function of the number of dynamical variables d , plotted on a log scale along the x-axis. (a-c) The reported test accuracy for each model is the average over 10 outer folds in the nested cross-validation and then over individuals. Error bars show the average SEM (standard errors first calculated across outer and inner rounds, then averaged across individuals). (a) (Top) Monkeys (Bartolo et al.) performing the reversal learning task. (Bottom) Mice (Akam et al.) performing the reversal learning task. (b) (Top) Rats (Miller et al.) performing the two-stage task. (Bottom) Mice (Akam et al.) performing the two-stage task. (c) Mice (Akam et al.) performing the transition-reversal two-stage task. (d-f) The reported test accuracy for each model is the average over unseen test trials (using the interspersed split protocol) and then over individuals. Error bars show the SEM across individuals. (d) Humans (Suthaharan et al.) performing the three-armed reversal learning task. (e) Humans (Bahrami et al.) performing the four-armed drifting bandit task. (f) Humans (Gillan et al.) performing the original two-stage task. Each model family may include multiple models, leading to two or more identical markers for a given d .

dimensionality. Together, these results confirm that the fitted RNNs (e.g., RNN-Sim) predict behavioral patterns that can be found in the data and that are not predicted by classical cognitive models.

The model recovery analysis described above is described in the revised manuscript starting from line 149:

To evaluate the limits of this flexibility, we simulated the behavior of RL and Bayesian inference agents in the reversal learning and two-stage tasks and fitted both RNN and cognitive models to these synthetic data (see Supplementary Results 1.1, Fig. S4, S5, S6, S7). We found that the tiny RNNs achieved predictive performance similar to the ground-truth model that generated the behavior, with the best-performing RNN having the same dimensionality as the ground-truth model. These results suggest that RNN models can serve as a superset of classical cognitive models despite using a single architecture consistently across tasks and datasets and requiring only minimal manual engineering. Incidentally, these results also demonstrate that these cognitive strategies are identifiable and robustly recoverable (see Supplementary Results 1.1 and Fig. S11, S13), and that our training procedure successfully prevented overfitting, which can be diagnosed when the fitted model achieves higher performance on the training dataset and lower on the test dataset relative to the data-generating model.

While the analysis above suggests that the fitted RNNs are recoverable as a whole (it can fit the data it generates more accurately than alternative models), it says nothing about *specific* predictions of the model. To empirically validate a specific model prediction (e.g., an inferred cognitive strategy), we followed an approach similar to model recovery as suggested by the reviewer. The approach involves generating artificial data by a set of models *that makes that specific prediction*, and by an equivalent set of models whose only difference is that they *don't make that same specific prediction*. The specific model prediction is validated if the observed empirical data contain signatures found in the artificial data generated by the first, but not the second set of models.

An intuitive implementation of this idea is to identify a behavioral hallmark for a given cognitive strategy, ensuring that this hallmark is observed in artificial data simulated by any model predicting that cognitive strategy, but not in artificial data simulated by any model not predicting the same strategy. The existence of such behavioral hallmark in the empirical data validates the cognitive strategy. In our manuscript, we followed this approach to validate some of the inferred cognitive strategies, such as the presence of a preference-dependent choice perseveration in the behavior of monkeys in the reversal learning task (Fig. S14a-d). Unfortunately, this approach is difficult to scale due to the difficulty in defining a behavioral hallmark for some cognitive strategies (similar to the challenge in distinguishing between model-based RL and Bayesian inference strategies in the two-stage task¹).

In our revised manuscript, we propose instead a more general and scalable approach based on a “behavior-feature identifier” (see Response 1.2 below and Methods). In contrast to conventional model recovery, this approach provides a model-agnostic form of validation to identify and verify the hallmark of the discovered strategy in the empirical data. For a given task, we collect the behavioral sequences generated by models that exhibit a specific feature (positive class) and by those that do not (negative class). An RNN identifier is then trained on these sequences to discern their classes. Subsequently, this identifier is applied to the actual behavioral sequences produced by subjects. In our data, this approach indicates that the actual behavioral sequences are consistently classified as the positive classes, thereby validating the strategies discovered by the RNN models (Fig. S27 – copied here as Fig. R3).

The behavior-feature identifier described above is mentioned in the revised manuscript starting from line 246:

Crucially, we validated each of these insights using targeted hypothesis testing (see Supplementary Results 1.2, Fig. S14) or using a novel behavior-feature identifier approach (see Methods and Fig. S27).

In sum, we have confirmed the validity of the inferred strategies using two approaches, one general to the model as a whole, and one specific to each model prediction. Using model recovery, we found that some patterns in the empirical data could be predicted by the RNN but no cognitive model; conversely, no patterns could be predicted by a cognitive model but not by the RNN. Using our proposed “behavior-feature identifier”, we were then able to validate the specific cognitive strategies discovered by the RNN.

1.2

Secondly, it is also critical to verify that tasks themselves can distinguish between subtly different strategies such as those picked up by RNNs that deviate slightly from existing cognitive models. I don't see any reason why the RNN-based strategies for inferring cognitive mechanisms should be held to a lower standard than cognitive models typically are, and these standards are well described in Wilson & Collins (2019, eLife).

Figure R3: **Model validation via behavior-feature identifier.** For a given task, we collect the behavioral sequences generated by the GRU model (with discovered features to be validated; positive class shown in blue) and by an alternative cognitive model (without GRU-discovered features; negative class shown in orange). We then train an RNN identifier on these sequences to predict their classes. This identifier is then applied to the actual behavioral sequences produced by subjects (shown in green). Each thin line represents a sequence of choices. Each thick line and corresponding shaded region is the average and standard error of model evidence across all sequences from a given model or the subject. (a) GRU (d=1) versus model-free strategy (d=1) fitted to one example monkey's behavior in the reversal learning task. (b) GRU (d=2) versus model-free strategy with value forgetting (d=2) fitted to one example monkey's behavior in the reversal learning task. (c) GRU (d=1) versus model-based strategy (d=1) fitted to one example rat's behavior in the two-stage task. (d) GRU (d=2) versus model-based mixture strategy (d=2) fitted to one example rat's behavior in the two-stage task. (e) GRU (d=1) versus model-based mixture strategy (d=2) fitted to one example rat's behavior in the two-stage task. These identifiers predicted that the actual behavioral sequences belong to the positive classes, thus successfully validating the strategies discovered by GRU models.

The reviewer raises another important point about the identifiability of strategies learned by RNNs given the available data. If different RNNs can fit behavior equally well, with each predicting a different strategy, interpreting a particular set of strategies while ignoring others can lead to incorrect conclusions.

One way to verify that the strategies revealed by a particular RNN are identifiable is via model recovery, as discussed in Response 1.1. For example, we can use the RNN fitted to empirical data to generate a synthetic dataset of similar size. We then fit a similar RNN to the synthetic dataset just generated. Since the synthetic data involves multiple random samples from the agent's policy, each dataset can lead to a potentially different fit and, ultimately, to different cognitive strategies. Using this approach in our data, we found that the RNNs recovered the ground-truth model dynamics (see Supplementary Results 1.1 and Fig. S11 — copied here as Fig. R4 —, and S13 — copied as Fig. R5). The phase portrait of the RNN fitted to synthetic data was strikingly similar to the phase portrait of the RNN fitted to the empirical data, leading to the identification of the same cognitive strategies. This suggests that the cognitive strategies discovered in the empirical data are unique and identifiable by the RNNs.

The approach above relies on synthetic data generation. While this approach is methodologically sound, our actual goal is to determine if the strategies discovered by our approach are robust to variations in the actual empirical data. In other words, would the same strategies be discovered by the various RNNs fitted to distinct realizations of the random processes in an experiment (e.g., random samples of stimuli, random samples from the animal's policy, etc.)? Obviously, our empirical data is a unique realization of such processes, and alternative realizations cannot be obtained in practice. An alternative approach that gives an even stronger robustness test is to fit RNNs to different trials of the same animal and examine the strategies discovered by each model fit. The most robust cognitive strategies will be identifiable across all of these fits.

Fortunately, this approach is easily implemented in our analysis pipeline since the nested cross-validation procedure used in our main analyses already requires fitting each model to different trials of the same animal (see Fig. 1c). For each model, we obtained 10 model instances from the 10 outer rounds, each trained on different trials and with varied hyperparameters. We found that the correlations among logits generated by GRU models across outer rounds are close to one, significantly surpassing correlations between logits generated by GRU models and logits by other cognitive models (see Fig. S12a-d — copied here as Fig. R6a-d —, for an example monkey and example rat). This suggests that GRU model instances consistently extract the same strategy from an individual animal's choice data, with only minor variations, demonstrating strong identifiability. Additionally, we noted an improvement in logit correlation with increases in the available training data (see Fig. S12e-f — copied here as Fig. R6e-f, comparing to Fig. S8a), suggesting the convergence of identified strategies.

The strategy convergence measure by logit correlation described above is mentioned in the revised manuscript starting from line 126:

Finally, these fitted RNNs produced highly robust and consistent predictions across model instances, suggesting that the strategies discovered by our approach are robust to variations intrinsic to empirical data (Fig. S12).

In addition to measuring the robustness of each cognitive strategy, this approach can also estimate the robustness of other features extracted from the model dynamics, including dynamics-level parameters such as preference setpoints in one-dimensional models and dynamical regression coefficients in higher-dimensional models. To do so, we first extract the parameter of interest from each of the 10 outer rounds. We then compute this parameter's mean and standard deviation (see the error bars in Fig. 3d-f, Fig. 4b, Fig. S15, Fig. S16). The minimal (sometimes invisible) error bars underscore the convergence and robustness of the extracted strategies and corresponding dynamics-level parameters. Finally, this approach can also estimate the robustness of parameters extracted from models fitted to artificial behavior (Fig. S13 — copied here as Fig. R5). Again, the minimal standard deviations suggest the convergence of extracted strategies.

The robustness of dynamics-level parameters described above is mentioned in the revised manuscript starting from line 149.

To evaluate the limits of this flexibility, we simulated the behavior of RL and Bayesian inference agents in the reversal learning and two-stage tasks and fitted both RNN and cognitive models to these synthetic data (see Supplementary Results 1.1, Fig. S4, S5, S6, S7). We found that the tiny RNNs achieved predictive performance similar to the ground-truth model that generated the behavior, with the best-performing RNN having the same dimensionality as the ground-truth model. These results suggest that RNN models can serve as a superset of classical cognitive models despite using a single architecture consistently across tasks and datasets and requiring only minimal manual engineering. Incidentally, these results also demonstrate that these cognitive strategies are identifiable and robustly recoverable (see Supplementary Results 1.1 and Fig. S11, S13), and that our training procedure successfully prevented overfitting, which can be diagnosed when the fitted model achieves higher performance on the training dataset and lower on the test dataset relative to the data-generating model.

Figure R4: **Tiny RNN models can recover the ground-truth phase portrait from simulated behavior generated by a cognitive model performing the two-stage task.** Ground-truth phase portrait (left) and RNN-recovered phase portrait (right). (a) A Bayesian inference (latent-state) model with $d = 1$. (b) A model-based model with $d = 1$. (c) A model-free (Q(1)) RL model with $d = 2$. (d) A model-based RL model (without value forgetting) with $d = 2$.

Figure R5: **The strategies extracted by tiny RNN models (fitted to artificial behavior generated by one model) are robust across iterations in the outer loop.** For each model, there are 10 model instances from the 10 iterations in the outer loop (in the nested cross-validation procedure) that are trained on different trials and with different hyperparameters. Left column: Each point is the preference setpoint extracted from the ground-truth model. Right column: Each point is the preference setpoint recovered by the GRU model fitted to the behavior generated by the ground-truth model (averaged over 10 outer rounds). Error bars are the standard deviations across outer rounds. (a-c) One-dimensional models (with ground-truth models fitted to one example rat) in the two-stage task. (a) Bayesian inference (latent-state) strategy ($d = 1$). (b) model-based strategy ($d = 1$). (c) GRU strategy ($d = 1$).

Figure R6: **The strategies extracted by tiny RNN models (fitted to animal's behavior) are robust across iterations in the outer loop.** For each model, there are 10 model instances from the 10 outer rounds (in the nested cross-validation procedure) that are trained on different trials and with different hyperparameters. Each point is the correlation coefficient between the logits provided by two model instances. Dashed lines are the average across points. The correlations of logits generated by GRU models across outer rounds are close to one, significantly higher than the the correlations between logits generated by GRU models and by other cognitive models. (a) One-dimensional GRU model and (b) two-dimensional GRU model for one example monkey in the reversal learning task. (c) One-dimensional GRU model and (d) two-dimensional GRU model for one example rat in the two-stage task. (e) The logit correlation between pairs of GRU model instances (from outer rounds) improves as the amount of available data increases, suggesting convergence of extracted strategy (showing one example monkey in the reversal learning task). (f) The logit correlation between pairs of GRU model instances (from outer rounds) improves as the amount of available data increases, suggesting convergence of extracted strategy (showing one example rat in the two-stage task).

1.3

The paper claims to reveal novel cognitive strategies, but I was not convinced that this was the case. One of the cognitive strategies revealed by the RNN looked identical to the “correlated rewards” version of Q-learning, other than small deviations in the impact of evidence for/against a chosen option at high levels of certainty. This deviation is what you’d expect if you not only learned a Q value, but built inertia by integrating recent choices, which is commonly used in cognitive models.

Thank you for this important comment. While we disagree about the reviewer’s specific example (see paragraph below on the “correlated rewards” version of Q-learning), we agree on the importance of justifying our novelty claim. In our original submission, our claim was based on the fact that we did not find prior reports of those strategies in the literature despite extensive search. However, since it is difficult to claim that something is “novel” with absolute certainty, we acknowledge the possibility that such reports may exist outside the literature we searched. Thus, prompted by the reviewer’s comment, we decided to conduct an even deeper literature search for any previous report of the strategies described in our manuscript. After this deeper literature search, we maintain our belief that the cognitive strategies we have reported are indeed qualitatively distinct from any existing model in terms of (i) mathematical formulations, (ii) model-fitting performance, and (iii) dynamical properties in logit analyses and phase portraits. Thus, we decided to maintain our claims in the revised manuscript.

We also wish to comment on the reviewer’s specific example. To the best of our knowledge, the “correlated rewards” version of Q-learning mentioned corresponds to the model-free RL strategy with one dynamical variable ($d = 1$; Fig. 3b, left). Our dynamical systems analysis suggests that this strategy is, indeed, similar to the strategy discovered by the RNN ($d = 1$; Fig. 3c), yet several critical differences can be found in their phase portraits. One of the differences that our manuscript highlights is a preference-dependent choice perseveration, a pattern of perseveration that we have not found in any prior reports. To ensure that this pattern is distinct from other similar models, we implemented two versions of model-free RL incorporating inertia (a classical implementation of choice perseveration, with formulas provided in Methods). Both of these conventional models exhibited inferior model-fitting performance compared to an RNN with the same number of dynamical variables (Fig. R7a). Additionally, the phase portraits of these models did not exhibit a separation between light blue and light red points at extreme logits, as seen in the RNN phase portraits (Fig. R7b-d). Finally, our targeted analyses suggest that the two model-free RL models augmented with inertia are qualitatively different from the pattern of preference-dependent choice perseveration discovered by the RNN (Fig. S14).

These models with inertia described above are now shown in the revised manuscript (Fig. 1e) and described in detail in the Methods section, starting from line 569:

Model-free strategy with inertia (d=2). The action values are updated as the model-free strategy (d=1). The action perseveration (inertia) is updated by:

$$\begin{aligned} X_t(a_{t-1}) &= X_{t-1}(a_{t-1}) + \alpha_{\text{pers}}(k_{\text{pers}} - X_{t-1}(a_{t-1})) \\ X_t(\bar{a}_{t-1}) &= X_{t-1}(\bar{a}_{t-1}) - \alpha_{\text{pers}}(k_{\text{pers}} + X_{t-1}(\bar{a}_{t-1})) \end{aligned} \quad (\text{R1})$$

where α_{pers} is the perseveration learning rate ($0 \leq \alpha_{\text{pers}} \leq 1$), and k_{pers} is the single-trial perseveration term, affecting the balance between action values and action perseverations.

Model-free strategy with inertia (d=3). The action values are updated as the model-free strategy (d=2). The action perseveration (inertia) is updated by the same rule in the model-free strategy with inertia (d=2).

We also present several supplementary analyses to both validate the discovered strategies and to differentiate them from classical cognitive models. These analyses further strengthened our confidence in the novelty of our discoveries. Furthermore, even if the novelty of specific cognitive strategies was not a factor, we believe that our framework offers several other advantages over traditional methods by recasting cognitive models and neural networks in the language of dynamical systems, and by representing these dynamics graphically, facilitating the discovery of cognitive strategies and circumventing the traditional cycle of model fitting, adjustment, and comparison.

1.4

In the two-stage tasks, it is most common for fitting to include mixture models that combine different model elements (for example model-based and model-free learning) as these models tend to fit behavior better – it

Figure R7: **Analysis of model-free RL models with inertia (typical choice perseveration)**. (a) Predictive performances of models fitted to one example monkey. Both model-free models with inertia are worse than the GRU models with a same number of dynamical variables. Error bars show the average SEM (standard errors first calculated across outer and inner rounds, then averaged across individuals). Logit patterns of (b) the GRU model ($d=1$), (c) model-free strategy with inertia ($d=2$), and (d) model-free strategy with inertia ($d=3$).

was not clear to me whether the best fitting RNN models for the two stage tasks were basically picking up on mixtures of the more basic strategies, or doing something unique.

Thanks for this suggestion. We implemented the mixture model ($d = 2$) (see Methods for details) and found its performance to be inferior to that of the GRU ($d = 2$) model (Fig. R8a). Moreover, behavior-feature identifiers predicted the rat's behavior as better aligned with the GRU model (embodying the "reward-induced indifference" effect) rather than with the mixture model ("MB ($d = 2$)" in Fig. S27d,e – copied here as Fig. R3d,e).

To facilitate the interpretation of these different model classes, we visualized the phase portraits for the RNN ($d = 1$, Fig. R8b) and the mixture model ($d = 2$, Fig. R8c). We also considered a one-dimensional GRU model trained to predict the choices of the two-dimensional mixture model, effectively compressing the behavior into a one-dimensional representation (Fig. R8d). We found that the mixture model could not capture the non-zero x -intercepts of the light blue points for $[A_1 S_2 R=0]$ and light red points for $[A_2 S_1 R=0]$ (Fig. R8c-d). This suggests that the traditional mixture model does not capture the RNN-discovered "reward-induced indifference" effect, highlighting a significant limitation in its ability to identify this effect.

The mixture model described above is included in the revised Fig. 1e, and in the revised Methods starting from line 609:

Model-based mixture strategy ($d=2$). This model is a mixture of the model-free strategy ($d=1$) and the model-based strategy ($d=1$). The net action values are determined by:

$$Q_t^{net}(A_i) = (1 - w)Q_t^{MF}(A_i) + wQ_t^{MB}(A_i), \quad (R2)$$

where w controls the strength of the model-based component. The action probabilities are generated via $\text{softmax}(\beta Q_t^{net}(A_1), \beta Q_t^{net}(A_2))$. $Q_t^{MF}(A_1)$ and $V_i(S_1)$ are the dynamical variables.

Figure R8: **Analysis of the mixture model (d=2).** (a) Predictive performances of models fitted to one example rat in the two-stage task. The mixture model is worse than the GRU model with a same number of dynamical variables. Error bars show the average SEM (standard errors first calculated across outer and inner rounds, then averaged across individuals). Logit patterns of (b) GRU model (d=1) fitted to the rat's behavior, (c) mixture strategy (d=2) fitted to the rat's behavior, and (d) GRU model (d=1) fitted to the behavior generated by mixture strategy (d=2).

1.5

Overall, it seems that the current manuscript presents a bunch of possible revealed strategies, without carefully examining or validating any of them, leaving it a bit unclear whether they really deviate qualitatively from cognitive models fit to the same sorts of data.

Thank you for this important feedback. The revised manuscript prioritizes quality over quantity: we now focus on a smaller set of strategies and devote more attention to each. We also present the analyses described in the answers above, including model recovery, parameter recovery, model validation, and head-to-head model comparisons. Any cognitive strategy or data analysis we could not fit into the main text is now presented in the supplementary materials.

1.6

The comparison of cognitive models to RNNs in terms of d seems pretty misleading. Clearly d means something very different in a cognitive model than an RNN. How many variables do I need to pass information about the past (ie. current state) into the next timestep? Probably just one, if I can reversibly compress the information – that is to say, if I need to store three binary variables, I could store them all in a single continuous variable (0-0.125 = 000, 0.125-2.5 = 001, and so on). For a small amount of precision loss, I could do the same thing with continuous variables, and the RNN is well posed to do this – it has gating parameters that could nonlinearly “unpack” the previous timestep in ways that are not afforded to standard cognitive models. For this reason, I think comparing d for cognitive and computational models is like comparing apples and oranges. I am not sure that I have a better alternative measure of complexity, but I do think that the authors should either come up with one, or refrain from comparing d across the model sets and interpreting it as a meaningful comparison.

Thank you for raising this point, which prompted us to perform a substantial amount of theoretical research. In sum, the reviewer is only partially correct. While one model can pack several *discrete* variables into a single variable and learn its dynamics, the same is not true about *continuous* variables – i.e., it is *not* compatible with learnable dynamics if one packs multiple continuous variables into one continuous variable without significant loss of precision. Intuitively, well-behaved functions (e.g., smooth functions) cannot unpack multiple continuous variables from a single continuous variable

without exhibiting pathological behavior. Below we present two statements with corresponding proofs by contradiction (Supplementary Discussion 2.3) to support this claim.

Statement 1. *A one-dimensional continuous RNN $h_{t+1} = f(h_t; \text{Input})$ with a linear readout, cannot accurately represent an irreducible two-dimensional dynamical system without significant precision loss for learning dynamics.*

Proof. Consider a two-dimensional RL model where the value of the chosen actions depend on reward prediction error, and the value of unchosen action remains unchanged (Fig. R9a). The model’s initial state is $(Q_L, Q_R) = (0, 0)$, and the first inputs alternate between $[A_L; R=1]$ and $[A_R; R=1]$ five times, for a total of 10 trials. Given that the readout from the one-dimensional hidden state h_t of the RNN to the output layer is linear, h_t can *only* represent a linear function of the logit L_t , i.e., $h_t = aL_t + b$, where a and b are coefficients and $L_t = \log[\Pr(A_L)/\Pr(A_R)]$. At the red and blue points explored by the RL model, since $Q_L = Q_R$, $L_t = 0$ (indicating equal probability), resulting in $h_t = b$. This implies that red and blue points should be represented approximately at b in the RNN ($h_{red} \approx h_{blue} \approx b$). By continuity, we have $f(h_{red}; A_L R = 1) \approx f(h_{blue}; A_L R = 1) \approx f(b; A_L R = 1)$. However, $h_{orange} = f(h_{red}; A_L R = 1)$, which strongly favors Q_L , should differ from $h_{green} = f(h_{blue}; A_L R = 1)$, which shows almost equal preference. This disparity contradicts the continuity of the function f . □

Statement 2. *A one-dimensional continuous RNN $h_{t+1} = f(h_t; \text{Input})$ with an arbitrarily nonlinear readout cannot accurately represent an irreducible two-dimensional dynamical system without significant precision loss for learning dynamics.*

Proof. Given the unconstrained nature of the readout, we assume that the one-dimensional RNN can learn an extremely nonlinear representation of two-dimensional state-space of the RL model. Generally, according to Netto’s theorem, continuous bijections between smooth manifolds preserve the dimension², meaning that there does not exist a continuous bijective mapping between two smooth manifolds of different dimensions (e.g., one-dimensional state space and a two-dimensional state space). However, by relaxing the conditions, we can consider space-filling curves, which are surjective continuous functions from one-dimensional spaces to two-dimensional spaces (e.g., a Hilbert curve). True space-filling curves are non-differentiable everywhere, and therefore, cannot be learned by RNNs. Consequently, we consider approximations to space-filling curves through finite iterations (thus only non-differentiable at finite points). For an illustration of the Hilbert curve in two-dimensional state space, see its third iteration in Fig. R9b and four iteration in Fig. R9c. We define the 2D-to-1D approximation function G , such that each two-dimensional point is mapped to its nearest position on the Hilbert curve. This method allows the approximation of two-dimensional space by a one-dimensional curve to achieve higher accuracy in higher-order iterations.

Consider the red and blue points, which are close in the two-dimensional space and similarly mapped to proximate positions on the Hilbert curve ($G(red) \approx G(blue)$). After a single trial of $[A_L; R=1]$, the red point transitions to the orange point $G(orange) = f(G(red))$ and the blue point transitions to the green point $G(green) = f(G(blue))$. According to the continuity of f , the representations of orange and blue points should be similar ($G(orange) \approx G(green)$). However, paradoxically, these points should occupy positions on the Hilbert curve that are far apart, as the orange and green points fall into different grid sections, thereby being mapped to disparate regions of the Hilbert curve ($G(orange)$ and $G(green)$). This indicates that, if the loss incurred from representing two-dimensional space by the one-dimensional curve (G) is minimized to be very small (e.g., through higher iterations), then the loss in accurately representing two-dimensional RL dynamics by one-dimensional RNN dynamics f will conversely be significantly large. □

The statements above prove that two continuous variables cannot be packed into a single continuous variable without incurring significant loss. The issue lies not in the representation of states, but in the pathological dynamics that would be necessitated by such a representation. By extension, a d -dimensional dynamical system with continuous state variables cannot be characterized by less than d dynamical variables. The relevance of this dimensionality is evidence in the fact that each of the d variables that characterize a dynamical system describe one function mapping d variables in trial t to d variables in trial $t + 1$. Indeed, according to one of the most classical textbooks on dynamical systems, the dimensionality of the system is one of its most relevant characteristics, and systems of equal dimensionality can be directly compared to one another³. Clearly, this dimensionality would lose its meaning if a dynamical system could be compressed into fewer dynamical variables without substantial loss.

Finally, we note that both cognitive models and RNNs are systems whose states change in discrete time steps. In both cases, a (possibly multivariate) function describes what future states follow from the current state, so that both meet the definition of a discrete dynamical system. Like any dynamical system, the number of functions describing the state of the system and its evolution is equal to its dimensionality, and this state cannot be compressed into fewer variables without substantial loss.

Since two generic dynamical systems of equal dimensionality can be directly compared, a cognitive model and an RNN of equal dimensionality are also directly comparable. We contend that the dimensionality is a fundamental aspect of cognitive models frequently overlooked by the research community.

Within neuroscience, the dimensionality of a dynamical system is fundamental for understanding neural population dynamics^{4,5}. For a recorded sample of N neurons, the instantaneous population firing rates can be conceptualized as a point within the entire state space (with the ambient dimensionality of N). Task-relevant computations typically rely on structured neural activity restricted to one or several manifolds^{5,6}. Intrinsic dimensionality is defined as the minimal number of continuous variables required to parameterize the manifold. Consequently, studies on intrinsic dimensions are prevalent in neuroscience. In our study, we utilize d_* to estimate the upper bound of intrinsic dimensionality.

In two recent theoretical studies, the dimensionality of a dynamical system have been used also to describe behavior^{7,8}. These studies define the dimensionality of behavior as the number of functions of the agent's past one needs to measure in order to accurately predict its future as well as possible. Our application of dimensionality in RNNs and cognitive models aligns closely with this definition.

The meaning of the dynamical variables described above is mentioned in the revised manuscript starting from line 133

The dimensionality of a given behavior (d_*) is defined as the minimal number of functions of the past required to optimize the predictability of future behavior^{7,8} (also see Supplementary Discussion 2.3).

Figure R9: **Illustration of two-dimensional state space of the model-free RL model, defined by the action values Q_L and Q_R .** Upon receiving $[A_L R=1]$, the red-point state transits to the orange-point state, while the blue-point state transits to the green-point state. (a) In the one-dimensional RNN, the readout from the hidden state to the output layer (before softmax) is linear. (b-c) In the one-dimensional RNN, the readout from the hidden state to the output layer (before softmax) can take an arbitrarily nonlinear form. Blue curves represent the (b) third and (c) fourth iterations of the Hilbert curves, representing the one-dimensional hidden state h of the RNN.

1.7

On a related note, it is shown that the RNNs outperform alternative strategies in terms of log-likelihoods, but the presentation here needs much more context. In particular, it is not clear how scientifically (or statistically, actually) significant the differences in log-likelihood are. The reader should not have to trust the authors that

a difference of 0.05 or 0.1 points in log-likelihood fit are either descriptively or mechanistically meaningful, though of course it is possible that they are. Given how heavily optimized cognitive models are (via the scientific process) our prior expectations should be that they describe the scenarios they were developed for extremely well. As such, showing that the RNNs give superior descriptions requires establishing that there was previously meaningful room for improvement, and that the RNNs are capturing important components of behavior being missed by the cognitive models. Some of the behavioral analyses start to go in this direction, but it's far from firmly established.

Thank you for raising this important point about the statistical and scientific significance of our results. We agree that more context is needed to interpret the log-likelihood differences properly. To contextualize the statistical significance of our results, we have added error bars to all figures showing log-likelihood comparisons in the main text. These error bars represent standard errors calculated across subjects (Fig. 1, 2), providing a clear visual indicator of the reliability of the differences between models. To contextualize the scientific significance of our results, we have calculated the percentage improvement in predictive accuracy (see Fig. S10 – copied here as Fig. R2). Across eight datasets and six tasks, we found that tiny RNNs correctly predict significantly more trials than the cognitive models with the same number of dynamical variables (except for the monkeys performing the reversal learning task where the predictive accuracies started to saturate).

The predictive accuracy described above is mentioned in the revised manuscript starting from line 115:

We found that very small RNNs predicted animals' choices more accurately than classical cognitive models and all of their variants across all tested datasets (Fig. 1e for three datasets and Fig. S2 for two additional datasets, evidenced by the fact that the lowest (best) scores in each plot are achieved by an RNN; also see Fig. S10a-c for test accuracies).

We also agree with the reviewer that cognitive models have been heavily optimized through the scientific process. However, our results also suggest that there was indeed meaningful room for improvement. For example, our approach revealed that models instantiating the discovered strategies predicted choices more accurately than alternative models, and these strategies were further validated by model recovery, parameter recovery, and model validation, as described in our responses to 1.1 and 1.2. In some cases, we were able to find specific behavioral signatures predicted by the RNNs and not by any cognitive models, such as a pattern of preference-dependent choice perseveration. More generally, RNNs outperformed even normative models, including a task-optimized Bayesian model. Finally, we note that other recently published papers have also found meaningful room for improvement over classical cognitive models (e.g., Ger et al 2024⁹). Overall, these results suggest that improvements in prediction accuracy are indeed possible with more flexible models such as RNNs, even in comparison to highly optimized cognitive models. A simple and intuitive argument is that, unlike most handcrafted cognitive models, RNNs are capable of describing the suboptimal and idiosyncratic behavior that individual animals often display.

1.8

Finally, while the authors note that a major issue with RNNs has been with interpretability, another issue with RNNs has been that of robustness and generalization. Getting RNNs to fit data is often trivial, but getting them to generalize to an untrained condition is often much more difficult, and perhaps a better test of whether the model is providing information that is theoretically useful (ie. revealing underlying computational principles) versus curve fitting to the training data. I had a bit of difficulty understanding exactly how the training/testing was done but my understanding is that it did not include any test of model generalization. I think supporting the core claim, that tiny RNNs can reveal cognitive strategies, would require showing at least some level of generalization beyond the training data. Perhaps an alternative would be to frame the claim instead as RNNs providing a low dimensional description of the data, rather than revealing the underlying cognitive strategies, but obviously this would be a slightly less impressive conclusion. More generally, there was not sufficient information about the holdout/cross validation procedure to fully evaluate it and the authors should be clear on exactly how models were trained/tested/evaluated.

We agree with the reviewer's that examining robustness and generalization in RNN models is crucial, and we apologize if our initial description of the cross-validation procedure was unclear. To guarantee that our models generalize to untrained conditions, our approach employed a nested cross-validation procedure in all reported analyses, unless specified otherwise. Our approach for training, testing, and validation is now described in detail in the revised manuscript, including a technical description in the Methods section and a visual explanation in Fig. 1c. In the technical description, we describe how we handle the temporal structure of the data, ensuring that our test sets represent true out-of-sample predictions. This approach ensures that the data used for testing the models is never used for training or validation.

The nested cross-validation procedure described above is mentioned in the revised manuscript starting from line 105:

Given the substantial difference in the number of free parameters between RNNs (e.g, 40-80 for 1-2 unit RNNs) and cognitive models (e.g, 2-10 parameters; Fig. S1), we compared models via nested cross-validation, a procedure that uses different trials to train, validate, and evaluate the models (Fig. 1c; see Methods for why AIC or BIC are not appropriate in this setting).

Our cross-validation approach also enabled tests of robustness, where we compared the information extracted from models fitted to different blocks of trials. As described in 1.2, these analyses suggest that the models were robust and not due to chance findings of particularly good model fits.

Reviewer 2

Summary

This article presents a novel set of tools to study the mechanisms that support behavior in non-human animals. The motivation to develop this tool is the recognition that behavioral modeling is an incremental process of discovery, with no clear signal of when a satisfactory model is obtained. The hope of developing these tools is to 1) automate discovery, and 2) identify a ceiling of explainability for a given data set. The novel set of tools consists in two main steps: 1) using tiny (1-4 GRUs) RNNs to build a predictive model of animals' behavior; 2) using dynamical systems approaches to interpret the trained RNNs. The first step also provides an estimation of the dimensionality of behavior. The authors show convincingly in three specific data-sets that their approach works well – i.e. is more predictive than classic families of models on these tasks, and may enable discovery of new mechanisms.

This is an exciting novel approach, tackling an important problem. The paper is didactic and thorough. It has the benefit of managing individuals, rather than groups, as has been done previously, and enabling a comparison of different types of models on fair grounding. Nonetheless, it is unlikely that this approach will generalize well to a broad class of behavior, and the article vastly oversells its promise on multiple points. I note that the authors anticipated and attempted to address some of my concerns, but I do not think this was successful. I elaborate below.

We are grateful for the constructive feedback provided by the reviewer. We have thoroughly addressed each point raised. Detailed responses to each point follow below.

2.1

First, the article should vastly tone down claims on multiple aspects. The use of “automatic” in the title and abstract is *not* warranted. The RNNs are indeed trained automatically, but once they are trained, researchers only have a tool to predict behavior, they absolutely have not discovered new mechanisms. This step fully relies on the second set of tools based on dynamical systems, and is far from automatic or accessible (see point 5).

We agree that some readers may perceive our use of the term “automatic” as an overstatement, so we removed it from the title, abstract, and main text in the revised manuscript. In our original submission, our intention was to describe the process of “*capturing* cognitive strategies from behavioral data” as automatic, in contrast to the conventional, non-automatic approach of cycling between model adjustment, fitting, and comparison, which we view as time-consuming and error-prone. However, since our approach does not automate the process of *interpretation*, as needed for discovering cognitive strategies, we no longer claim that the process of discovery is automatic.

In the revised manuscript, instead of claiming “automatic discovery,” we now describe our approach as “a systematic framework for identifying and interpreting cognitive strategies.” This better reflects the semi-automated nature of our method, where RNNs capture behavioral patterns, but human interpretation is still crucial. We believe these changes provide a more accurate and balanced presentation of our work, addressing the reviewer’s concerns about overstatement while still conveying the significant advantages and novel insights offered by our approach.

2.2

Second, the authors emphasize *tiny* RNNs, and say *1-2* units in the abstract and intro, even though one of the tasks necessitates 4. This is not a quibble, as the difficulty of the second step (interpretation) is very non-linear in the number of units. Furthermore, it is very likely that other tasks will necessitate more units (see next point).

We appreciate the reviewer’s attention to detail and agree that our initial emphasis on 1-2 units in the abstract and introduction did not fully capture the full range of dimensionalities across tasks. We have updated the abstract and introduction to more accurately reflect the range of dimensionalities used across all tasks. The revised text now states starting from line 49:

In contrast to previous approaches, however, our framework uses very small RNNs, often composed of just 1-4 units, which greatly facilitates their interpretation.

We also agree with the reviewer that larger RNNs can outperform smaller RNNs (e.g., those with 1-4 units) in tasks of higher complexity. In the revised manuscript, we have added a paragraph in the Results section discussing the relationship between task complexity and the number of units required. This includes an explanation of why some tasks (like the transition-reversal two-stage task) may be better fit with larger RNNs, and how this relates to the underlying complexity of the behavior being modeled. Our revision also considers the issue of dimensionality more extensively, including three additional datasets with more complex tasks and a dynamic regression framework for analyzing RNNs of higher dimensionality, as described in subsequent responses. Issues related to the scalability of our approach to more complex tasks and higher-dimensional models are also included in the Discussion section starting from line 358:

While effective across multiple tasks, our approach may face challenges in scaling to more complex tasks. Future work should thus explore alternative architectures, training methods, or interpretability techniques to address these limitations.

2.3

Scaling to other tasks. While the authors select three tasks of seemingly increasing complexity to attempt to show generalizability of the method, the three tasks share two important features that likely make behavior dimensionality low: the fact that choices are binary, and the fact that there is only one choice state. If either of those is untrue, as is the case even in non-human animal tasks (e.g. Rikhye et al 2018; Sweis et al 2018, Science) and frequently in human cognitive tasks (e.g. probabilistic selection task, Frank et al 2004), the dimensionality will have to be significantly higher. To convincingly argue that this tool is generalizable, it would have been better to use tasks that increase not in design complexity, but in complexity along those dimensions. There is a difference between the claim that low-D GRU nets don’t overly constrain solutions on this task, and the claim that they won’t in general for cognitive tasks.

Thank you for raising this extremely important point about the generalizability of our method to more complex tasks. We agree that demonstrating the applicability of our approach to tasks with higher dimensionality and complexity is crucial for establishing its broader utility. Accordingly, one of the most important components of our revision is the inclusion of more complex tasks and of a new technique for analyzing models of higher dimensionality, as described subsequently.

To extend our method beyond tasks with “binary choice” and “one choice state” to more complex scenarios, we have added three new datasets involving human subjects performing reward-learning tasks of greater complexity: (i) a three-armed reversal learning task featuring three actions; (ii) a four-armed drifting bandit task featuring four actions and continuous rewards; (iii) the original two-stage task featuring six actions across three choice states and two decision stages. These tasks are described in Fig. 2 – copied as Fig. R10 – and detailed in the Methods section. Our findings from these tasks are described in a new section in the Results titled “Predicting choices with tiny RNNs for humans tasks”. This section demonstrates that our approach can effectively model and interpret behavior in tasks with more than two actions, multiple choice states, and continuous rewards. This new section in Results starts from line 181:

To expand the applicability of our approach, we next examined how tiny RNNs perform on human decision-making tasks, which typically involve fewer trials per subject and use slightly more complex designs than animal studies. We applied our method to analyze three tasks, representing a range of experimental paradigms commonly used in cognitive neuroscience research: a three-armed reversal learning task (three actions; 160 trials per subject), a four-armed drifting bandit task (four actions and continuous rewards; 150 trials per subject), and the original two-stage task (six actions and three choice states; 200 trials per subject) (Fig. 2c). Given the limited per-subject data, we used knowledge distillation and implemented an interspersed split protocol to train and evaluate the models (see Methods; also see the cross-subject split protocol in Fig. S39). As before, we compared tiny RNNs to over 10 established cognitive models for these tasks.

To address the challenges of interpreting higher dimensional models, we have also developed and implemented a new analysis method based on dynamical regression, designed to analyze vector fields in higher-dimensional models ($d \geq 2$). This

Figure R10: **Model performance on using knowledge distillation.** (a) Knowledge distillation framework: A large teacher network (TN) is trained on data from multiple subjects; the subject ID corresponding to each datapoint, provided as input, enables subject-specific embeddings e_s to be learned. A tiny student network (SN) is then trained on single-subject data to match TN's output probabilities. (b) Model predictive performance for a representative mouse in the transition-reversal two-stage task, across varying dataset sizes. Student RNN outperforms the best model-free RL model for all dataset sizes. Note different x-axis scales for < 3000 and > 3000 trials. (c) Human tasks structures. *Left:* Three-armed reversal learning: Subjects choose between actions A_1 - A_3 , each associated with a reward probability that changes over time. *Center:* Four-armed drifting bandit: Subjects choose between actions A_1 - A_4 , with each associated with a 0-100 reward that fluctuates over time. *Right:* Original two-stage task: Subjects first choose action A_1 or A_2 at the choice state, transitioning into one of two second-stage states, S_1 or S_2 . Subjects then choose action B_1/B_2 or C_1/C_2 , each probabilistically yielding a reward. Reward probabilities change over time. (d) Model performance (cross-validated trial-averaged negative log-likelihood; lower is better) vs. number of dynamical variables d , averaged over subjects using interspersed split protocol. Tiny RNNs outperform classical models in all tasks. Identical markers within a plot represent different variants of a model class. Error bars: SEM across individuals. *Left:* Three-armed reversal learning. *Center:* Four-armed drifting bandit. *Right:* Original two-stage.

method, described in a new section of the Results (“Interpreting and comparing multi-dimensional models”), is particularly applicable to complex tasks with more actions, more choice states, and continuous rewards. Fig. 4 – copied here as Fig. R12– illustrates this analysis.

To address the challenges of limited per-subject data in human studies, we have introduced two new training protocols: (i) an interspersed split protocol (results in Fig. 2 – copied as Fig. R10), and (ii) a cross-subject split protocol (results in Fig. S39 – copied as Fig. R11), described in detail in the Methods section. These protocols allow us to train, validate, and test models effectively even when each subject contributes only a few trials.

Figure R11: **Application of the cross-subject split protocol to human datasets.** Predictive performances (trial-averaged negative log-likelihood; lower is better) in three tasks (one per panel). Performances are displayed as a function of the number of dynamical variables d , plotted on a log scale along the x-axis. The reported performance for each student RNN model and cognitive model is the average over all subjects on the unseen subjects’ test trials using the cross-subject split protocol. Error bars show the SEM across individuals. Each dynamical variable in the GRU model family corresponds to one network unit; in cognitive models, dynamical variables can correspond to action values, state values, multi-trial choice perseveration, learned state-transition probabilities, among others. Each model family may include multiple model variants, leading to two or more identical markers for a given d . (a) Three-armed reversal learning task. (b) Four-armed drifting bandit task. (c) Original two-stage task.

Finally, we have expanded the Discussion section to address the scalability of our approach more explicitly. We discuss both the successes and challenges encountered when applying our method to these more complex tasks, and outline potential future directions for further improving the method’s applicability to high-dimensional behaviors.

2.4

The authors attempt to make the case that the method will be applicable to humans (where many fewer data points are available for training) via a knowledge distillation framework. The approach is very interesting, and might potentially work, however, the treatment is much too superficial to be convincing. For example, it is only tested in one non-human animal with the most data, and is not tested with actual human data, which is likely to be much higher dimensional and have very different data sizes/statistics. It might be better left for supplements or follow up work where it can actually be thoroughly tested. As it is, with the emphasis on “cognitive modeling” rather than “behavioral”, the readers really get the impression that this is applicable to complex cognitive tasks and humans (despite the authors honest stating of “animal” in the abstract); and I think this is highly misleading at this point in the tool validation.

We appreciate the reviewer’s critical assessment of our knowledge distillation framework and its applicability to human studies. We agree that our initial treatment of this approach was too superficial. To comprehensively address these concerns, the revised manuscript applies our knowledge distillation framework to three new datasets of human behavior, each featuring significantly fewer trials per subject than typical animal experiments: (i) A three-armed reversal learning task (160 trials per subject); (ii) A four-armed drifting bandit task (150 trials per subject); (iii) The original two-stage task (200 trials per subject). These results are now presented in a new main figure (Fig. 2c-d – copied as Fig. R10c-d) and described in detail in a new Results section titled ‘Predicting choices with tiny RNNs for humans tasks’. We also demonstrate how our dynamical systems approach can be applied to interpret the tiny RNNs trained on human data, revealing novel insights into human decision-making strategies. These findings are presented in Fig. 4c-e and described in a new section of the Results (“Interpreting and comparing multi-dimensional models”).

Additionally, we have significantly expanded our Methods section to provide a detailed description of how we adapted our knowledge distillation approach for human data. This includes: (a) An explanation of the interspersed split protocol, which addresses the challenges of limited per-subject data; (b) A description of the cross-subject split protocol, which leverages data from multiple participants to perform zero-shot individual-level predictions for unseen subjects; (c) A discussion of how we handle the higher dimensionality often present in human tasks.

Finally, we have added a paragraph in the Discussion that candidly addresses the current limitations of applying our approach to human data and outlines future directions for improvement. This includes a discussion of how our method might be extended to even more complex cognitive tasks and datasets, starting from line 358:

While effective across multiple tasks, our approach may face challenges in scaling to more complex tasks. Future work should thus explore alternative architectures, training methods, or interpretability techniques to address these limitations.

2.5

Second set of tools – dynamical systems. First, as stated above, this component is highly non automatic, and really pushes back the problem of discovery to another complex interpretation problem; it is not obvious that in many cases this problem will be easier than the one it is trying to replace. Furthermore, the tools offered only apply to specific cases. For example, the logit analysis requires only two choices. The authors acknowledge this late in the paper, and say that it doesn't matter as one can always do one choice vs. all others – but are insights likely to be easily discovered that way? It is known that multiple choices impact decision making in non-trivial ways, that can be discovered by considering errors, for example; something that would not be easily handled by this approach. Same, as soon as the dimensionality is $d > 1$, the tools become even less tractable, in a way that is highly non-linear with task complexity (e.g. need to plot per possible task event, which could increase multiplicatively with number of states, choices, outcomes, etc.)

We appreciate the reviewer's insightful comments on the challenges associated with our dynamical systems approach, particularly for higher-dimensional models and more complex tasks. We agree that our initial presentation did not adequately address these issues. Since the revised manuscript now includes three additional tasks of higher complexity, we also describe an approach for interpreting the models fitted to data from these more complex tasks. Most notably, we have developed and implemented a new "dynamical regression" method specifically designed to analyze and interpret models with more than two dynamical variables ($d \geq 2$). By linearizing a model's vector fields, this method provides a first-order approximation of the dynamics for any given input, allowing us to extract meaningful insights even from higher-dimensional models. We describe this method in a new section of the Results titled "Interpreting and comparing multi-dimensional models", Fig. 4b-e – copied here as Fig. R12b-e. In the Supplementary Materials, we also provide details about the application of dynamical regression to the three novel tasks: Dynamical regression analysis of the three-armed reversal learning task (Section 1.4), Dynamical regression analysis of the four-armed drifting bandit task (Section 1.5), and Dynamical regression analysis of the original two-stage task (Section 1.6). These analyses show that our approach can yield interpretable insights even for tasks with more than two actions, continuous rewards, and multiple choice states.

We have expanded our Discussion to more explicitly address the current limitations of our dynamical systems approach, particularly for very high-dimensional tasks. We outline potential future directions for extending these methods to even more complex scenarios.

2.6

Dynamical systems:

- a validity of constant input regimes. In a real experiment participants may not go anywhere near the fixed points, so (a) networks will be doing out of sample generalization which is notoriously finicky for neural networks and (b) the important dynamical considerations may be obscured by effectively irrelevant fixed point topologies.
- b Is the constant input of a single stimulus a realistic regime for any of the models? The focus on asymptotic behavior, while understandable from an analytic approach, constrains little when the system is rarely at rest dynamics can be vastly different to those implied under the static 1 input regime (e.g. oscillations in 1d system)

We appreciate the reviewer's insightful comments on the validity and relevance of our dynamical systems analysis, particularly regarding constant input regimes and asymptotic behavior. We agree that these are important considerations that warrant further discussion and analysis.

Figure R12: Dynamical systems analyses for interpretation and comparison of multi-dimensional models (a) Vector field analysis of two-dimensional models fitted to the choices of one monkey in the reversal learning task. Each panel illustrates the effect of one input on the state variables (axes). Black arrows: flow lines indicating state changes per trial; white crosses: attractor states; dashed lines: indifference states; orange arrows: readout vectors; background color: dynamics speed (purple: slow; green: medium; yellow: fast). *Left*: Model-free RL model; axis-aligned arrows indicate that only the value of the chosen action is updated in each trial, with values converging to the reward magnitude. *Right*: Two-unit RNN. *Top*: in unrewarded trials ($R = 0$), convergence of arrows to diagonal line suggest a drift-to-the-other pattern. **(b)** Dynamical regression analysis for 2D model-free RL (left) and two-unit RNN (right). P_i , ΔP_i : preference and preference change for action A_i . Regression coefficients describe how each action preference changes in each trial as a function of its own current value (β_{P_i}), of the current preference for the other action (β_{P_j}), and independently of one's current preferences (baseline β_0). Points represent mean coefficients across outer rounds and error bars represent standard deviations across outer rounds. Circled coefficients in RNN model indicate drift-to-the-other pattern. **(c-e)** Dynamical regression analysis for a model-free RL model and an RNN with equal dimensionality $d > 2$, fitted to human behavior. Violin plots show distributions over subject-level coefficients. **(c)** Three-dimensional models fitted to data from three-armed reversal learning task. P_i , ΔP_i : preference and preference change for action A_i . β_0 is the constant term in the linear regression. **(d)** Four-dimensional models fitted to data from four-armed drifting bandit task. P_i , ΔP_i : preference and preference change for action A_i . β_R is the coefficient for the continuously-valued reward in the linear regression. **(e)** Three-dimensional models fitted to data from original two-stage task. L_i , ΔL_i : logit and logit change for each choice state (L_1 : first-stage, L_2 , L_3 : second-stage states).

In this work, we introduce the dynamical system perspective to compare models in a unified way. To deepen our analysis of these dynamical systems, it is essential to extract key features from their dynamics. We choose to focus on sets of attractors as they offer a concise summary of each model and effectively distinguish between models (Fig. 3). Importantly, for one-dimensional model-free and model-based RL models, the slopes (corresponding to negative learning rates) and x-intercepts (corresponding to attractors) for each trial condition *fully* characterize the models' behavior. Thus, these x-intercepts (attractors) capture crucial properties of these systems. When needed, we can also examine other dynamical system properties, such as the shape of logit patterns, or coefficients in the dynamical regression method (see below). For instance, non-monotonic logit changes in the Bayesian inference models are indicative of the Bayesian strategy. It is well-recognized that conventional model analyses fail to identify behavioral signatures that distinguish between Bayesian inference and model-based behaviors in the two-stage task¹. Conversely, our dynamical systems approach uniquely distinguishes Bayesian inference and model-based models based on logit patterns (e.g., see Fig. S11 — copied as Fig. R4—, Fig. 5) or their fixed points (e.g., Fig. 3e).

We agree that the attractor of a given input specifies the asymptotic behavior under the constant repeated input. However, beyond just *asymptotic* behavior, the attractor (together with its basin) also dictates the *instant* direction of state updates from any given system states upon receiving an input I . For example, in one-dimensional systems studied in the manuscript (all with one stable fixed point), the *one-step* update of system state $L(t)$ is similar to a “prediction error” with a variable learning rate (see “Effective learning rate and slope” in Methods), where $\Delta L \propto L_I^* - L(t)$, with L_I^* representing the logit attractor for input I . We have revised the term from “asymptotic preference” to “preference setpoint” to better emphasize this interpretative aspect. Additionally, the areas near fixed points for a given input can be frequently visited. For example, in a one-dimensional model-free RL model, the fixed point for [A1 R=0] is $L_I^* = 0$ and the vicinity of $L = 0$ can be reached in just one trial of input [A1 R=1] or [A2 R=1], depending on the prior system state.

The instant property of preference setpoints described above is mentioned in the revised manuscript starting from line 252

A preference setpoint u_I for input I represents an agent's normalized, asymptotic preference for one action over another after repeated exposure to input I (i.e., $u_I = L_I^* / \max_I L_I^*$). Beyond representing long-term behavior, preference setpoints indicate the *instantaneous* direction of change in the system's state when presented with input I .

Although the vicinity of fixed points (or line attractors) can be frequently visited, the dynamics around fixed points are slow. The slow dynamics result in small updates to system state upon receiving the input, and thereby cause slight changes to the output action probabilities. On the other hand, given a finite amount of data, the output probability can only be inferred with a finite precision. Therefore, slow-dynamics regimes are less accurate (relative to the precision) compared to fast-dynamics regimes. For instance, the directions of arrows near the approximate line attractor in Fig. 4b (left) — copied as Fig. R12 — are likely less constrained by the loss. To address this issue, we introduced the dynamical regression method (see 2.5), effectively providing a first-order global approximation of the one-step dynamics for a given input, rather than focusing solely on local dynamics around fixed points. Importantly, this method provides a *complete* descriptions for linear models (including model-free RL models). This method has discovered numerous insights about human behavior (Supplementary Results 1.4, 1.5, 1.6), none of which was revealed by previous works analyzing neural networks trained on same datasets^{10,11}. Incorporating these insights into cognitive models leads to improved performance (Supplementary Results 1.4, 1.5, 1.6).

In one-dimensional models specified by the logit $L(t)$, the one-step dynamics for a given input can be linearly approximated as $\Delta L \sim \beta_0 + \beta_L L$, where β_0 (after normalization) is similar to the preference setpoint, and $-\beta_L$ is similar to learning rates in RL models. Indeed, β_0 (after normalization) yielded results align closely with the preference setpoints extracted from attractors, as reported in Fig. 3 d-f). Additionally, $\beta_L \approx -1$ is an indication of oscillations in one-dimensional systems. Although such oscillations were not detected in any dataset presented in this manuscript, they have been observed in another ongoing study involving a human decision-making dataset.

Minor comments:

2.7

Dynamical variables = units in RNN, but not defined in cognitive models in the main text

Thank you for the suggestion. In the main text and figures of the revised manuscript, we make sure to define dynamical variables for both RNNs and cognitive models.

The definition of dynamical variables is mentioned in the revised manuscript starting from line 91:

In the two RL model families, dynamical variables represent action values updated via RL algorithms: model-free RL updates these values directly from reward input, and model-based RL updates them indirectly through a

state-transition model. In the Bayesian inference family, dynamical variables represent belief states updated via Bayesian inference.

2.8

Fig. 2-3 and others: likelihoods are plotted as averages across folds and individuals. It would be important to get a sense of uncertainty, either by showing individuals, or at minimum by including distributions/errorbars.

A great suggestion. We have added error bars for the log likelihoods in all of the relevant figures.

2.9

Fig. 4. Measures RNNs' ability to predict behavior. It is known that a model can predict well without generating behavior well (Palminteri, Wyart, Koechlin 2017). The authors visualize some success for GRUs in figure 1, but do not quantify this elsewhere.

Thank you for raising the important distinction between prediction and generation to our attention. We agree that the ability to generate realistic behavior is a crucial aspect of model evaluation that we did not adequately address in our original manuscript. In the revised manuscript, as well as in this document's responses to questions 1.1 and 1.2, we report the results of several analyses aimed at assessing a model's ability to generate realistic behavior, including model recovery and parameter recovery. A particularly powerful approach that we proposed in this category performs model validation via a behavior-feature identifier. This approach demonstrates that RNNs can generate behaviors with statistics that aligns more closely with the actual subjects' behavior than with behavior predicted by the best classical models. For further details, please refer to our responses to reviewer comments 1.1 and 1.2.

2.10

Have authors considered looking into what optimal behavior dimensionality is in each task, and seeing how this compares to RNNs? I.e. are the 1, 2, 4 found for the three task due to the fact that this is the dimensionality needed to solve the tasks?

Thank you for this interesting suggestion. It would indeed be interesting to compare the dimensionality of optimal behavior with the dimensionality of observed behavior.

Determining the dimensionality of optimal behavior in the three studied animal tasks requires estimating the probability of each latent state and updating these belief states via Bayesian inference. The dimensionality of optimal behavior in each task, therefore, depends directly on the number of latent states in that task. For example, in a standard reversal learning task, one latent state (X_1) corresponds to a higher reward probability for A_1 , and the other latent state (X_2) corresponds to a higher reward probability for A_2 , for a total of two latent states. Since the two latent states are mutually exclusive, $P(X_2) = 1 - P(X_1)$, and thus the agent's belief over latent states can be fully characterized by a single probability. Similarly, a standard two-stage task has two latent states corresponding to a higher reward probability for S_1 and S_2 , and the agent's belief over latent states can also be characterized by a single probability. Finally, the transition-reversal two-stage task has three latent states representing high (H), low (L), and medium (M) reward probabilities, and two latent states (A and B) representing each configuration of the transition probabilities. Since these states combine factorially, the task's unique latent states are HA, HB, MA, MB, LA, LB, for a total of six latent states. The agent's belief over latent states will thus require five probabilities, with $P(X_j) = 1 - \sum_{i \neq j} P(X_i)$. In sum, optimal behavior in these tasks have dimensionality $d = 1$, $d = 1$, and $d = 5$, respectively. We note that, however, in concrete tasks other than the standard versions above, the dimensionality of the optimal behavior is much higher, and even not analytically tractable, because these tasks incorporate additional task designs, such as the task-performance-based adaptive block transitions in the two-stage and transition-reversal two-stage tasks^{12,13}.

Determining the dimensionality of observed behavior, on the other hand, requires fitting RNNs of different sizes and determining the number of units d_* for which the RNN's performance is no worse than the best performing model. Note that this number can vary across individuals, as individual animals may require varying numbers of variables to accurately describe the behavior (1-4 variables for each animal in each task). Examining the dimensionality of behavior for individual animals in these tasks, we found that observed behavior has dimensionality $1 \leq d \leq 2$, $1 \leq d \leq 3$, and $1 \leq d \leq 4$, respectively (see Fig. S3).

So, the dimensionality of optimal behavior is related to the dimensionality of observed behavior in a general sense: the dimensionality of optimal behavior directly relates to task complexity, which in turn may affect the behavior of animals in those tasks.

2.11

Another reason it might not generalize to some tasks is that the assumption that data can be separated into folds for cross-validation. In many tasks, there are long-term dependencies on the data that make such approaches non accessible. Even when such dependencies exist (e.g. if participants do multiple independent blocks of the same task), meta-learning across blocks is frequent.

We believe that, in the described scenarios, RNNs also demonstrate greater generalizability compared to cognitive models. In the three human datasets we studied, each subject contributes only a single block of choices. Each block, due to learning effects, cannot be subdivided into multiple folds for cross-validation purposes. To address this issue, we have proposed two training-validation-testing split protocols. In the interspersed split protocol (see Fig. 2 – copied as Fig. R10 – and Methods), for all models examined, we use all available trials as model inputs, but separate training trials, validation trials, and test trials as model targets. The protocol in Methods starts from: line 877:

In the three human datasets, each subject only performs one block of 100-200 trials. In the standard practice of cognitive modeling, the cognitive models are trained and tested on the same block, leading to potential overfitting and exaggerated performance. While it is possible to directly segment one block into three sequences for training, validation, and testing, this might introduce undesired distributional shifts in the sequences due to the learning effect. To ensure a fair comparison between RNNs and cognitive models, here we propose a new interspersed split protocol to define the training, validation, and testing trials, similar to the usage of goldfish loss to prevent the memorization of training data in language models¹⁴. Specifically, we randomly sample without replacement $\sim 75\%$ trial indexes for training, $\sim 12.5\%$ trial indexes for validation, and $\sim 12.5\%$ trial indexes for testing (three-armed reversal learning task: 120/20/20; four-armed drifting bandit task: 110/20/20; original two-stage task: 150/25/25). We then feed in the whole block of trials as the model’s inputs, obtain the output probabilities for each trial, and calculate the training, validation, and testing losses for each set of trial indexes, separately. This protocol guarantees the identical distribution between three sets of trials.

In the cross-subject split protocol (see Fig. S39 – copied as Fig. R11– and Methods), we train teacher networks on subjects from training data and tested them on held-out subjects. We then train the student networks to predict teacher’s choice probabilities for a single subject (from the teacher’s test data), and tested student networks to predict actual choices of the subject. In both protocols, our RNNs outperformed these cognitive models. With these two protocols, our RNNs are better equipped than the cognitive models to extract unknown long-term dependencies, to analyze meta-learning across blocks (when there are multiple blocks per subject), and to generalize to unseen subjects.

2.12

It would be important to acknowledge that the rule discovery process would be served by researchers analyzing data more systematically in a model independent way, rather than only focusing on reproducing unique model-predicted behavioral patterns. The discovered rules given in the paper, for example, should be identifiable by careful data visualization.

We have addressed this point with meticulous data analysis. Please see model recovery/identifiability, parameter recovery, and model validation in our reply to 1.1 and 1.2.

2.13

Drift to other rule – the forgetting rule implemented by the authors drifts to 0. This is not what all models use – for example Collins & Frank 2012 instead drift towards initial value (which is not 0, but a prior expected value in between 0 and 1), such that bad outcomes are forgotten towards better values, and good ones toward lower values, indeed both drift towards indifference. Have authors considered this more reasonable version of the forgetting capture behavior as well as the “drift to the other” one? If so, the discovery would only be due to a poor implementation choice in the first place.

The forgetful model-free RL model, as proposed in Collins & Frank 2012¹⁵, is added in the revision, which exhibited the worst predictive performance among all two-dimensional models (including our original model-free RL models with decay) (Fig. R13). We note that the forgetful model-free RL model dictates that the *unchosen* action value drifts towards a non-zero initial value. In contrast, our drift-to-other rule specifies that the *chosen* action value, *after receiving no reward*, drifts towards the other action value. These two rules are qualitatively different.

This model is mentioned in the revised manuscript starting from line 558

Model-free strategy with value forgetting to mean (d=2). This model is the “forgetful model-free strategy” proposed in¹⁵. The chosen action value is updated as in the previous model. The unchosen action value $Q_t(\bar{a}_{t-1})$, instead, is gradually forgotten to a initial value ($\tilde{V} = 1/2$):

$$Q_t(\bar{a}_{t-1}) = DQ_{t-1}(\bar{a}_{t-1}) + (1 - D)\tilde{V}, \quad (R3)$$

where D is the value forgetting rate ($0 \leq D \leq 1$).

Figure R13: Predictive performances of models fitted to one example monkey.

2.14

Fig 7 Meta learning RL. The claim that task optimized RNNs have not been analyzed using a dynamical systems approach is wildly incorrect (both in general and in the neuro-cognitive modelling field).

Thank you for your insightful comment. We do not find a sentence in the manuscript that directly says “task optimized RNNs have not been analyzed using a dynamical systems approach”. The sentence most relevant to this point is: “We note that this approach is distinct from another common use of neural networks in neuroscience, where parameters are adjusted to achieve optimal performance on a specific task. In such cases, researchers compare the behavior and neural representations of task-optimized networks to those of biological organisms, aiming to gain insights into the latter. However, without a rigorous framework guiding these comparisons, the insights generated by this approach are often limited.” We agree that dynamical systems approaches are frequently applied to analyze RNNs in the neuroscience field. In this context, we were mostly referring to meta-RL models (e.g.,¹⁶⁻¹⁸). We acknowledge that the phrasing was imprecise and could lead to misunderstanding. To address this, we removed the sentence in question from the original text.

2.15

Stable and unstable fixed points should be represented differently.

Thank you for your suggestion. All the fixed points reported in the main text are stable fixed points (attractors). We additionally updated Fig. S37 – copied as Fig. R14– in the revised manuscript.

The stability is mentioned in the revised manuscript starting from line 216

Model-free RL exhibits three types of stable fixed points (attractors), corresponding to high preference for A_1 , high preference for A_2 , and indifference.

2.16

Are there no analytic results for the types of dynamics 1d and 2d GRUs can represent (# number of fixed points, oscillations etc.) beyond just sampling randomly (fig S15)? See for example Jordan, Sokol, Park ICLR 2019 (not peer-reviewed)

Thank you for the insightful reference. This paper explores the continuous limit of low-dimensional GRUs. Although this continuous limit effectively captures certain dynamical behaviors (e.g., attractors) in our tasks, it does not account for other behaviors (e.g., oscillations between actions in some one-dimensional discrete dynamical systems). We have included this reference in the revised manuscript to acknowledge these theoretical considerations starting from line 355

Figure R14: **Possible phase portraits captured by one-unit GRU models.** In the GRU model, the effect of a constant input can be absorbed into the input biases (see Methods). We thus can determine its logit-change functions from the parameters (weights and biases) while ignoring the constant input. We randomly sampled a few sets of parameters (i.i.d. from a uniform distribution between -10 and 10) to generate phase-portrait curves (here the logits are normalized by the maximum). Empirically, we observed that these phase portraits have at most three fixed points where the logit change vanishes (e.g., green and blue curves and crosses; \times crosses for stable fixed points, and $+$ crosses for unstable fixed points), indicating that the complexity of dynamics that can be captured is limited by the expressive power of one-unit GRUs.

However, the GRU updating equation may limit the complexity of dynamics captured by tiny RNNs (see theoretical analysis in¹⁹ and Fig. S37 for the case of one-dimensional discrete dynamics)

2.17

There was surprisingly little focus on individual differences (only in supplement and little discussion). It seems like one of the main strengths of tiny GRUs is the ability to fit to individuals' data, and bring a more nuanced analysis of individual fits (e.g., in what regions of the phase plain do the animals differ?)

Thank you for the suggestion. We have presented different aspects for individual differences (e.g., logit patterns, dimensionality, etc.) for subjects in reversal learning task (Fig. S14, Fig. S3a,b – copied here as Fig. R15), in two-stage task (Fig. S15, Fig. S3c,d), in transition-reversal two-stage task (Fig. S16, Fig. S3e), in three-armed reversal learning task (Fig. S9a, Fig. S29), in four-armed drifting bandit task (Fig. S9b, Fig. S32), and in original two-stage task (Fig. S9c, Fig. S35).

We mentioned these in the revised manuscript from line 362

The ability of tiny RNNs to accurately model individual-subject behavior makes them promising for studying individual differences in decision-making (e.g., Fig. S14, S3, S9, S15, S16), a key aspect of computational psychiatry

Reviewer 3

Summary

Ji-An Li and colleagues use tiny RNNs to capture and understand learning and decision making in rodents and non-human primates. The small size of these RNNs greatly facilitates the interpretability of the networks' properties. Moreover, the authors make another important contribution by demonstrating how tools used to measure dynamical systems can be applied to neuroscience/psychology/artificial intelligence questions and data to represent and further understand model predictions and agents' behaviors. The data and conclusions presented in the current manuscript are novel, important, amazing on their own in terms of the insights they bring to cognitive sciences dealing with either biological or artificial agents. However, this is just the tip of the iceberg. The combination of minimally complex neural networks with interpretive tools from dynamical systems has massive potential to answer longstanding questions and open up numerous new avenues of inquiry.

It has already shown that canonical approaches to modeling learning and decision making fail to explain several aspects of rodent and non-human primate behavior. That result is perhaps not surprising because we know biological agents deviate from (constraint-free) optimal learning and decision rules. Crucially though, this paper gives us a new set of tools to better identify and then understand how and why biological agents' behavior diverges from normative approaches.

a Reversal learning task (Bartolo et al.)

c Two-stage task (Miller et al.)

e Novel two-stage task (Akam et al.)

b Reversal learning task (Akam et al.)

d Two-stage task (Akam et al.)

Figure R15: Tiny RNNs outperform classical cognitive models in predicting animals' choices at the individual level, providing estimated dimensionalities in three tasks. (a-e) The predictive performance of models for one example animal (top) and the distribution of the number d_* of dynamical variables for each animal (bottom). The estimated d_* is determined such that an RNN model (GRU) with $d = d_*$ (but not $d > d_*$) dynamical variables significantly outperforms all RNN models with $d < d_*$ dynamical variables. (a) Monkeys (Bartolo et al.) performing the reversal learning task: for one example monkey, the GRU model with $d = 2$ outperforms all other models (top). Both monkeys have a dimensionality of $d_* = 2$ (bottom). (b) Mice (Akam et al.) performing the reversal learning task: for one example mouse, the GRU model with $d = 1$ outperforms all other models. Seven mice have a dimensionality of $d_* = 1$ and the remaining three have a dimensionality of $d_* = 2$ (bottom). (c) Rats (Miller et al.) performing the two-stage task: for one example rat, the GRU model with $d = 2$ outperforms all other models. Four rats have a dimensionality of $d_* = 2$ (bottom). (d) Mice (Akam et al.) performing the two-stage task: for one example mouse, the GRU model with $d = 2$ outperforms all other models. Eight of ten mice have a dimensionality of $d_* = 2$ (bottom). (e) Mice (Akam et al.) performing the transition-reversal two-stage task: for one example mouse, the GRU model with $d = 4$ outperforms all other models. Most mice have a dimensionality d_* between 2 and 4 (bottom). Each model family may include multiple models, leading to two or more identical markers for a given d .

If there is a downside to this paper, it is that there is so many interesting findings and implications from this work that the authors must treat some of them rather superficially. To be clear, I think they've achieved a good balance of depth and breadth given space constraints. It is obvious that this work is just the beginning of a super exciting phase of discovery that will be spurred on by these innovations and results.

I don't have any major concerns with this work, but here are some points that I think are worth further consideration.

Thank you for your positive comments. We have incorporated your suggestions in the revised manuscript.

3.1

This may be beyond the scope of a paper already crammed full of cool new things, but one thing I'd be interested to read the authors' thoughts on is how to aggregate and/or compare the RNN fits across groups of agents. The paper focuses on fits to individual agents. Most often in studies of biological agents, the goal is to compare agents in terms of some categorical variable (intervention vs control, diseased vs healthy, etc) or a continuous measure of individual differences. I think there are a number of ways one could go about using either the fitted networks and/or their dynamical properties to compare and contrast agents. Could the authors give some guidance on the most appropriate methods in this regard, or ideally add an example implementation to the paper? I realize that including an implementation of biological agents is probably infeasible given the low number of animals in each study used here. However, even an implementation with simulated agents designed to differ in some way may be very helpful for many readers.

Thank you for the insightful suggestions. We agree that aggregating individual results across agents could provide valuable insights. With the inclusion of three human datasets in our study, we have a better opportunity to demonstrate such aggregation. To this end, we can characterize the distribution of individual dimensionality for each dataset (Fig. S3, S9). Furthermore, by performing the dynamical regression on each individual, we can aggregate the resulting regression coefficients to generate group-level descriptions (e.g., violin plots in Fig. 4 —copied as Fig. R12 c-e —, S29, S32, S35). When there are multiple subject groups, these group-level summaries allow for meaningful comparisons across groups.

We have mentioned these analyses in the revised manuscript, starting from line 287:

We then used the same approach to analyze human behavior in the three human tasks introduced previously (Fig. 2c), performing separate regressions for each subject-specific RNN trained with knowledge distillation.

3.2

A key point of the paper — as far as I understood it — is that the tiny RNNs that best fit and reproduce the animals' behavior on these tasks do not correspond to ANY of the canonical models proposed to underly such behavior in human and non-human animals. However, the first paragraph of the Discussion states, "our approach demonstrated that animal behavior aligns more closely with model-free RL algorithms than alternative cognitive models." What data or comparison is this statement based on? Figure 5b and c does show that the GRU is more similar to model-free than latent state, but still qualitatively different than both, in one monkey on

the reversal learning task. However, Figure 5e shows that MF predicts different preference orders than the GRU on the Miller two-stage task after uncommon transitions, while both latent-state and model-based at least get the preference ordering correct. All cognitive models are clearly a poor quantitative fit to the pattern generated by the GRU, but it is hard to see how we could judge the pattern to be closest to MF (in that task at least). Again, the point of the paper is that none of these cognitive algorithms are a good qualitative or quantitative fit to animals' behavior, even on these very simple behavioral tasks. The tiny RNN approach lets us discover and describe functions that actually match animals' behavior. So the sentence noted above seems to me to be both misleading and unnecessary.

Thank you for the suggestion. In the three animal tasks, we observed that the animal strategies often exhibit linear features in logit patterns, which are more akin to those of model-free RL agents (Fig. 3b left), and contrast with the non-monotonic logit patterns typical of Bayesian-like agents (Fig. 3b right). In other words, we found no evidence to support the use of Bayesian-like strategies in these tasks, which is what we intended to convey in the original Discussion.

That said, we agree that the animal strategies in the reversal learning task and the two-stage task exhibit several qualitative differences compared to model-free RL agents. To avoid any potential confusion, we have removed the sentence in question from the revised manuscript.

References

- [1] Thomas Akam, Rui Costa, and Peter Dayan. Simple plans or sophisticated habits? state, transition and learning interactions in the two-step task. *PLoS computational biology*, 11(12):e1004648, 2015.
- [2] Hans Sagan. *Space-filling curves*. Springer Science & Business Media, 2012.
- [3] Steven H Strogatz. *Nonlinear dynamics and chaos: with applications to physics, biology, chemistry, and engineering*. CRC press, 2018.
- [4] Saurabh Vyas, Matthew D Golub, David Sussillo, and Krishna V Shenoy. Computation through neural population dynamics. *Annual review of neuroscience*, 43:249–275, 2020.
- [5] Mehrdad Jazayeri and Srdjan Ostojic. Interpreting neural computations by examining intrinsic and embedding dimensionality of neural activity. *Current opinion in neurobiology*, 70:113–120, 2021.
- [6] Peiran Gao, Eric Trautmann, Byron Yu, Gopal Santhanam, Stephen Ryu, Krishna Shenoy, and Surya Ganguli. A theory of multineuronal dimensionality, dynamics and measurement. *BioRxiv*, page 214262, 2017.
- [7] William Bialek. On the dimensionality of behavior. *Proceedings of the National Academy of Sciences*, 119(18):e2021860119, 2022.
- [8] Surya Ganguli. Measuring the dimensionality of behavior. *Proceedings of the National Academy of Sciences*, 119(43):e2205791119, 2022.
- [9] Yoav Ger, Moni Shahar, and Nitzan Shahar. Using recurrent neural network to estimate irreducible stochasticity in human choice-behavior. *eLife*, 13, 2024.
- [10] Matan Fintz, Margarita Osadchy, and Uri Hertz. Using deep learning to predict human decisions and using cognitive models to explain deep learning models. *Scientific reports*, 12(1):4736, 2022.
- [11] Maria K Eckstein, Christopher Summerfield, Nathaniel D Daw, and Kevin J Miller. Predictive and interpretable: Combining artificial neural networks and classic cognitive models to understand human learning and decision making. *BioRxiv*, pages 2023–05, 2023.
- [12] Kevin J Miller, Matthew M Botvinick, and Carlos D Brody. Value representations in the rodent orbitofrontal cortex drive learning, not choice. *Elife*, 11:e64575, 2022.
- [13] Thomas Akam, Ines Rodrigues-Vaz, Ivo Marcelo, Xiangyu Zhang, Michael Pereira, Rodrigo Freire Oliveira, Peter Dayan, and Rui M Costa. The anterior cingulate cortex predicts future states to mediate model-based action selection. *Neuron*, 109(1):149–163, 2021.
- [14] Abhimanyu Hans, Yuxin Wen, Neel Jain, John Kirchenbauer, Hamid Kazemi, Prajwal Singhania, Siddharth Singh, Gowthami Somepalli, Jonas Geiping, Abhinav Bhatele, et al. Be like a goldfish, don't memorize! mitigating memorization in generative llms. *arXiv preprint arXiv:2406.10209*, 2024.

- [15] Anne GE Collins and Michael J Frank. How much of reinforcement learning is working memory, not reinforcement learning? a behavioral, computational, and neurogenetic analysis. *European Journal of Neuroscience*, 35(7):1024–1035, 2012.
- [16] Jane X Wang, Zeb Kurth-Nelson, Dharshan Kumaran, Dhruva Tirumala, Hubert Soyer, Joel Z Leibo, Demis Hassabis, and Matthew Botvinick. Prefrontal cortex as a meta-reinforcement learning system. *Nature neuroscience*, 21(6):860–868, 2018.
- [17] Safa Alver and Doina Precup. What is going on inside recurrent meta reinforcement learning agents? *arXiv preprint arXiv:2104.14644*, 2021.
- [18] Vladimir Mikulik, Grégoire Delétang, Tom McGrath, Tim Genewein, Miljan Martic, Shane Legg, and Pedro Ortega. Meta-trained agents implement bayes-optimal agents. *Advances in neural information processing systems*, 33:18691–18703, 2020.
- [19] Ian D Jordan, Piotr Aleksander Sokół, and Il Memming Park. Gated recurrent units viewed through the lens of continuous time dynamical systems. *Frontiers in computational neuroscience*, 15:678158, 2021.

RESPONSE TO REVIEWERS

We thank the reviewers for their insightful feedback, which has been instrumental in guiding the revisions and improvements to our manuscript.

This document includes detailed responses to each reviewer comment. Editor and reviewer comments are shown in black, our responses in blue, and quotes from the revised manuscript in green.

Reviewer 1

Summary

The authors have done an impressive job of addressing the criticisms raised by myself and the other reviewers. I believe that the revised manuscript is considerably improved, and only have a few remaining concerns.

We sincerely appreciate your positive feedback on our revised manuscript. Below, we provide a detailed, point-by-point response to each of your comments.

1.1

The first has to do with magnitude of improvements in fit. The tiny RNNs fit better, but not by much. It is certainly not clear whether the differences between the best RNN and the best cognitive model are meaningfully different. From figure 1e it appears the differences between RNN models and cognitive models are incredibly small, in real terms (1% difference in likelihood of chosen action per trial) – raising question as to whether the difference is meaningful – I guess, to some extent, this is addressed by the fact that the authors show that they can use the RNN fits and analysis methods to inspire changes to the cognitive models that improve fits... however, I think at the higher level, there is always a question of whether one is discovering principles (ie. things we expect to generalize beyond this particular task or model system), and while I don't think that the authors necessarily need to prove that their method does this, I think some balanced discussion about the issue would give the reader a better perspective on what the results mean.

We appreciate the reviewer's observation regarding the magnitude of improvements in fit between tiny RNNs and classical cognitive models. While the differences in predictive likelihood may appear small, we have shown that these differences are statistically robust and can have meaningful implications in tasks involving hundreds or thousands of trials. More importantly, the fit improvements offered by our approach are remarkably consistent across multiple datasets, tasks, and species, suggesting they reflect genuine advantages of the tiny RNN approach rather than chance variations. For example, in the reversal learning tasks (where we find the smallest absolute improvements in model fitting), tiny RNNs significantly outperformed the best cognitive models with the same number of dynamical variables in monkeys (Bartolo et al.) measured by cross-entropy ($d = 1$: $M = -0.008$, 95% CI $[-0.012, -0.004]$, $p = 3e - 4$; $d = 2$: $M = -0.015$, 95% CI $[-0.021, -0.010]$, $p = 1e - 5$), and mice (Akam et al.; $d = 1$: $M = -0.011$, 95% CI $[-0.013, -0.009]$, $p = 2e - 20$; $d = 2$: $M = -0.005$, 95% CI $[-0.007, -0.003]$, $p = 7e - 6$). All other tasks analyzed in the manuscript show even more substantial benefits of tiny RNNs.

We also agree with the reviewer that the primary value of our approach lies not in quantitative improvements to model fits, but rather in uncovering novel behavioral patterns not easily captured by traditional approaches. Since these patterns are largely interpretable, they can be easily verified in different tasks or model systems, supporting the discovery of truly generalizable principles of behavior. Note that such discoveries would be much less likely with a black-box method offering similar fit improvements as ours. This highlights that the value of our approach lies in combining fit improvements with interpretability.

To expand on this point, we believe that the discovery of previously overlooked cognitive mechanisms has greater potential to advance our scientific knowledge than improvements in predictions without the distillation of these improvements into generalizable knowledge. Consider an example from physics: while Newtonian mechanics accounts for most of Mercury's perihelion precession, general relativity reveals an additional 43 arcseconds per century – a tiny fit improvement that, nonetheless, revealed previously unknown mechanisms. Note that we are not claiming that our results are comparable in importance to the discovery of general relativity; rather, we use this example to emphasize that extracting generalizable knowledge from data, even when fit improvements are modest, can be more valuable than focusing solely on the magnitude of predictive performance gains.

In our manuscript, the generalizability of our *approach* is evidenced by its success across diverse experimental paradigms, from simple reversal learning to complex multi-stage decision tasks, and across multiple species including mice, rats, monkeys, and humans. The generalizability of the *specific strategies*, in turn, is evidenced by the fact that the same cognitive

mechanisms were observed in different folds of trials for a single subject. Additionally, some mechanisms (like “drift-to-the-other” updating) were found across multiple individuals supporting the idea that they reflect general learning strategies rather than task-specific effects.

To make this point about generalizability concrete, for the reversal learning task, we have previously analyzed datasets from two species – monkeys (Bartolo dataset) and mice (Akam dataset) – performing task variants differing in state-reward types, reward probabilities, and reversal designs (see Methods). Similarly, for the two-stage task, we analyzed datasets from two species – rats (Miller dataset) and mice (Akam dataset) – performing task variants differing in state-reward types and block-transition designs (see Methods). Although there were some individual differences, we found that these strategies consistently recurred across animal species and task variants (Fig. R1), suggesting the generalizability of RNN-discovered insights.

Figure R1: **Generalizability of identified features across task variants and animal species.** (a-b) Logit analysis of one example monkey (Bartolo et al., left) and one example mouse (Akam et al., right) performing the reversal learning task. (a) Both individuals show state-dependent learning rate, state-dependent perseveration, and reward-dependent bias. (b) Both individuals show the pattern of the drift-to-the-other rule. (c) Preference setpoint analysis of one example rat (Miller et al., left) and one example mouse (Akam et al., right) performing the two-stage task, showing reward-induced indifference.

We have revised the discussion section in light of the reviewer’s suggestions, starting from line 343:

Our approach offered robust and statistically significant improvements over traditional methods. While the improvements in some cases were numerically modest, the value of our approach lies in its consistent success in discovering interpretable patterns across various species and tasks. Additionally, we found that the same strategies were often observed across multiple individuals (Fig. S26), supporting the idea that they reflect

general learning strategies rather than task-specific effects.

1.2

My second concern is that I still feel like the paper overstates the novelty of the mechanisms uncovered using their small RNNs. For example, the authors say:

“These three signatures — “state dependent learning rate”, “state-dependent perseveration”, and “reward-dependent bias” — are absent from all cognitive model variants considered and from any published analysis of these tasks in the literature.”

While this may be true in these particular task variants, it is not like these ideas are completely new.

I assume that by “state dependent learning rate” the authors refer to the curve in the dark blue and red lines in figure 3c, which that the logit change associated with a reward are largest when that reward was inconsistent with the action preference of the model. It may be that this hasn’t been noted in the particular reversal learning task that is studied here, but there is whole field of study on adaptive learning rates, stemming from animal models of learning (Pearce Hall) and connecting to Bayesian inference in changing environments (Nassar 2010), that shows in general terms what is observed in the RNN dynamics on rewarded trials, namely that prediction errors carry more influence on behavioral updating when they are unexpected, and therefore more likely to reflect a mismatch between stored expectations and the current environment.

I think when the authors say “Preference dependent perseveration” they are referring to the curving pink and light blue lines in fig 3c. If so, I think this effect relates to the finding that “learning rate” as implemented in humans at least, is not deterministic, but instead includes trial-to-trial variability (Findling 2019). Since this variability is not observable to the experimenter (and fit models), it means that actions provide some “clue” as to the unobservable state of the system (current level of action preference). Thus, observing an unrewarded action that is inconsistent with the “inferred” choice preferences from a model can indicate that the animal actually has a less extreme choice preference than the model-estimated one. This has been captured in cognitive models that incorporate stochastic trial by trial learning rates, which provide a better fit to data, and my guess is that it would produce something similar to the light blue/pink curves.

I’m a bit unsure what reward dependent bias means in terms of the dynamics plot – so I find it harder to comment on that one.

But the point is that I am not particularly convinced that the signatures of learning revealed by the RNN are new, they seem quite related to principles that have been identified and characterized in other contexts. Just to be clear, I do not see this as a weakness of the method or paper. If the RNN framework can reveal signatures of behavior in a single task that are closely related to principles that have been observed in other tasks, this provides evidence that it can be used as a scientific tool, not only to describe data collected within a particular task paradigm, but to generalize principles that extend beyond it. Indeed, I would be highly skeptical if the RNNs revealed a completely new set of principles that bore no resemblance to empirical and theoretical work done over the last 70 years of studying learning. That said, I DO think that the authors could provide a much more balanced discussion of the contributions of their methods, and their relationship to the broader understanding of learning that has been shaped by cognitive models, and their view of how these tools might be combined to aid scientific discovery going forward.

We appreciate the reviewer’s request that we better situate our findings within the broader literature. We agree that the mechanisms revealed by our RNNs align with broader principles of learning and decision-making, which strengthens rather than diminishes their value. Let us address each mechanism:

1. **State-dependent learning rate:** Our RNNs revealed that learning rates increase when rewards contradict current beliefs — i.e., the learning rate is larger when perceived environmental volatility is higher¹. This aligns with extensive work on adaptive learning rates, from Pearce-Hall models to Bayesian inference in volatile environments²⁻⁴. The RNNs independently discovered this principle in reversal learning, demonstrating their ability to capture fundamental learning mechanisms.
2. **State-dependent perseveration** (Fig. 3c, evident by the non-overlapping light blue and light red curves): We appreciate the reviewer’s alternative interpretation through the lens of trial-to-trial variability in learning rates⁵. When seemingly identical animal’s preferences lead to different actions, this could indeed reflect unobservable factors affecting animal’s preferences that are partially revealed by subsequent choices. This interpretation offers an interesting perspective on our findings and highlights how RNNs can bridge multiple theoretical frameworks.

3. **Reward-dependent bias:** This refers to the observation (Fig. 3c) that the attractor position for the dark blue curves ($|L_{A1,R=1}^*|$) is more extreme than for the dark red curves ($|L_{A2,R=1}^*|$), i.e., $|L_{A1,R=1}^*| - |L_{A2,R=1}^*| > 0$. In contrast, the standard action bias, which is independent of reward, predicts $|L_{A1,R=0}^*| - |L_{A2,R=0}^*| = |L_{A1,R=1}^*| - |L_{A2,R=1}^*| > 0$, a prediction not supported by Fig. 3c. While this represents a nuanced version of standard action bias, it emerged naturally from the RNN's learning process.

We have revised the manuscript to better contextualize these findings within existing literature and downplayed claims of complete novelty. As suggested by the reviewer, we now emphasize how our RNN framework independently discovers established principles while providing mechanistic insights into their implementation. This demonstrates the framework's value as a scientific tool that can both validate existing theories and reveal how they manifest in specific behavioral contexts. The revised section starts from line 347:

Several strategies identified by the RNNs had been overlooked by cognitive models specifically designed for these tasks. They do, however, reflect established principles of learning and decision-making, such as adaptive learning rates²⁻⁴, trial-to-trial variability⁵, and side biases⁶ (see Supplementary Discussion 2.1) while providing mechanistic insights into their implementation in these specific tasks. This suggests that our framework may accelerate scientific discovery, not only by modeling data specific to a given task, but also by uncovering generalizable principles that transcend task-specific contexts.

1.3

I still have some reservations about using the same x-axis to compare cognitive and RNN models in figure 1E. I appreciate the point that the authors are making and their response to my specific concern in the last round. However, I would just ask that the authors to 1) state clearly the working definition of dimensionality that they are using in the paper, 2) differentiate it from the definition that relates to expressivity and parameters that is often used in machine learning research, and 3) acknowledge that the working definition of dimensionality (counting dynamical variables) is imperfect in its assessment of encoding dimensions in that that two different models might encode task relevant information more or less efficiently across available dynamical variables. I think it also would be fair to mention that the RNNs are incentivized to make efficient use of their dynamic variables, whereas the variables in cognitive models are constructed by hand, typically without any prioritization of such efficiency... though I guess this may be a matter of opinion.

Thank you for these specific suggestions. We agree that greater clarity is needed regarding dimensionality. In our work, dimensionality refers to the minimum number of dynamical variables required to predict future behavior — specifically, the variables that evolve over time and summarize past experiences to guide actions. This notion of dimensionality is fundamental in the mathematical theory of dynamical systems and is often used in computational neuroscience, despite being rarely discussed in cognitive science. However, as indicated by the reviewer, our usage of dimensionality indeed differs from expressivity (parameter count) used in machine learning, which determines the complexity of mappings a model can implement. The connection between these notions of dimensionality and expressivity has been studied in computational neuroscience using low-rank RNNs, demonstrating their complementary roles⁷.

While counting dynamical variables provides a principled comparison metric, we acknowledge its limitations. Different models may encode task-relevant information with varying efficiency across their variables. RNNs, with a constrained state space, are inherently incentivized to use their variables efficiently. In contrast, cognitive models use hand-designed variables based on theoretical constructs, without explicit optimization for dimensional efficiency.

We incorporated your suggestions into the revised manuscript, starting from line 355:

While counting dynamical variables provides a principled comparison metric, different models may encode task-relevant information with varying efficiency — RNNs are optimized for dimensional efficiency through training, whereas cognitive models use hand-designed variables based on theoretical constructs. Our study highlights the important distinction between the dimensionality of dynamics and model expressivity (e.g., nonlinearity and parameter count), which determines the complexity of input-output mappings a model can implement. These two concepts give complementary notions of model complexity in dynamical systems⁷⁻¹⁰. The identification of dynamical variables from experimental data is an active research area across neuroscience, complex systems, and physics, with efforts to extract key variables from neural data, physical system recordings, and multiscale complex systems¹⁰⁻¹³.

1.4

Minor:

“In particular, RNNs outperformed all ideal Bayesian observer models, which perform exact inference based on knowledge of the task structure, suggesting that animal behavior in these tasks is not optimal.”

The authors present this as a finding of their method, but this is an observation that could be made simply from comparing the cognitive models – and indeed, has been made previously in the literature many times.

Thank you for pointing this out. We have revised the manuscript to reframe this point, emphasizing that this observation aligns with previous findings in the literature, starting from line 115:

In particular, RNNs outperformed all ideal Bayesian observer models, which perform exact inference based on knowledge of the task structure, suggesting that animal behavior in these tasks is suboptimal (aligning with previous findings in the literature; also see Supplementary Discussion 2.1).

1.5

It is noteworthy that Animal behavior on 2 step task looks considerably different from human behavior on 2-step tasks... perhaps this

The reviewer’s comment appears incomplete, but we have addressed the observed differences based on our interpretation of the query. We note that the “two-stage task” used for animals (Miller dataset and Akam dataset) is a simplified version of the “original two-stage task” used for humans (Gillan dataset). Specifically, the “two-stage task” omits the second-stage actions from the “original two-stage task” to improve the relative payoff for model-based strategies compared to chance-level and model-free strategies¹⁴. As a result, these task design differences may directly lead to distinct cognitive strategies.

1.6

“Even after convergence, we identified subtle deviations from the Bayesian inference model (Fig. 5b-c). “

I think the authors mean to refer to figures 5a-b here?

Thank you, we have corrected the figure references in the main text, ensuring they now appropriately refer to Figures 5a-b.

Reviewer 2

Summary

We thank the authors for a very thorough and responsive revision that we think has very significantly improved the manuscript. We appreciated the new tasks analyses, the new dynamical regression method, the new cross-validation method, and the new model comparison metrics. We agree the paper is now ready for publication. We add here three comments; two of them are minor and more curiosity on our part and shouldn’t hold back publication. One we think is important to handle before publication, but should be a formality for the authors to handle.

Thank you for your encouraging feedback and your recognition of the improvements made in the manuscript. Below, we provide a detailed, point-by-point response to each of your comments.

2.1

1. Code availability. Currently, this is stated as “available from the lead author upon request”. In 2024, for a paper fully based on developing a new method, this is absolutely not sufficient. Full, reproducible code should be posted on an open repository and fully available. We think this needs to be changed before publication.

We agree with the reviewer on the importance of making our code accessible before publication. This comment prompted us to document our code, so that it can be easily understood and adapted by anyone interested. The code is publicly available on github: <https://github.com/jil095/tinyRNN>.

The new statement of code availability can be found on line line 408:

The code used to reproduce the results in this paper is available at <https://github.com/jil095/tinyRNN>.

2.2

2. Minor quibble/question, does not need to be addressed before publication. We are uncertain on theorem 1 and 2 in section 2.3 of the supplement (and Figure S40). We're not confident here, but we feel this is somewhat "hand wavy". It feels like a real trained embedding of a one dimensional variable into two dimensions could be formed to maximize similarity of adjacent points along the natural manifold of activity in that two dimensional space better than Hilbert Curves which appear quite naive. Not clear these proofs are relevant to the paper given the nature of trained representations.

We apologize for any confusion caused. We believe these are several related but distinct questions. Consider the state space \mathbb{R}^2 of the two-dimensional RNN, the 1D-to-2D mapping $G : \mathbb{R} \rightarrow \mathbb{R}^2$, and the one-dimensional dynamics $f : \mathbb{R} \rightarrow \mathbb{R}$.

1. Can a one-dimensional variable accurately represent points within this two-dimensional space using a linear mapping G (up to a precision loss)? Our answer is no. Our example in Statement 1 (Supplementary Discussion 2.3) has refused this possibility. In this case, an optimally trained one-dimensional variable will capture the principal dimension of the two-dimensional space, but it cannot represent the orthogonal dimension at the same time, inevitably resulting in significant representation loss.

2. Can a one-dimensional variable accurately represent points within this two-dimensional space using a nonlinear mapping G (up to a precision loss)? Our answer is yes. We have provided the construction using space-filling curves in Statement 2 (Supplementary Discussion 2.3), and other nonlinear mappings G might work as well (this question was raised by Reviewer 1 in the first-round review).

3. Can a one-dimensional variable accurately represent points within this two-dimensional space (up to a precision loss) and be compatible with simple one-dimensional dynamics (e.g., continuous or smooth)? Our answer is no. We provided theoretical analyses for space-filling curves in Statement 2: a one-dimensional variable can accurately represent points within this two-dimensional space, but is not compatible with continuous, learnable one-dimensional dynamics. While a complete proof in the most general case remains open, we conjecture that if the mapping G is highly nonlinear, then the corresponding one-dimensional dynamics $f : \mathbb{R} \rightarrow \mathbb{R}$ must exhibit a comparable degree of nonlinearity. Consequently, such dynamics are inherently fragile and lack robustness, which in turn precludes effective learning via gradient descent methods.

We agree with the reviewer that Hilbert curves are merely a (fairly regular and easily understandable) possibility for a highly nonlinear embedding. While trained embeddings may differ in their geometry, we believe that it is an instructive example to consider. We have added the clarification to the revised Supplementary Discussion 2.3.

2.3

3. Line 190: "We found that student RNNs with 5-20 units provided the best fit to human behavior, despite the limited trials per subject." It looks like authors only tested up to 20 units, which leaves open the possibility that more units would have fit even better. However we agree though that this doesn't diminish the main overall claim of this section that tiny RNNs improve upon cognitive models of equal dim for more fully encapsulating behavioral strategies, so again, this is a minor quibble that shouldn't block acceptance.

Thank you for raising this interesting observation. To address this, we have analyzed the performance of 50-unit student RNNs (Fig. R2). Our results indicate that, depending on the task, 50-unit RNNs performed similarly (or slightly better or worse) compared to 20-unit RNNs. These findings have been incorporated into the revised manuscript, starting from line 188:

We found that student RNNs with 5-20 units provided the best fit to human behavior (see Fig. S9 for 50-unit RNNs' similar performance), despite the limited trials per subject.

2.4

(Remarks on code availability):

Code is not available, so that code review was not possible. See main review.

We have addressed this concern in our earlier response regarding code availability.

Reviewer 3

Summary

Figure R2: **Model performance using knowledge distillation, showing 50-unit RNNs, compared to Fig. 2.** (a) Three-armed reversal learning. (b) Four-armed drifting bandit. (c) Original two-stage.

The authors have done extensive work to address all reviewers' comments. My concerns have been sufficiently addressed and I have no further comments. I still think this a great paper.

Thank you for your kind words and for recognizing the extensive efforts we put into addressing the reviewers' comments.

References

- [1] Payam Piray and Nathaniel D Daw. A model for learning based on the joint estimation of stochasticity and volatility. *Nature communications*, 12(1):6587, 2021.
- [2] Timothy EJ Behrens, Mark W Woolrich, Mark E Walton, and Matthew FS Rushworth. Learning the value of information in an uncertain world. *Nature neuroscience*, 10(9):1214–1221, 2007.
- [3] Matthew R Nassar, Robert C Wilson, Benjamin Heasley, and Joshua I Gold. An approximately bayesian delta-rule model explains the dynamics of belief updating in a changing environment. *Journal of Neuroscience*, 30(37):12366–12378, 2010.
- [4] J Yu Angela and Peter Dayan. Uncertainty, neuromodulation, and attention. *Neuron*, 46(4):681–692, 2005.
- [5] Charles Findling, Vasilisa Skvortsova, Rémi Dromnelle, Stefano Palminteri, and Valentin Wyart. Computational noise in reward-guided learning drives behavioral variability in volatile environments. *Nature neuroscience*, 22(12):2066–2077, 2019.
- [6] Kevin J Miller, Matthew M Botvinick, and Carlos D Brody. Value representations in the rodent orbitofrontal cortex drive learning, not choice. *Elife*, 11:e64575, 2022.
- [7] Alexis Dubreuil, Adrian Valente, Manuel Beiran, Francesca Mastrogiuseppe, and Srdjan Ostojic. The role of population structure in computations through neural dynamics. *Nature neuroscience*, 25(6):783–794, 2022.
- [8] Francesca Mastrogiuseppe and Srdjan Ostojic. Linking connectivity, dynamics, and computations in low-rank recurrent neural networks. *Neuron*, 99(3):609–623, 2018.
- [9] Manuel Beiran, Alexis Dubreuil, Adrian Valente, Francesca Mastrogiuseppe, and Srdjan Ostojic. Shaping dynamics with multiple populations in low-rank recurrent networks. *Neural Computation*, 33(6):1572–1615, 2021.
- [10] Adrian Valente, Jonathan W Pillow, and Srdjan Ostojic. Extracting computational mechanisms from neural data using low-rank rnns. *Advances in Neural Information Processing Systems*, 35:24072–24086, 2022.
- [11] Christopher Langdon and Tatiana A Engel. Latent circuit inference from heterogeneous neural responses during cognitive tasks. *bioRxiv*, pages 2022–01, 2022.
- [12] Boyuan Chen, Kuang Huang, Sunand Raghupathi, Ishaan Chandratreya, Qiang Du, and Hod Lipson. Automated discovery of fundamental variables hidden in experimental data. *Nature Computational Science*, 2(7):433–442, 2022.
- [13] Pantelis R Vlachas, Georgios Arampatzis, Caroline Uhler, and Petros Koumoutsakos. Multiscale simulations of complex systems by learning their effective dynamics. *Nature Machine Intelligence*, 4(4):359–366, 2022.

- [14] Thomas Akam, Rui Costa, and Peter Dayan. Simple plans or sophisticated habits? state, transition and learning interactions in the two-step task. *PLoS computational biology*, 11(12):e1004648, 2015.